# RNE: PLUG-AND-PLAY DIFFUSION INFERENCE-TIME CONTROL AND ENERGY-BASED TRAINING

**Jiajun He**
University of Cambridge
jh2383@cam.ac.uk

**José Miguel Hernández-Lobato**
University of Cambridge
jmh233@cam.ac.uk

**Yuanqi Du**[†]
Cornell University
yuanqidu@cs.cornell.edu

**Francisco Vargas**[†]
Xaira Therapeutics
vargfran@gmail.com

## ABSTRACT

Diffusion models generate data by removing noise gradually, which corresponds to the time-reversal of a noising process. However, access to only the denoising kernels is often insufficient. In many applications, we need the knowledge of the marginal densities along the generation trajectory, which enables tasks such as inference-time control. To address this gap, in this paper, we introduce the RADON-NIKODYM ESTIMATOR (RNE). Based on the concept of the *density ratio* between path distributions, it reveals a fundamental connection between marginal densities and transition kernels, providing a flexible plug-and-play framework that unifies (1) diffusion density estimation, (2) inference-time control, and (3) energy-based diffusion training under a single perspective. Experiments demonstrate that RNE delivers strong results in inference-time control applications, such as annealing and model composition, with promising inference-time scaling performance, and achieves a simple yet efficient regularisation for training energy-based diffusion models. Additionally, our proposed RNE is modality-agnostic and applicable not only to continuous diffusion models but also to their discrete diffusion counterparts.

## 1 INTRODUCTION AND BACKGROUND

Diffusion models (Ho et al., 2020; Song et al., 2021a;b) are a class of flexible generative models that excel in generating high-quality samples from complex data distributions, and have had a broad impact across a wide range of applications from image (Rombach et al., 2022; Karras et al., 2022), video (Ho et al., 2022) and text (Austin et al., 2021) generation, designing novel proteins (Watson et al., 2023), materials (Zeni et al., 2025) and capturing transient structures in chemical reactions (Duan et al., 2023). Diffusion models rely on a pair of forward and backwards stochastic differential equations (Eqs. (1) and (2)) to transport between data distribution and a tractable prior distribution, commonly selected as Gaussian. They then parametrise the *score* function $\nabla \log p_t$ with a time-dependent network, and when trained to optimality, Eqs. (1) and (2) will be the time-reversal of each other. This allows us to generate high-quality samples by simulating the backward SDE in Eq. (2) starting from a Gaussian. Equivalent to diffusion models is the more flexible stochastic interpolants (Albergo et al., 2023; Ma et al., 2024; Gao et al., 2025) parametrisation of diffusion models (Eqs. (3) and (4)).

| DM Characterisation - Fixed $\sigma_t$ | | SI Characterisation - $\forall \epsilon_t > 0$ |
|---|---|---|
| $dX_t = f_t(X_t)dt + \sigma_t \overrightarrow{dW_t}$ $\quad$ (1) | $\xrightarrow[\forall \epsilon_t \geq 0]{v := f - \frac{\sigma^2}{2} \nabla \log p}$ | $dX_t = \left( v_t(X_t) + \frac{\epsilon_t^2}{2} \nabla \log p_t(X_t) \right) dt + \epsilon_t \overrightarrow{dW_t}$ $\quad$ (3) |
| $dX_t = \left( f_t(X_t) - \sigma_t^2 \nabla \log p_t(X_t) \right) dt + \sigma_t \overleftarrow{dW_t}$ $\quad$ (2) | | $dX_t = \left( v_t(X_t) - \frac{\epsilon_t^2}{2} \nabla \log p_t(X_t) \right) dt + \epsilon_t \overleftarrow{dW_t}$ $\quad$ (4) |

In Eq. (1), $f_t$ is typically chosen as a linear function. For example, $f_t(x) = -\beta_t x$ for vaiance-preserving process (Ho et al., 2020; Song et al., 2021a) with an extra hyperparameter $\beta_t$, or $f_t \equiv 0$ for variance-exploding process (Song et al., 2021a; Karras et al., 2022). Note that one can always

---

[†]Last authors.

recover the classical DM characterisation from its SI perspective by setting $\epsilon_t \leftarrow \sigma_t$. This unifying parametrisation allows us to establish theoretical connections to existing work and extend our methodology to models trained via flow matching (Albergo et al., 2023; Lipman et al., 2022).

Strictly generating samples that resemble the overall data distribution may lack practical applications. Fortunately, the progressive generation process of diffusion models naturally allows us to apply more flexible probabilistic inference, unlocking a variety of approaches and applications. For instance, in diffusion posterior sampling and inference-time steering (Dhariwal & Nichol, 2021; Ho & Salimans, 2022; Song et al., 2023a; Chung et al., 2023; Song et al., 2023b; Trippe et al., 2023; Rozet et al., 2024; Schneuing et al., 2024; Kong et al., 2025), the goal is to generate samples that satisfy specific constraints or exhibit desired attributes. In diffusion model composition (Liu et al., 2022; Du et al., 2023; Ajay et al., 2023; Biggs et al., 2024; Skreta et al., 2024; Thornton et al., 2025), multiple diffusion models are combined to produce samples with richer attributes. Also, in sampling tasks, diffusion models can be used to accelerate standard algorithms such as annealed importance sampling or parallel tempering (Doucet et al., 2022; Chen et al., 2024; Zhang et al., 2025).

While heuristic methods such as guidance can be effective for these tasks, they often introduce bias due to *ad hoc* design choices. By contrast, probabilistic inference techniques offer a principled approach to eliminating such bias and can lead to more reliable performance. A central requirement for applying these techniques is to evaluate or approximate the sample density under a pretrained diffusion model along the generation trajectory. One classic approach reformulates the diffusion process as a probability flow ODE (PF-ODE, Song et al., 2021b) and applies the instantaneous change-of-variables formula (Chen et al., 2018); however, this is computationally prohibitive, as it requires calculating the divergence of the score network at every denoising step. To address this difficulty, some works have developed sequential Monte Carlo (SMC) algorithms based on twisting functions or Feynman–Kac formulations, which bypass the need for explicit density evaluation (Wu et al., 2023; Skreta et al., 2025; Singhal et al., 2025). Other approaches introduce diffusion density estimators leveraging the Feynman–Kac formula or Itô's lemma (Huang et al., 2021; Premkumar, 2024; Karczewski et al., 2024; Skreta et al., 2024). Alternatively, one can directly train energy-parametrised diffusion models (Du et al., 2023; Phillips et al., 2024; Thornton et al., 2025; Zhang et al., 2025), which provide explicit access to the unnormalised marginals along the process.

**Our contributions.** Despite the above advances, these methods remain disparate in their scope. The connections between these approaches remain unclear, and many depend on specialised designs, which can limit their applicability. In this paper, we close this gap with RADON–NIKODYM ESTIMATOR (RNE), a *unified, flexible, and plug-and-play* framework that enables density estimation, SMC weight computation for inference-time control, and better training of energy-based diffusion.

- For inference-time control, RNE can *compute SMC weights for **any** sampling process without re-deriving the formula*, enabling a wide variety of options, such as Chung et al. (2023), Song et al. (2023b), and Singhal et al. (2025). This opens up broader design spaces and offers *better inference-time scaling performances.* For energy-based training, RNE yields a simple yet effective regulariser, significantly improving the learned energy with negligible computational overhead.

- RNE *generalises and unifies a wide range of established methods*—such as the twisted diffusion sampler (Wu et al., 2023), Feynman–Kac steering (Singhal et al., 2025), Feynman–Kac corrector (Skreta et al., 2025), guidance corrector (Lee et al., 2025), Itô density estimator (Karczewski et al., 2024; Skreta et al., 2024), Feynman–Kac density estimator (Huang et al., 2021; Premkumar, 2024), and Fokker–Planck regulariser (Plainer et al., 2025)—that may appear distinct at first glance.

- RNE is not restricted to Gaussian diffusion. It applies broadly to any generative model that admits a pair of dynamics that are time-reversal. This includes stochastic interpolants and bridge models (Shi et al., 2023; Peluchetti, 2023; Albergo et al., 2023), as well as more processes in other modalities such as continuous-time Markov chains (CTMC, Lou et al., 2023; Shi et al., 2024).

## 2 METHODS

In many applications of diffusion models, including inference-time steering or model composition, we need access to the marginal density $p_t$ at time step $t$ of the diffusion process. Unfortunately, this is generally intractable for a score-based diffusion model. Instead, in most cases, it is easy to access the transition kernels (e.g., denoising or noising kernels) of the diffusion model. Therefore, a natural question is: *can we connect the transition kernels of an SDE with its marginal densities?*

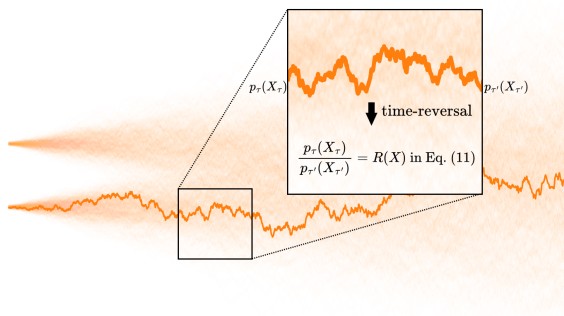
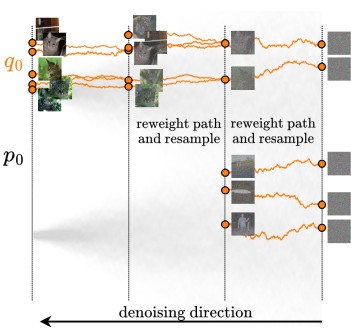

(a) Radon-Nikodym Estimator (RNE).  (b) Inference-time control with RNC.

Fig 1: Conceptual illustration of our proposed approach. (a) RNE leverages the fact that RND between time-reversal processes is 1 to calculate marginal densities. (b) RNC applies RNE to calculate importance weights for inference time control.

Before considering diffusion models in the form of Eqs. (1) and (2), it is helpful to consider their discrete counterparts. The transition kernels in discrete time are defined by the conditional densities $p_{n|n+1}$ and $p_{n+1|n}$. Therefore, the question we are seeking to answer is: *can we connect conditional densities with marginal densities?* In fact, this is precisely what Bayes' rule states:

$$p_{n|n+1}(X_n|X_{n+1})p(X_{n+1}) = p_{n+1|n}(X_{n+1}|X_n)p(X_n), \quad \forall (X_n, X_{n+1}) \in \mathcal{X} \times \mathcal{X}. \quad (5)$$

Under a Bayesian interpretation, the forward and backward transition kernels play the roles of the likelihood and the Bayesian posterior. Ordinarily, one's goal is to infer the posterior, but here—assume both kernels are available—we can directly form their ratio to compute the ratio of marginals.

A similar conclusion also exists in continuous time through the concept of time-reversal. Specifically, considering the SDE evolving from $p_\tau$ to $p_{\tau'}$:

$$dX_t = \mu_t(X_t)dt + \epsilon_t \overrightarrow{dW_t}, \quad X_\tau \sim p_\tau, \quad (6)$$

given regularity on $\mu_t$, one can define its time-reversal (Anderson, 1982; Nelson, 1967):

$$dX_t = \nu_t(X_t)dt + \epsilon_t \overleftarrow{dW_t}, \quad X_{\tau'} \sim p_{\tau'}, \quad (7)$$

where $\nu_t = \mu_t - \epsilon_t^2 \nabla \log p_t$, and $p_t$ is the law for $X_t$. The forward and backward diffusion processes in Eqs. (1) and (2) (or equivalently Eqs. (3) and (4)) exemplify this time-reversal pairing.

A key observation is that while the processes in Eqs. (6) and (7) evolve in opposite directions, they induce the same probability measure over the path space. Therefore, their Radon-Nikodym derivative (informally, the "density ratio") is always 1. Let $\overrightarrow{\mathbb{P}}^\mu$ and $\overleftarrow{\mathbb{P}}^\nu$ be the path measures of Eqs. (6) and (7) respectively. We have $d\overrightarrow{\mathbb{P}}^\mu/d\overleftarrow{\mathbb{P}}^\nu(Y_{[\tau,\tau']}) = 1$, where $Y_{[\tau,\tau']}$ is the solution to *any* Itô process within the time-horizon $[\tau, \tau']$, with diffusion coefficient $\epsilon_t$. The expression is merely the definition of time reversal. However, when discretising the SDEs with, e.g., the Euler-Maruyama integrator, we can easily see how this definition connects marginals with transition kernels. Concretely, we first split the time horizon with $N$ time steps $\tau = t_1 < t_2 < \cdots < t_N = \tau'$. Then at each step, with $\Delta t_n = |t_{n+1} - t_n|$, the forward and backward processes are defined as

$$p_{n+1|n}^\mu(X_{t_{n+1}}|X_{t_n}) = \mathcal{N}(X_{t_{n+1}}|X_{t_n} + \mu_{t_n}(X_{t_n})\Delta t_n, \epsilon_{t_n}^2 \Delta t_n I), \quad (8)$$

$$p_{n|n+1}^\nu(X_{t_n}|X_{t_{n+1}}) = \mathcal{N}\left(X_{t_n}|X_{t_{n+1}} - \nu_{t_{n+1}}(X_{t_{n+1}})\Delta t_n, \epsilon_{t_{n+1}}^2 \Delta t_n I\right). \quad (9)$$

After discretising, $d\overrightarrow{\mathbb{P}}^\mu/d\overleftarrow{\mathbb{P}}^\nu(Y_{[\tau,\tau']}) = 1$ becomes $\frac{p_\tau(Y_\tau) \prod_{n=1}^{N-1} p_{n+1|n}^\mu(Y_{t_{n+1}}|Y_{t_n})}{p_{\tau'}(Y_{\tau'}) \prod_{n=1}^{N-1} p_{n|n+1}^\nu(Y_{t_n}|Y_{t_{n+1}})} \approx 1$.

This approximation is exact as $N \to \infty$. Formally, we define the quantity $R$ as follows:

**Definition 2.1.** Consider the forward and backward SDEs in Eqs. (6) and (7). Let $Y$ be the solution to an arbitrary process with the same diffusion coefficient. Following Eqs. (8) and (9)'s discretisation, we define [1]

$$R_\mu^\nu(Y_{[\tau,\tau']}) = \lim_{N \to \infty} \frac{\prod_{n=1}^{N-1} p_{n|n+1}^\nu(Y_{t_n}|Y_{t_{n+1}})}{\prod_{n=1}^{N-1} p_{n+1|n}^\mu(Y_{t_{n+1}}|Y_{t_n})}. \quad (10)$$

With this definition, we obtain the following identity for the $\mu$ and $\nu$ processes satisfying time-reversal.

$$p_\tau(Y_\tau)/p_{\tau'}(Y_{\tau'}) = R_\mu^\nu(Y_{[\tau,\tau']}) \tag{11}$$

We note that the limit in Eq. (10) can be formalised (Berner et al., 2025) thus $R_\mu^\nu$ can be expressed as

$$R_\mu^\nu(Y_{[\tau,\tau']}) = \exp\Big(\int_\tau^{\tau'} \frac{1}{\epsilon_t^2} \nu_t \cdot \overleftarrow{\mathrm{d}Y_t} - \int_\tau^{\tau'} \frac{1}{\epsilon_t^2} \mu_t \cdot \overrightarrow{\mathrm{d}Y_t} + \frac{1}{2}\int_\tau^{\tau'} \frac{1}{\epsilon_t^2}(||\mu_t||^2 - ||\nu_t||^2)\mathrm{d}t\Big), \tag{12}$$

For a more detailed discussion on this expression, we defer to Vargas et al. (2023b, eq. 14-15). For readers not familiar with Itô integrals, we highlight that our approach can be fully understood and implemented as Eq. (10) with finite $N$ and simple Gaussian kernels. That said, the connection to its continuous-time counterpart will enable us to design better estimators.

In summary, as conceptually shown in Fig. 1a, for the denoising process of a pretrained diffusion model, we can always pair it with its time-reversal, up to training error. By exploiting the fact that the Radon-Nikodym derivative between any diffusion process and its time-reversal is identically one, we obtain a simple and intuitive formula that ties together marginal densities and transition kernels. We call this identity the RADON–NIKODYM ESTIMATOR (RNE).

A direct application of this relation is for density estimation: when $\tau' = 1$, $p_{\tau'}$ is tractable, typically as a Gaussian distribution. Interestingly, when writing $R$ in continuous-time as Eq. (12), we will recover the estimator with density-augmented SDE (Karczewski et al., 2024, Theorem 1), and equivalently, in their concurrent work, Itô density estimator (Skreta et al., 2024, Theorem 1). We will discuss this connection and application in more detail in Appendix C.1. In the following sections, we will mainly focus on demonstrating RNE for inference-time control and energy-based training.

## 2.1 RNE for Inference-time Control

RNE provides a flexible and plug-and-play approach for calculating the weight of Sequential Monte Carlo (SMC) for inference-time control. Given a pretrained diffusion model which samples from distribution $p_0$ (or two pretrained diffusion models for $p_0^{(1)}$ and $p_0^{(2)}$), we may want to generate samples from a new target $q_0$ without retraining the model(s). This includes, but not limited to: **(1) annealing:** $q_0 \propto p_0^\beta$; **(2) reward-tilting/posterior sampling:** $q_0 \propto p_0 \exp(r)$ with a reward/likelihood $r$; and **(3) classifier-free guidance ($\alpha = 1 - \beta$) or model product ($\alpha = \beta$):** $q_0 \propto (p_0^{(1)})^\alpha (p_0^{(2)})^\beta$.

One naive approach to generate samples from $q_0$ is importance sampling (IS). Specifically, we can estimate the density $p_0$ of generated samples from the pretrained diffusion model, and then calculate the importance weight as $q_0/p_0$. However, when $p_0$ and $q_0$ differ significantly, the importance weight will have a large variance, rendering this approach infeasible in practice. Therefore, we consider applying importance resampling along the sampling process, essentially forming the Sequential Monte Carlo (SMC) algorithm. In the following, we first describe SMC, and then introduce RNE to calculate the importance weights, which we refer to as the Radon-Nikodym Corrector (RNC).

### 2.1.1 Sequential Monte Carlo

Conceptually, SMC distributes the burden of importance sampling across the entire path, thereby reducing variance at each step. To apply SMC in this setting, we first define a sequence of intermediate target distributions corresponding to each time step. Next, we specify a proposal process from which samples are drawn. Since particles generated by the proposal may not align perfectly with the intermediate targets, we apply importance resampling to progressively realign the particles with the intended sequence of targets. To apply this procedure, we introduce three key components:

1. **a backward sampling ("proposal") process:** $\mathrm{d}X_t = a_t(X_t)\,\mathrm{d}t + \epsilon_t \overleftarrow{\mathrm{d}W_t};$     (13)

2. **a forward auxiliary ("target") process:** $\mathrm{d}Y_t = b_t(Y_t)\mathrm{d}t + \epsilon_t \overrightarrow{\mathrm{d}W_t};$     (14)

3. **intermediate target marginal densities:** $q_t$.

Note that the sampling and target processes are not the same; they are generally not required to be time-reversals. We hence denote the process in Eq. (13) by $X$ and that in Eq. (14) by $Y$. In fact, we have a large flexibility in designing these components. We will discuss the choices of the backward and forward processes in Section 2.1.3. For the intermediate marginal, we can heuristically choose:

$$\textbf{anneal:}q_t \propto p_t^\beta; \; \textbf{reward-tilting:}q_t \propto p_t \exp(r_t); \; \textbf{CFG \& product:}q_t \propto (p_t^{(1)})^\alpha (p_t^{(2)})^\beta \tag{15}$$

---

[1]The limit in Eq. (10) can be understood in an almost sure sense.

where $r_t$ is an intermediate reward which can be heuristically crafted (Wu et al., 2023), or be a reward model trained on the corresponding noisy data (Kong et al., 2025).

We now consider how to apply these components. Assume we have $M$ particles $\{X_{\tau'}^{(m)}\} \sim q_{\tau'}$ at time $\tau'$, now we consider how to obtain particles following $q_\tau$ at time $\tau$ ($\tau < \tau'$). We first evolve the particles along the backward sampling process in Eq. (13) to time step $\tau$, resulting in $M$ trajectories $\{X_{[\tau,\tau']}^{(m)}\}$. However, these trajectories will not follow the marginal density $q_\tau$ at time $\tau$. Therefore, we need to resample to ensure asymptotically unbiased samples from $q_\tau$, as explained in the following.

Let $\overleftarrow{\mathbb{Q}}^a$ be the path measure of Eq. (13) in $t \in [\tau, \tau']$ with initial density at $\tau'$ as $q_{\tau'}$, and let $\overrightarrow{\mathbb{Q}}^b$ be the path measure of Eq. (14) in $t \in [\tau, \tau']$ with initial density at $\tau$ as $q_\tau$. Here we slightly abuse the notion for simplicity: we should understand $\overleftarrow{\mathbb{Q}}^a$ as $\overleftarrow{\mathbb{Q}}_{[\tau,\tau']}^{a,q_{\tau'}}$, the path measure starting from $q_{\tau'}$ at time $\tau'$ and ends at time $\tau$, and also understand $\overrightarrow{\mathbb{Q}}^b$ similarly. The importance weight of $X_{[\tau,\tau']}$ is then

$$w_{[\tau,\tau']}(X_{[\tau,\tau']}) = \mathrm{d}\overrightarrow{\mathbb{Q}}^b/\mathrm{d}\overleftarrow{\mathbb{Q}}^a(X_{[\tau,\tau']}) = q_\tau(X_\tau)/q_{\tau'}(X_{\tau'})\left[R_b^a(X_{[\tau,\tau']})\right]^{-1}, \qquad (16)$$

where $R_b^a$ are defined in Eq. (10). Note that while Eq. (16) calculates the weight over path space, we can verify that this yields a correct importance weight for the marginal $q_\tau$, as shown in Appendix H.1.

We then perform self-normalised importance resampling: first, we normalise the importance weights $\bar{w}^{(m)} \leftarrow \frac{w_{[\tau,\tau']}(X_{[\tau,\tau']}^{(m)})}{\sum_{m'=1}^{M} w_{[\tau,\tau']}(X_{[\tau,\tau']}^{(m')})}$, and sample $M$ indices from the Categorical distribution defined with these weights $\{i_m\} \sim \text{Categorical}\left(\bar{w}^{(1)}, \cdots, \bar{w}^{(M)}\right)$. We return the particles corresponding to the resampled indices. This ensures the samples at time $\tau$ follow the desired target $q_\tau$ as $M \to \infty$. We repeat this pipeline until reaching $q_0$ and this process is conceptually illustrated in Fig. 1b.

### 2.1.2 CALCULATING IMPORTANCE WEIGHTS WITH RNC

Now, we consider how to calculate the importance weight in Eq. (16). The term $R_b^a$ is simply a ratio of products of Gaussian densities when discretised, the only unknown term is the ratio between two marginals $q_\tau(X_\tau^{(m)})/q_{\tau'}(X_{\tau'}^{(m)})$. Fortunately, as we define the intermediate target $q_t$ by modifying the marginal $p_t$ of the pre-trained diffusion model as exemplified in Eq. (15), we can express this unknown ratio using the pre-trained model's marginals. Precisely, plugging in the RNE in Eq. (10): $p_\tau(Y_\tau)/p_{\tau'}(Y_{\tau'}) = R_\mu^\nu(Y_{[\tau,\tau']})$, we obtain the following results:

---

**RN Corrector (RNC).** Consider a pair of time-reversal forward and backward SDEs in Eqs. (6) and (7) with drifts $\mu_t$ and $\nu_t$ (or two pairs: $\mu^{(1)}$ & $\nu^{(1)}$ and $\mu^{(2)}$ & $\nu^{(2)}$). The SMC weight in Eq. (16) is given by

Anneal: 
$$w_{[\tau,\tau']} \propto \left[R_\mu^\nu(X_{[\tau,\tau']})\right]^\beta \left[R_b^a(X_{[\tau,\tau']})\right]^{-1}. \qquad (17)$$

Reward: 
$$w_{[\tau,\tau']} \propto \frac{\exp(r_\tau(X_\tau))}{\exp(r_{\tau'}(X_{\tau'}))} R_\mu^\nu(X_{[\tau,\tau']})\left[R_b^a(X_{[\tau,\tau']})\right]^{-1}. \qquad (18)$$

CFG & Product: 
$$w_{[\tau,\tau']} \propto \left[R_{\mu^{(1)}}^{\nu^{(1)}}(X_{[\tau,\tau']})\right]^\alpha \left[R_{\mu^{(2)}}^{\nu^{(2)}}(X_{[\tau,\tau']})\right]^\beta \left[R_b^a(X_{[\tau,\tau']})\right]^{-1}. \qquad (19)$$

---

In summary, when performing SMC with RNC, we start from a pair of time-reversal forward and backward processes, which provides an estimate for the marginal density ratio. We then choose the sampling process, target process and intermediate marginal defined in Eqs. (13) to (15), and calculate the SMC weight as above. Importantly, all components in the importance weight can be approximated by Gaussian kernels [2]. *Note that, in practice, we only need to calculate $\Delta \log w$ for each denoising step with negligible computation cost.*

### 2.1.3 DESIGN CHOICES OF THE SAMPLING AND TARGET PROCESS

Notably, the RN Corrector works for any choice of drifts $a_t$ and $b_t$ in Eqs. (13) and (14). We now examine two specific scenarios for designing the sampling and target processes:

- In the first scenario, suppose we have access to the perfect diffusion model—that is, we know the exact forms of both $\mu_t$ and $\nu_t$, and can therefore evaluate the forward and backward kernels $p_{n+1|n}^\mu$

---

[2]As a concrete example, let's inspect the anneal IS weights in Eq. (17):

$$w_{[\tau,\tau']} \approx \left(\frac{\prod_{n=1}^{N-1} p_{n|n+1}^\nu(X_{t_n}|X_{t_{n+1}})}{\prod_{n=1}^{N-1} p_{n+1|n}^\mu(X_{t_{n+1}}|X_{t_n})}\right)^\beta \frac{\prod_{n=1}^{N-1} p_{n+1|n}^b(X_{t_{n+1}}|X_{t_n})}{\prod_{n=1}^{N-1} p_{n|n+1}^a(X_{t_n}|X_{t_{n+1}})}, \qquad (20)$$

and $p_{n|n+1}^{\nu}$ as $N \to \infty$. In this setting, we are free to choose any sampling and target processes, and this formulation supports flexible applications including annealing, reward-tilting or product.

- if the diffusion model is imperfectly trained, we only have access to the denoising drift $\nu_t$ parameterised by the imperfect score network[3]. Hence, we no longer get access to its time-reversal term with drift $\mu_t$. In this case, we can set the target process's drift $b_t$ to cancel this unknown term. However, this formulation is limited to reward-tilting as we will explain later.

☛ **Perfect diffusion model: flexible design choices**    Assume a perfect diffusion model where the noising and denoising process with forward and backward drift $\mu_t$ and $\nu_t$ are time-reversals. In this case, we enjoy great flexibility in choosing the sampling and target process. The only approximation error then arises from the time discretisation, which disappears as the discretisation steps $N \to \infty$.

In Appendix C.3, we list some heuristic choices of the sampling and target processes for anneal, reward-tilting and produce cases. *Note that they are not exhaustive—any suitable heuristics can be used without altering the core SMC algorithm implementation.* We present an example pseudocode in Appendix A to highlight this Macro-like property of RNC.

Interestingly, the expressions in Eqs. (17) to (19) recover FKC (Skreta et al., 2025) as special cases for certain choices of $a$ and $b$. We discuss this connection in Appendix C.4. FKC derives its weights via the Feynman-Kac PDE and then designs the sampling process to cancel the costly divergence term. Therefore, FKC has restrictive design choices. By contrast, our RNC features higher flexibility in selecting these processes, yet still incurs no extra computational overhead, allowing us to heuristically select a process pair that may reduce variance (Jarzynski, 1997; Neal, 2001). Moreover, FKC requires deriving the weight formula for each task (anneal, product, etc), while RNC provides a macro-style "*plug-and-play*" recipe for computing importance weights.

☛ **Imperfect diffusion model: choice for cancellation (reward-tilting)**    So far, we assumed a perfect diffusion model, which gives us access to an SDE and its reversal. However, in practice, we typically encounter model imperfection and time-discretisation errors when calculating the marginal density ratio Eq. (11). Consequently, the resampled $X_\tau$ will not follow $q_\tau$ exactly, even as the number of samples $M \to \infty$. Fortunately, in the reward-tilting case, we can still obtain exact importance weights, despite the discretisation and score estimation errors:

**Proposition 2.2** (Exact SMC weight for reward-tilting with imperfect diffusion model)**.**
$$w_{[\tau,\tau']} \propto \frac{\exp(r_\tau(X_{t_1=\tau}))}{\exp(r_{\tau'}(X_{t_N=\tau'}))} \frac{\prod_{n=1}^{N-1} p_{n|n+1}^{\nu}(X_{t_n}|X_{t_{n+1}})}{\prod_{n=1}^{N-1} p_{n|n+1}^{a}(X_{t_n}|X_{t_{n+1}})} \tag{21}$$

We provide a detailed derivation and explain why it is only applicable to reward-tilting in Appendix C.5. This formulation recovers the Twisted Diffusion Sampler (TDS, Wu et al., 2023, eq.11), and follow-up works such as Dou & Song (2024) and Feynman-Kac Steering (Singhal et al., 2025).

In summary, RNC allows us to compute the importance-sampling weights for SMC without requiring explicit knowledge of the marginal density, thereby providing great flexibility in designing inference-time control while maintaining a plug-and-play algorithm.

## 2.2 RNE FOR REGULARISING ENERGY-BASED DIFFUSION MODEL

Another application of RNE is to improve the training of energy-based diffusion models. These models have a variety of applications in machine-learning force fields (Arts et al., 2023), free-energy estimation (Máté et al., 2024), neural sampler (Phillips et al., 2024; Zhang et al., 2025) and model composition (Du et al., 2023), among others. Concretely, we aim to train a diffusion model whose network outputs a scalar energy. However, the denoising score matching objective (Vincent, 2011) suffers from a "blindness" issue (Zhang et al., 2022), leading to inaccurate energy estimates.

RNE offers a natural way to enhance the accuracy of the energy-based diffusion model. Specifically, in addition to the standard DSM, we introduce the following regularisation to enforce Eq. (11).

$$\mathcal{R} = \mathbb{E}_{\mathrm{sg}(X_{[\tau,\tau']})} \| \mathrm{sg}(\log R_\mu^\nu(X_{[\tau,\tau']})) + \log p_{\tau'}(X_{\tau'}) - \log p_\tau(X_\tau) \|^2 \tag{22}$$

---

[3]The normal perspective is that we define the noising process but do not know the perfect reversal process. However, we can also interpret this case as we know the denoising process defined via the learned network, while not having access to its exact time-reversal along the noising direction.

where $\log p_\tau$ and $\log p_{\tau'}(X_\tau)$ are given by the energy-parametrised diffusion model and $[\tau, \tau']$ is a randomly selected time horizon. `sg` represents stop-gradient. In practice, we can select a small time increment $\Delta t$, and apply this regularisation between randomly selected adjacent time steps $t$ and $t + \Delta t$. We then calculate $R_\mu^\nu$ using a single forward and a single backward kernel. As discussed in Appendix E and proved in Appendix H.5, this regularisation is equivalent to the regularisation derived from the Fokker–Planck equation in continuous time (Plainer et al., 2025). However, our approach does not require computing or estimating the divergence, providing a more efficient alternative.

## 2.3 RNE FOR CTMC

RNE conceptually only requires a pair of dynamics that are time reversals of each other; hence it can be applied to other modalities, such as continuous-time Markov chains (CTMCs). All the results discussed above remain valid; the only difference is that $R$ is now defined in terms of the rate matrices. We provide further details on CTMC-RNE in Appendix D.

## 3 STABILISING RNE WITH REFERENCE AND CONVERGENCE ANALYSIS

So far, we define $R$ in Eq. (10), and apply this concept for inference time control and energy-based training. However, in practice, calculating $R$ by directly discretising the forward and backwards SDE as in Eq. (10) can lead to instability and larger accumulated error. We provide an intuition behind this instability in Appendix G.1. At a high level, this issue arises because, at each discretisation step, the variances of the forward and backward kernels are misaligned.

To address this issue, we introduce an analytical reference (Vargas et al., 2023a): consider an SDE with linear drift $\phi$ whose initial state $\pi_0$ is Gaussian, In this case, the marginal density $\pi_t$ remains Gaussian at all times, and one can derive the exact time-reversal drift $\psi_t = \phi_t - \epsilon_t^2 \nabla \log \pi_t$. Let $\overrightarrow{\Pi}^\phi$ and $\overleftarrow{\Pi}^\psi$ be the path measures of this analytical pair. We can rewrite Eq. (10) as follows:

$$R_\mu^\nu(Y_{[\tau,\tau']}) = \frac{p_\tau(Y_\tau)}{p_{\tau'}(Y_{\tau'})} \frac{\mathrm{d}\overleftarrow{\mathbb{P}}^\nu}{\mathrm{d}\overrightarrow{\mathbb{P}}^\mu}(Y_{[\tau,\tau']}) = \frac{p_\tau(Y_\tau)}{p_{\tau'}(Y_{\tau'})} \frac{\mathrm{d}\overleftarrow{\mathbb{P}}^\nu}{\mathrm{d}\overleftarrow{\Pi}^\psi}(Y_{[\tau,\tau']}) \frac{\mathrm{d}\overrightarrow{\Pi}^\phi}{\mathrm{d}\overrightarrow{\mathbb{P}}^\mu}(Y_{[\tau,\tau']}), \tag{23}$$

$$\approx \frac{\pi_\tau(Y_\tau)}{\pi_{\tau'}(Y_{\tau'})} \frac{\prod_{n=1}^{N-1} p_{n|n+1}^\nu(Y_{t_n}|Y_{t_{n+1}})}{\prod_{n=1}^{N-1} p_{n|n+1}^\psi(Y_{t_n}|Y_{t_{n+1}})} \frac{\prod_{n=1}^{N-1} p_{n+1|n}^\phi(Y_{t_{n+1}}|Y_{t_n})}{\prod_{n=1}^{N-1} p_{n+1|n}^\mu(Y_{t_{n+1}}|Y_{t_n})}. \tag{24}$$

By introducing the reference process $\Pi$, we obtain the Radon–Nikodym derivative path measures along the same direction, ensuring the variance of transition kernels is aligned after discretisation. In this work, we choose $\pi_0$ to be Gaussian for simplicity. However, it is not the only option—we can also use a Gaussian mixture adaptive to data. Using a reference similar to the data distribution may offer more accurate results, as observed by Noble et al. (2024) in the context of neural samplers. Additionally, we highlight that the reference process only involves calculating Gaussian kernels without any extra network evaluation, and hence has almost no computational overhead in practice.

We also note that direct Euler-Maruyama discretisation (as described in Eq. (57) in the appendix) of the continuous RNE in Eq. (12) does not have such instabilities, providing a competitive practical alternative. We include a detailed discussion in Appendix G.2. However, using our proposed reference still offers more accurate results, which we empirically verify in Fig. 18 in Appendix G.2.

We now present our reference-based RNE's convergence rate. Let's consider the case where we use the diffusion model defined in Eqs. (1) and (2), where $\mu_t = f_t$ is a linear function, and $\nu_t = \mu_t - \sigma_t^2 \nabla \log p_t$ is the backward drift. We choose a reference process whose forward drift $\phi_t = \mu_t$, and the initial marginal to be Gaussian. Let's denote the drift of this time-reversal as $\psi_t = \mu_t - \sigma_t^2 \nabla \log \pi_t$. In this case, we have the following non-asymptotic guarantee:

**Proposition 3.1.** *Consider the case where $\mu_t$ is a linear forward drift, as in diffusion models. We choose an analytic reference by setting its drift as $\mu_t$ and $\pi_0$ to be Gaussian, and denote its time-reversal drift as $\psi_t$. Assuming $Y$ has bounded $L^p$ moments, it follows that:*

$$|| \log R_\mu^\nu(Y_{[\tau,\tau']}) - \log R^N(\hat{Y}_{[\tau,\tau']})||_{L^2} \leq \mathcal{O}(\sqrt{\Delta t}) \tag{25}$$

*$||.||_{L^2}$ denotes the $L^2$ norm, $\hat{Y}$ is the Euler-Maruyama discretisation of $Y$, and $\log R^N(\hat{Y}_{[\tau,\tau']})$ is our discretised RNE estimator $R^N(\hat{Y}_{[\tau,\tau']}) = \frac{\pi_\tau(\hat{Y}_\tau)}{\pi_{\tau'}(\hat{Y}_{\tau'})} \frac{\prod_{n=1}^{N-1} p_{n|n+1}^\nu(\hat{Y}_{t_n}|\hat{Y}_{t_{n+1}})}{\prod_{n=1}^{N-1} p_{n|n+1}^\psi(\hat{Y}_{t_n}|\hat{Y}_{t_{n+1}})} \frac{\prod_{n=1}^{N-1} p_{n+1|n}^\phi(\hat{Y}_{t_{n+1}}|\hat{Y}_{t_n})}{\prod_{n=1}^{N-1} p_{n+1|n}^\mu(\hat{Y}_{t_{n+1}}|\hat{Y}_{t_n})}.$*

Tab 1: Inference-time annealing on ALDP. *SMC will reduce sample diversity, which predominantly influences $W_2$. Therefore, $W_2$ for "anneal score" should not be directly compared against SMC methods. Instead, energy and distance TVD are less sensitive to sample diversity and are more comparable.

| Metric | Energy TV($\downarrow$) | Distance TV($\downarrow$) | Sample $W_2(\downarrow)$ |
|---|---|---|---|
| Anneal score (wo SMC) | 0.794 | 0.023 | **0.173*** |
| FKC | 0.338 | 0.022 | 0.289 |
| RNC ($c_a = 1, c_b = 0$) | 0.386 | 0.017 | 0.282 |
| RNC ($c_a = 0.6, c_b = 0.4$) | **0.034** | **0.011** | 0.253 |

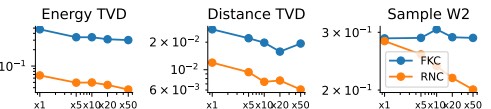
Energy TVD    Distance TVD    Sample W2

Fig 3: Inference-time scaling on ALDP.

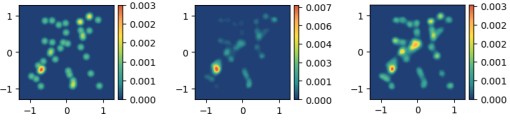
(a) Ground truth    (b) DSM    (c) RNE reg.

Fig 4: Learned density on 2D GMM.

Fig 2: Energy TVD (left), sample $W_2$ (middle), and accumulated weight variance (right) by different pairs of $(c_a, c_b)$ for annealing on ALDP.

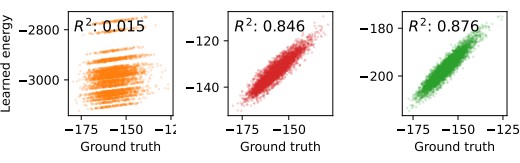
(a) DSM    (b) Dual SM    (c) RNE reg.

Fig 5: Learned energy vs. GT on 100D GMM.

Tab 2: Quality of samples obtained by running denoising process (denoted as DM) and running MCMC on learned energy at $t = 0$.

| Training method | Sample Method | Sample $W2$ |
|---|---|---|
| DSM | DM | 0.1811 |
| | MCMC | 0.9472 |
| RNE Reg | DM | 0.1809 |
| | MCMC | 0.1836 |

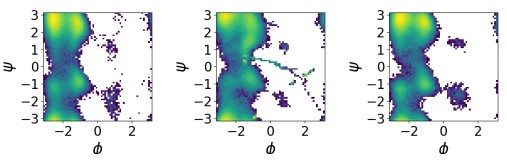
(a) Ground truth    (b) DSM    (c) RNE reg.

Fig 6: Samples by MCMC on learned energy.

We further derive an error bound on the importance weights, considering both the discretisation error and the error in the score network:

**Proposition 3.2.** *(Discrete time and approximate score bound) Following (Lee et al., 2023; Chen et al., 2022), we assume* $||\nabla \log p_\tau(\hat{Y}_\tau) - s_\tau^\theta(\hat{Y}_\tau)||_{L^2} \leq \epsilon_{\text{score}}$, *then*

$$|| \log w^{\text{exact}}(\hat{Y}_{[\tau,\tau']}) - \log w_\theta^{\text{RNC}}(\hat{Y}_{[\tau,\tau']})||_{L^2} \leq E\epsilon_{\text{score}} + P'\sqrt{\Delta t} \tag{26}$$

*where* $E, P'$ *are positive constants,* $w_\theta^{\text{RNC}}(\hat{Y}_{[\tau,\tau']})$ *is the discrete-time weight estimated by* $R^N(\hat{Y}_{[\tau,\tau']})$ *with the learned score* $s_\tau^\theta$, $w^{\text{exact}}(\hat{Y}_{[\tau,\tau']})$ *is the exact SMC weight.*

We prove these results with a more detailed discussion in Appendix G.3.

## 4 EXPERIMENTS

In this section, we conduct comprehensive experiments to evaluate our approach for both inference-time control and energy-based training. Please refer to Appendix for more details on the experiments.

**Inference-time annealing**   We evaluate our proposed method for inference-time annealing on a small molecule, alanine dipeptide (ALDP). We train the model on $T_{\text{high}} = 800K$ and anneal it to $T_{\text{low}} = 300K$. Since SMC typically suffers from low diversity, we use a batch size of 500 and collect 50 batches to calculate the metrics. We compare RNC against FKC (Skreta et al., 2025) and, for reference, a baseline that merely rescales the score without SMC correction. For RNC, we apply Eq. (17) to calculate the importance weights, and select the sampling and target process heuristically as Eq. (34), where we control the drift $a_t$ and $b_t$ via a hyperparameter $\lambda_t^a$ and $\lambda_t^b$. We evaluate two different choices of sampling and target processes: (**1**) $\lambda_t^a = -\epsilon_t^2 T_{\text{high}}/T_{\text{low}}, \lambda_t^b = 0$. By Proposition C.3, this is theoretically identical to FKC. (**2**) we further introduce free parameters $c_a$ and $c_b$: $\lambda_t^a = -\epsilon_t^2 T_{\text{high}}/T_{\text{low}} \cdot c_a, \lambda_t^b = \epsilon_t^2 T_{\text{high}}/T_{\text{low}} \cdot c_b$.

We report in Tab. 1 the Wasserstein-2 ($W_2$) distance between the ground-truth and generated samples, alongside the total variation distance (TVD) computed on both energy and interatomic-distance histograms. Figure 2 shows a sweep over the coefficients $(c_a, c_b)$. When $c_a = 1, c_b = 0$, RNC achieves a similar performance to FKC, which echoes Proposition C.3. More importantly, by selecting different $(c_a, c_b)$, RNC attains higher flexibility and enhanced performance compared to FKC.

We also compute the variance of the importance weights accumulated over the entire sampling trajectory for various $(c_a, c_b)$. As shown in Figure 2 (right), choices with $c_a + c_b$ within $1 \pm 0.2$ tend to minimise the variance, which also correlates with high sample quality. However, while the pattern for variance persists across different targets, the optimal balance of $c_a$ and $c_b$ for achieving the highest sample quality can be task-dependent. For example, for unimodal targets with a sharp peak, a lower ESS reduces diversity but can actually be advantageous in ensuring the sample is closer to the peak. Conversely, for multimodal targets, maintaining a higher ESS preserves greater diversity and helps prevent mode collapse. To illustrate this trade-off, we include inference-time annealing results and additional analysis for the Lennard-Jones (LJ) system and Mixture-of-Gaussian in Appendix F.3.

**Inference-time product: multi-target structure-based small-molecule ligand design** Following Skreta et al. (2025), we evaluate RNC for model product with multi-target structure-based small-molecule ligand design. For a detailed introduction to the background of this task, please refer to Appendix I.2. In summary, we consider sampling from $q_0 \propto (p_0^{(1)} p_0^{(2)})^\beta$, where $p_0^{(1)}$ and $p_0^{(2)}$ represents the diffusion model's outcome conditional on two protein targets. In our experiments, we set $\beta = 2$ following the optimal hyperparameter used by Skreta et al. (2025). We compare our RNC method with FKC (Skreta et al., 2025) and baseline, "Sum score", which samples from the denoising process by directly summing the scores conditioned on each target without SMC.

For RNC, similar to the annealing case, we can choose any sampling and target processes. Here, we heuristically select the options in Eq. (36), where we set $\lambda_t^{a,1} = \lambda_t^{a,2} = -\epsilon_t^2 \beta \cdot c_a$ and $\lambda_t^{b,1} = \lambda_t^{b,2} = \epsilon_t^2 \beta \cdot c_b$. In Tab. 4, we present the result obtained with $c_a = 1, c_b = 0$, which is theoretically equivalent to FKC in continuous time. We observe that these results achieve similar performance, up to the inherent stochasticity in the generation process. We also report the performance obtained with $c_a = 1.0, c_b = 0.2$. As shown in Tab. 4, both FKC and RNC variants are significantly better than the heuristic score summation, at the price of lower diversity. Furthermore, RNC offers higher flexibility in choosing the sampling and target process, providing a visible gain over FKC, particularly with more ligands that have better docking scores than both reference ligands.

**Flexible controls: stitching and reward-tilting for maze navigation** One advantage of RNC is that it can be intuitively and seamlessly extended to tasks that require more flexible controls. Here, we consider a maze-navigation task by stitching together diffusion models trained on short trajectories. Formally, letting the short trajectories follow $p_0$, we aim to sample $[X^{(1)}, \ldots, X^{(L)}] \sim q_0 \propto$

Tab 3: Success rate of trajectory stitching by guidance without SMC and with RNC across 5 tasks.

| | task 1 | task 2 | task 3 | task 4 | task 5 |
|---|---|---|---|---|---|
| wo SMC | 0.501 | 0.669 | 0.585 | 0.288 | 0.714 |
| RNC | 1.000 | 1.000 | 1.000 | 1.000 | 1.000 |

$\exp\big(r([X^{(1)}, \ldots, X^{(L)}])\big) \prod_l p_0(X^{(l)})$, where the reward $r$ is defined to impose first trajectory starts from the initial position and the last trajectory ends at the target, and consecutive trajectories are connected. Despite the complexity of this task, RNC can still provide SMC weights for it without changing the general formula. We use the `pointmaze-medium-stitch-v0` dataset from Park et al. (2024), and consider 5 different pairs of initial points and targets. We include more experimental details in Appendix I.3. We visualise the unstitched trajectories and the samples generated by SMC in Fig. 7, using different colours to represent different short trajectories. We also report the success rate in Tab. 3. For comparison, we include results without SMC, where we only use guidance from the gradient of the reward. We observe that RNC increases the success rate to 100%, demonstrating the flexibility and practical applicability of our method.

**Inference-time scaling** RNC allows us to increase the number of particles during inference to obtain better performance. We showcase this property with the annealing experiments on alanine dipeptide (ALDP). Specifically, we follow the setting we used in Tab. 1, with $c_a = 0.6, c_b = 0.4$, and evaluate the performance scaling with different batch sizes: 100, 500, 1000, 2000, 5000. For comparison, we also report the corresponding FKC results. Fig. 3 shows the sample quality scaling with different numbers of particles. RNC not only features better sample quality but also presents better scaling properties, especially for sample diversity, as reflected by the sample $W_2$ distance.

**Training energy-based models** We train energy-based diffusion with RNE regularisation on both the Gaussian mixture and the alanine dipeptide (ALDP). In Fig. 4, we visualise the learned density for the standard denoising score matching (DSM) and for DSM with RNE regularisation on 2D GMM. We obtain the density value by exponentiating and normalising the learned negative energy at $t = 0$. The unregularised DSM fails to capture the target accurately, whereas RNE regularisation enables

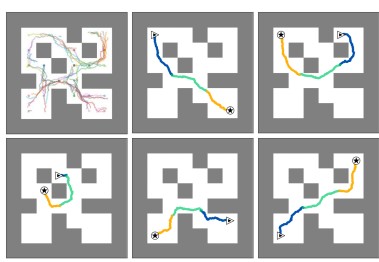

Fig 7: Visualisation of stitched trajectory for maze navigation. We also show examples of unstitched short trajectories in the upper-left corner.

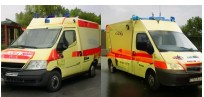 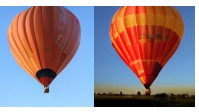 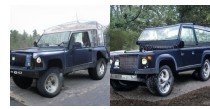

(a) Tilting with RNC. **Prompt from left to right:** (1) *A yellow ambulance*; (2) *An orange balloon with red spots*; (3) *A blue jeep.*

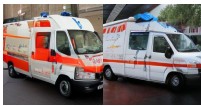 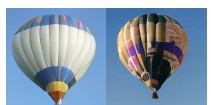 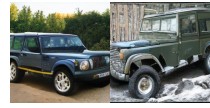

(b) Generation without SMC.

Fig 8: Visualisation of prompt-reward-tilting on *masked discrete diffusion* on ImageNet-256.

Tab 4: Multi-target SBDD performances. Better than known denotes the percentage of generated ligands with lower docking scores than both of two ground truth reference ligands. $P_1$ and $P_2$ are docking scores for two pockets. Div. assesses the pairwise difference over molecular fingerprints. Val. & Uniq. denotes the percentage of ligands that are both valid and unique. Qual. denotes the percentage of ligands that have good physicochemical properties.

| | Better than known. (↑) | $(P_1 * P_2)$ (↑) | $\max(P_1, P_2)$ (↓) | $P_1$ top-1 (↓) | $P_2$ top-1 (↓) | Div. (↑) | Val. & Uniq. (↑) | Qual. (↑) |
|---|---|---|---|---|---|---|---|---|
| Sum score | $0.345_{\pm 0.288}$ | $65.110_{\pm 17.802}$ | $-7.222_{\pm 1.348}$ | $-9.411_{\pm 1.574}$ | $-9.769_{\pm 1.758}$ | $0.881_{\pm 0.010}$ | $0.927_{\pm 0.147}$ | $0.134_{\pm 0.087}$ |
| FKC | $0.608_{\pm 0.390}$ | $\mathbf{82.371_{\pm 24.928}}$ | $\mathbf{-8.296_{\pm 1.450}}$ | $-9.437_{\pm 1.733}$ | $-10.035_{\pm 1.601}$ | $0.814_{\pm 0.043}$ | $0.925_{\pm 0.113}$ | $0.192_{\pm 0.191}$ |
| RNC ($c_a = 1, c_b = 0.0$) | $0.589_{\pm 0.413}$ | $81.186_{\pm 26.158}$ | $-8.122_{\pm 1.588}$ | $\mathbf{-9.650_{\pm 1.608}}$ | $-10.075_{\pm 1.663}$ | $0.823_{\pm 0.027}$ | $0.942_{\pm 0.069}$ | $0.222_{\pm 0.173}$ |
| RNC ($c_a = 1, c_b = 0.2$) | $\mathbf{0.649_{\pm 0.356}}$ | $81.771_{\pm 24.673}$ | $-8.112_{\pm 1.660}$ | $\mathbf{-9.585_{\pm 1.885}}$ | $\mathbf{-10.102_{\pm 1.525}}$ | $\mathbf{0.836_{\pm 0.025}}$ | $\mathbf{0.950_{\pm 0.066}}$ | $\mathbf{0.223_{\pm 0.202}}$ |

the model to recover the energy at $t = 0$ relatively exactly. In Fig. 5, we evaluate our approach on a 100D GMM, and also include dual score matching (Guth et al., 2025) as a baseline. We can see both RNE and dual score matching significantly improve the accuracy of the learned energy.

To assess its potential as a conservative machine-learning force field (MLFF), we train energy-based diffusion models on ALDP samples at 300K, then run MCMC on the learned energy at $t = 0$ and visualise the resulting Ramachandran plot in Fig. 6, where the RNE-regularised model closely reproduces the ground-truth distribution. Importantly, throughout training, we only use samples, without accessing the energies or scores. From Tab. 2, the RNE regularisation does not noticeably influence the quality of the diffusion model itself, showcasing our RNE's flexibility and applicability.

RNE regularisation can also be applied to bridge models such as stochastic interpolants (Albergo et al., 2023). Learning an accurate energy path can improve the accuracy of free-energy estimation via thermodynamic integration (TI, Kirkwood, 1935; Máté et al., 2025). We demonstrate this by estimating the solvation free energy of the alanine dipeptide, using the dataset and systems described in He et al. (2025a). Further details on the background and setup are provided in Appendix I.6. We report the values estimated without and with RNE in Tab. 5, where RNE regularisation substantially improves the accuracy of the results.

Tab 5: ALDP solvation free energy estimated with thermodynamic integration.

| Reference Value | TI wo RNE (Máté et al., 2025) | TI w. RNE (Ours) |
|---|---|---|
| $29.43 \pm 0.01$ | $27.30 \pm 0.45$ | $\mathbf{29.28 \pm 0.04}$ |

**RNE for CTMC** We also verify RNE for CTMC. More precisely, we consider CFG debiasing plus tilting the generation with ImageReward (Xu et al., 2023) with a prompt. We take the MaskGIT (Chang et al., 2022) pretrained on ImageNet-256 by Besnier et al. (2025). MaskGIT defines a (latent) mask image model that predicts the conditional distributions of masked positions given a masked sample. Therefore, similar to Ren et al. (2025), we can turn MaskGIT into a (latent) masked discrete diffusion model by introducing a stochastic masking schedule following Shi et al. (2024). For more details, please refer to Appendix I.4. We visualise the results in Fig. 8. As shown, RNE achieves strong alignment between the generated images and the target prompts, demonstrating both the effectiveness of RNE on CTMC and its scalability to larger image-generation settings.

## 5 CONCLUSION

In this paper, we introduce the RADON–NIKODYM ESTIMATOR (RNE). It leverages the fact that *for any diffusion process we consider, we can pair it with its time-reversal*, and the Radon–Nikodym derivative of the forward path measure with respect to the reverse path measure is always equal to one. This principle lets us decouple marginal densities from transition kernels, yielding a highly flexible and plug-and-play method for density estimation, inference-time control and energy-based training, for diffusion models across modalities. We discuss its limitations in Appendix B.

## LLM USAGE DISCLOSURE

LLM was used at the sentence level to correct grammar.

## ACKNOWLEDGEMENTS

We acknowledge Alexander Denker, Julius Berner, and Kirill Neklyudov for their insightful feedback and discussion on our manuscript. We acknowledge the helpful discussion with Michael Plainer, which drew our attention to the importance of energy-based diffusion models. We acknowledge Tony OuYang for the helpful discussion on energy-based training, which inspired us to analyse the equivalence between RNE-regularisation and Fokker-Planck regularisation. We also acknowledge Zijing Ou for pointing out several typos in our manuscript. JH acknowledges support from the University of Cambridge Harding Distinguished Postgraduate Scholars Programme. JMHL acknowledges support from a Turing AI Fellowship under grant EP/V023756/1.

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

# APPENDIX

## A    IMPLEMENTATION OUTLINES ON INFERENCE-TIME CONTROL

In this section, we illustrate the key methods that need to be implemented when a user wants to implement different heuristic choices for $a_t$ and $b_t$ (the different sampling and target process choices).

We provide the RNC anneal weights as an example and illustrate how the $R$ factor computation in the `rne` method does not need to be modified as we change from task to task.

```python
class GuidedSDE(self):

    # Model forward logprob
    def fwd_mu(self,xt,xtm1,t):
        ...

    # Model back logprob
    def fwd_nu(self,xt,xtm1,t):
        ...

    # Implement target logprob
    def fwd_b(self,xt,xtm1,t):
        ...

    # Implement sample logprob
    def bwd_a(self,xt,xtm1,t):
        ...

def rne(fwd, bwd, t, s, ...):
    # Discretise [t, s] with N steps
    ...
    for n in range(N):
        ...
        fn = fwd(..., tn)
        bn = bwd(..., tn)
        lnR += bn - fn
    return lnR

def rnc_ann(sde, beta, t, s, ...):
    fmu, bnu = sde.fwd_mu, sde.bwd_nu
    fb, ba = sde.fwd_b, sde.bwd_a
    lnRmunu = rne(fmu, bnu, t, s,...)
    lnRab = rne(fb, ba, t, s,...)
    lnw = -lnRab + lnRmunu * beta
    return lnw
```

Fig 9: Psuedocode illustrating development pipeline for RNC (exemplified with annealing). The user only needs to implement the sampling and target kernels; changing these does not change the rest of the downstream code nor require re-derivation, unlike FKC (Skreta et al., 2025). In practice, we also add the analytical reference to the implementation of `rne` as described in Section 3. To do this, we simply add/substract the forward/backward kernel and the corresponding Gaussian marginals to `lnR`, which also does not need to be rewritten from task to task.

## B    LIMITATIONS

Our proposed method still encounters the following limitations: (1) RNC suffers from the common limitations of SMC, including bias from self-normalised importance weight when the sample size is small, and low diversity in produced samples. Also, when the target is significantly far from the pretrained diffusion, SMC will not perform well. (2) One variation of RNC relies on the assumption that our pretrained diffusion model is perfect. While this is also the assumption for other previous approaches (Skreta et al., 2025), it will lead to biased annealing/composition results.

## C    SUPPLEMENTARY METHODS

This section presents supplementary methods and technical details that are not covered in the main manuscript:

- In Appendix C.1, we describe in detail how to apply RNE for density estimation, and outline its connection to previous methods.
- In Appendix C.2, we extend the density estimation framework using importance sampling.

We then turn to the use of RNE for inference-time control. Recall that in Section 2.1.3, we discussed two scenarios: one assumes a perfect diffusion model, while the other relaxes this assumption at the cost of more restricted design choices. We now provide additional details on these cases:

- In Appendix C.3, we assume a perfect model and list several heuristic choices for the sampling and target processes (i.e., $a_t$, $b_t$ in Eqs. (13) and (14)).

- In Appendix C.4, under the perfect-model assumption, we show how RNC recovers FKC as a special case.
- In Appendix C.5, we consider imperfect diffusion models. We explain how the SMC weight in Proposition 2.2 is derived and why it is only applicable in the reward-tilting setting.

## C.1 RNE FOR DIFFUSION DENSITY ESTIMATION

As we discussed in main text, a direct application of Eq. (11) is for density estimation: when $\tau' = 1$, $p_{\tau'}$ is tractable, typically as a Gaussian distribution. This leads to the following conclusion:

> **RN Density Estimator (RNDE).** Consider a pair of forward and backward SDEs in Eqs. (6) and (7) with drift $\mu_t$ and $\nu_t$ which are time-reversal of each other. Let $Y$ be the solution to an arbitrary process with the same diffusion coefficient, going either forward or backward. With $R$ defined in Eq. (10), the SDE's marginal density $p_t$ is given by
>
> $$p_t(Y_t) = p_1(Y_1)R_\mu^\nu(Y_{[t,1]}). \tag{27}$$

Given a perfect diffusion model, the RHS of the estimator is tractable up to discretisation error: $p_1$ is a Gaussian density, and $R_\mu^\nu$ can be calculated by the noising and denoising kernel with Eq. (10).

**Connection to previous works.** The relation in Eq. (27) coincides with the density-augmented SDE (Karczewski et al., 2024, Theorem 1), and equivalently, in their concurrent work, Itô density estimator (Skreta et al., 2024, Theorem 1). More precisely, their density estimator states that, for a *perfectly* pretrained diffusion model defined in Eqs. (1) and (2) (e.g., $\sigma_t = \epsilon_t$, $\mu_t = f_t$ and $\nu_t = f_t - \sigma_t^2 \nabla \log p_t$), letting $Y$ be the solution to *any* backward process with the same diffusion coefficient as the diffusion model, $\log p_t(Y_t)$ follows the following SDE:

$$\mathrm{d} \log p_t(Y_t) = -\Big(\nabla \cdot f_t(Y_t) + \nabla \log p_t(Y_t) \cdot (f_t(Y_t) - \frac{\sigma_t^2}{2}\nabla \log p_t(Y_t))\Big)\mathrm{d}t + \nabla \log p_t(Y_t) \cdot \overleftarrow{\mathrm{d}Y_t}. \tag{28}$$

Despite the theoretical equivalence, Eq. (27) is more flexible and practically applicable. Eq. (28) is only computationally feasible for diffusion models where $f_t$ is linear and its divergence term $\nabla \cdot f_t$ is constant. By contrast, Eq. (27) can be applied efficiently to any bridge model whose marginal on one side is tractable, including stochastic interpolants (Albergo et al., 2023), bridge-matching models (Shi et al., 2023; Peluchetti, 2023), and escorted AIS samplers (Vaikuntanathan & Jarzynski, 2008).

Moreover, Eq. (27) provides an alternative—more flexible—perspective on why the Itô density estimator remains valid even when any backward process generates $Y$: for a perfect diffusion, the Radon-Nikodym derivative between the noising forward and the denoising backward process is always one and Eq. (27) always holds. From this viewpoint, there is even no need for $Y$ to satisfy a backward SDE—it can follow processes in any direction, and the estimator still applies. Therefore, this estimator not only can be applied to estimate the density on samples $Y_t$ obtained from a backward SDE trajectory $Y_{[t,1]}$, but it can also estimate the density on arbitrary values $Y_t = y_t$, such as samples on a hold-out test set, like the cases considered by Kingma et al. (2021). To achieve this, we can simulate an SDE, forward in time, from $y_t$, and apply Eq. (27) on this forward trajectory.

We additionally highlight that Skreta et al. (2024, Appendix D) also wrote down the Gaussian-based discrete-time estimator to derive Eq. (28). However, differently, we advocate directly using the form in Eq. (27)—not only because it's more computationally accessible, but also because the RND perspective behind Eq. (27) naturally facilitates enhancements via reference processes and importance sampling, yielding a more stable and accurate estimator, as we will discuss in Section 3.

## C.2 RNE FOR DIFFUSION DENSITY ESTIMATION WITH IMPORTANCE SAMPLING

In the above sections, we make use of the fact that the Radon–Nikodym derivative between a process and its time-reversal is identically one. This provides us with an intuitive algorithm for density estimation. In this section, we provide an alternative approach for density estimation, without relying on the concept of time-reversal. Instead, it leverages the importance sampling perspective:

**Proposition C.1.** *let $p_t(x_t)$ be the marginal density of $X_t = x_t$ satisfying the backwards SDE:*

$$\mathrm{d}X_t = \nu_t(X_t)\mathrm{d}t + \epsilon_t\overleftarrow{\mathrm{d}W_t}, \quad X_1 \sim p_1. \tag{29}$$

Consider a forward process $Y_t$ with drift $u_t$, and define $R_u^\nu$ as Eq. (10), we have

$$p_t(x_t) = \mathbb{E}\left[p_1(Y_1)R_u^\nu(Y_{[t:1]})\middle|Y_t = x_t\right], \tag{30}$$

where the expectation is taken over the forward process within the time horizon $[t, 1]$ conditional on $Y_t = x_t$. When discretised with $t = t_1 < t_2 < \cdots < t_N = 1$:

$$p_{t_1}(x_{t_1}) \approx \mathbb{E}\left[p_{t_N}(Y_{t_N})\frac{\prod_{n=1}^{N-1} p_{n|n+1}^\nu(Y_{t_n}|Y_{t_{n+1}})}{\prod_{n=1}^{N} p_{n+1|n}^u(Y_{t_{n+1}}|Y_{t_n})}\middle|Y_{t_1} = x_{t_1}\right]. \tag{31}$$

A detailed proof can be found in Appendix H.2. However, to provide more intuition, we showcase its derivation from the standard variational inference perspective (Blei et al., 2017; Kingma et al., 2013):

**Variational Inference Macros**    **Pathwise Counterparts**

marginalise:  $p(x) = \int p(x, z)dx,$    $p_{t_1}(x) = \int p(Y_{t_1} = x, Y_{t_2:N})dY_{2:N},$

condition:  $p(x) = \int p(x|z)p(z)dx,$    $p_{t_1}(x) = \int p(Y_{t_1} = x, Y_{t_2:N}|Y_{t_N})p_{t_N}(Y_{t_N})dY_{2:N},$

re-weight:  $p(x) = \mathbb{E}_{q(z|x)}\left[\boxed{\dfrac{p(x|z)}{q(z|x)}}\,p(z)\right].$    $p_{t_1}(x) = \mathbb{E}_{q(Y_{t_2:N}|Y_{t_1}=x)}\left[p_{t_N}(Y_{t_N})\boxed{\dfrac{p(Y_{t_1} = x, Y_{t_2:N}|Y_{t_N})}{q(Y_{t_2:N}|Y_{t_1} = x)}}\middle|Y_{t_1}=x\right].$

RNE!    RNE!

We also note that Proposition C.1 generalises the RN Density estimator in Appendix C.1. As in the setting of a perfect time-reversal, a single Monte Carlo sample suffices to recover $p_t$. Hence, Eq. (30) degrades to Eq. (27). Proposition C.1 also offers an intuitive derivation of the Feynman-Kac density relation proposed by Huang et al. (2021), as stated in the following Corollary:

**Corollary C.2.** *(Huang et al., 2021) The relation in Eq. (30) can be simplified to*

$$p_t(x_t) = \mathbb{E}_{Z_{[t,1]}\sim\overrightarrow{\mathbb{P}}^\nu}\left[p_1(Z_1)\exp\left(\int_t^1 \nabla \cdot \nu_{t'}(Z_{t'})\mathrm{d}t'\right)\middle|Z_t = x_t\right], \tag{32}$$

where $\overrightarrow{\mathbb{P}}^\nu$ represents a forward process with drift $\nu_t$[4].

## C.3  HEURISTIC CHOICE OF $a_t$ AND $b_t$ FOR INFERENCE-TIME CONTROL

After discussing RNE for density estimation, we now return to inference-time control. In Section 2.1.3, we highlighted that we have the freedom to choose any of the sampling and target processes. In this section, we list some heuristics that can be considered. We note that these are by no means exhaustive: any suitable heuristics can be used without altering the core SMC algorithm.

Consider the diffusion model defined in Eqs. (1) and (2) or its SI characterisation in Eqs. (3) and (4). To align our notation with the standard diffusion model literature, recall that

$$\begin{aligned}
\mu_t &= \mathrm{v}_t + \epsilon_t^2/2\nabla \log p_t = f_t + (\epsilon_t^2 - \sigma_t^2)/2\nabla \log p_t \\
\nu_t &= \mathrm{v}_t - \epsilon_t^2/2\nabla \log p_t = f_t - (\epsilon_t^2 + \sigma_t^2)/2\nabla \log p_t
\end{aligned} \tag{33}$$

Then, we may choose

Anneal:  $\quad a_t = f_t + \lambda_t^a \nabla \log p_t, \quad b_t = f_t + \lambda_t^b \nabla \log p_t \tag{34}$

Reward:  $\quad a_t = f_t + \dfrac{\epsilon_t^2 - \sigma_t^2}{2}\nabla \log p_t + \lambda_t^a g_t, \quad b_t = f_t - \dfrac{\epsilon_t^2 + \sigma_t^2}{2}\nabla \log p_t + \lambda_t^b g_t \tag{35}$

CFG & Product:  $\quad a_t = f_t + \lambda_t^{a,1} \nabla \log p_t^{(1)} + \lambda_t^{a,2} \nabla \log p_t^{(2)},$

$\qquad\qquad\qquad\quad b_t = f_t + \lambda_t^{b,1} \nabla \log p_t^{(1)} + \lambda_t^{b,2} \nabla \log p_t^{(2)} \tag{36}$

where the hyperparameter $\lambda$ can be heuristically selected or tuned, and $g_t$ can be designed/learned to approximate the $h$-transform (Uehara et al., 2025; Domingo-Enrich et al., 2024; Denker et al., 2024) or set heuristically (Chung et al., 2023; Wu et al., 2023; Song et al., 2023b; Singhal et al., 2025).

---

[4]We emphasise the different between $\overrightarrow{\mathbb{P}}^\nu$ and the time reversal of Eq. (29). The former directly runs in forward with drift $\nu_t$, inducing a new path measure, while the latter defines the same path measure as Eq. (29).

### C.4 "FKC ⊆ RNC"

We now show the choices which recover FKC (Skreta et al., 2025) as special cases:

---

**Proposition C.3** ("FKC ⊆ RNC"). *RNC with the following $a_t$, $b_t$ and $\epsilon_t$ is equivalent to FKC:*

*Anneal:* $\quad a_t = f_t - \eta\sigma_t^2 \nabla \log p_t, \quad b_t = f_t - (\eta\sigma_t^2 - \beta\epsilon_t^2)\nabla \log p_t, \quad \epsilon_t = \zeta\sigma_t,$

*Product:* $\quad a_t = f_t - \eta\sigma_t^2 \left(\nabla \log p_t^{(1)} + \nabla \log p_t^{(2)}\right),$

$$b_t = f_t - \left(\eta\sigma_t^2 - \epsilon_t^2\beta\right)\left(\nabla \log p_t^{(1)} + \nabla \log p_t^{(2)}\right), \quad \epsilon_t = \zeta\sigma_t, \qquad (37)$$

*CFG:* $\quad a_t = f_t - \sigma_t^2 \left((1-\beta)\nabla \log p_t^{(1)} + \beta\nabla \log p_t^{(2)}\right), \quad b_t = f_t, \quad \epsilon_t = \sigma_t,$

*where $\eta = \beta + (1-\beta)c$ and $\zeta = \sqrt{1 + (1-\beta)2c/\beta}$ for $c \in [0, 1/2]$, following the definition in FKC (Skreta et al., 2025, Propositions 3.1, 3.2, 3.3).*

---

FKC derives its weights via the Feynman-Kac PDE and then designs the sampling process to cancel the costly divergence term. Therefore, FKC has very restrictive design choices. By contrast, our RNC features higher flexibility in selecting these processes, yet still incurs no extra computational overhead, allowing us to heuristically select a process pair that may reduce variance (Jarzynski, 1997; Neal, 2001). Moreover, FKC requires deriving the weight formula for each task (anneal, product, etc), while RNC provides a macro-style "*plug-and-play*" recipe for computing importance weights.

### C.5 EXACT SMC WEIGHT FOR IMPERFECT DIFFUSION MODEL

We now consider the SMC weight for the imperfect diffusion model we discussed in Proposition 2.2.

Before discussing the results, we distinguish two sources of error: (1) the pretrained diffusion model will not perfectly reproduce the training data distribution due to imperfect score and discretisation errors; (2) for RNC, when calculating the SMC weight using the relation in Eq. (11), this equation does not exactly hold due to imperfect time-reversal and discretisation. The first error is intrinsic to the diffusion model and beyond our control; we aim to address the latter here. More concretely, we define $p_t$ as the Law of samples under the *imperfect* diffusion model with *discretisation error*, and our aim is to generate samples from $q_t$ defined in terms of this $p_t$, following Eq. (15).

We still consider the time horizon $[\tau, \tau']$ as an example. To account for the error arising from both model imperfection and discretisation, we will conduct our discussion in discrete time with $N$ steps $\tau = t_1 < t_2 < \cdots < t_N = \tau'$. Let $p_{n|n+1}^\nu(X_{t_n}|X_{t_{n+1}})$ and $p_{n|n+1}^a(X_{t_n}|X_{t_{n+1}})$ be the denoising kernel for the imperfect diffusion model and our chosen sampling kernel, respectively. The SMC weight in Proposition 2.2 is exact. We repeat the result here for easier reference:

$$w_{[\tau,\tau']} \propto \frac{\exp(r_\tau(X_{t_1=\tau}))}{\exp(r_{\tau'}(X_{t_N=\tau'}))} \frac{\prod_{n=1}^{N-1} p_{n|n+1}^\nu(X_{t_n}|X_{t_{n+1}})}{\prod_{n=1}^{N-1} p_{n|n+1}^a(X_{t_n}|X_{t_{n+1}})} \qquad (38)$$

We now answer the following questions: *Why is this SMC weight exact, and why does this only apply to reward-tilting?*

First, analogous to the concept of time-reversal, we denote the posterior density for the diffusion denoising kernel as $p_{2:N|1}(X_{t_{2:N}}|X_1)$. According to Bayes's rule, we have

$$p_{t_1}(X_{t_1})p_{2:N|1}(X_{t_{2:N}}|X_{t_1}) = p_{t_N}(X_{t_N})\prod_{n=1}^{N-1} p_{n|n+1}^\nu(X_{t_n}|X_{t_{n+1}}) \qquad (39)$$

Note that we do not know the tractable form of $p_{2:N|1}$. However, as we will immediately see, we can cancel this term with our chosen target process and hence eliminate the need to calculate it. Concretely, similar to the target process in Eq. (14), here we can also choose an arbitrary target conditional density $q^{\text{target}}(X_{t_{2:N}}|X_{t_1})$, and the SMC weight is defined as

$$w_{[\tau,\tau']} \propto \frac{\exp(r_\tau(X_{t_1=\tau}))}{\exp(r_{\tau'}(X_{t_N=\tau'}))} \frac{p_{t_1}(X_{t_1})}{p_{t_N}(X_{t_N})} \frac{q^{\text{target}}(X_{t_{2:N}}|X_{t_1})}{\prod_{n=1}^{N-1} p_{n|n+1}^a(X_{t_n}|X_{t_{n+1}})} \qquad (40)$$

By Eqs. (39) and (40), we have

$$w_{[\tau,\tau']} \propto \frac{\exp(r_\tau(X_{t_1=\tau}))}{\exp(r_{\tau'}(X_{t_N=\tau'}))} \frac{\prod_{n=1}^{N-1} p_{n|n+1}^\nu(X_{t_n}|X_{t_{n+1}})}{p_{2:N|1}(X_{t_{2:N}}|X_{t_1})} \frac{q^{\text{target}}(X_{t_{2:N}}|X_{t_1})}{\prod_{n=1}^{N-1} p_{n|n+1}^a(X_{t_n}|X_{t_{n+1}})}, \quad (41)$$

The term $p_{2:N|1}(X_{t_{2:N}}|X_1)$ is intractable, and $q^{\text{target}}$ we can freely choose without affecting the correctness of SMC. Therefore, if we set $q^{\text{target}} = p_{2:N|1}$, these two terms will cancel and all terms left in the importance weight will be tractable. In continuous time, $q^{\text{target}} = p_{2:N|1}$ will converge to the time-reversed denoising SDE of the imperfect diffusion model.

It is important to note that the same cancellation cannot be applied to the annealing or the product case. In fact, the derivation is correct until the step of Eq. (41). Taking the annealing case as an example, this step gives us the following SMC weight:

$$w_{[\tau,\tau']} \propto \frac{(\prod_{n=1}^{N-1} p_{n|n+1}^\nu(X_{t_n}|X_{t_{n+1}}))^\beta}{(p_{2:N|1}(X_{t_{2:N}}|X_1))^\beta} \frac{q^{\text{target}}(X_{t_{2:N}}|X_{t_1})}{\prod_{n=1}^{N-1} p_{n|n+1}^a(X_{t_n}|X_{t_{n+1}})}, \quad (42)$$

However, we then cannot set $q^{\text{target}}(X_{t_{2:N}}|X_{t_1}) = p_{2:N|1}^\beta(X_{t_{2:N}}|X_{t_1})$ to cancel the intractable term. This is because $p_{2:N|1}^\beta$ is *normalised*. We may set $q^{\text{target}}(X_{t_{2:N}}|X_{t_1}) = p_{2:N|1}^\beta(X_{t_{2:N}}|X_{t_1})/Z$. But $Z = \int p_{2:N|1}^\beta(X_{t_{2:N}}|X_{t_1})\mathrm{d}X_{t_{2:N}}$, which is NOT a constant but a function of $X_{t_1}$.

## D    RNE FOR DISCRETE DIFFUSION

As we discussed in Section 2.3, RNE can be seamlessly adapted to discrete diffusion with Continuous Time Markov Chains (CTMC) (Campbell et al., 2022; Lou et al., 2023; Shi et al., 2024). To do so we first define the $R$ quantity for CTMC.

**Definition D.1.** (CTMC $R$) Given a CTMC $Y$ in the time horizon $[\tau, \tau']$ and rate matrices $Q_t, Q_t'$ corresponding to CTMC's evolving in different time directions in the time interval $[\tau, \tau']$, we define

$$R_Q^{Q'}(Y_{[\tau,\tau']}) = \exp\left(\int_\tau^{\tau'} Q_s'(Y_s, Y_s) - Q_s(Y_s, Y_s)\,\mathrm{d}s + \sum_{s, Y_s^- \neq Y_s} \log\left(\frac{Q_s'(Y_s^-, Y_s)}{Q_s(Y_s, Y_s^-)}\right)\right), \quad (43)$$

where $\sum_{s, Y_s^- \neq Y_s}$ sums over all points where $Y_s$ switches ("*jumps*") between states.

It is easy to construct the marginal density estimator:

> **Proposition D.2.** *RN Density Estimator (RNDE). Consider a pair of forward and backward CTMCs which are time-reversal of each other and with rate matrices $Q_t, Q_t'$.*
>
> *Let $Y$ be the solution to an arbitrary CTMC, going either forward or backward. Then the marginal density $p_t$ of the CTMC with rate matrix $Q$ is given by*
>
> $$p_t(Y_t) = p_1(Y_1)\exp\left(\int_t^1 Q_s'(Y_s, Y_s) - Q_s(Y_s, Y_s)\,\mathrm{d}s + \sum_{s, Y_s^- \neq Y_s} \log\left(\frac{Q_s'(Y_s^-, Y_s)}{Q_s(Y_s, Y_s^-)}\right)\right). \quad (44)$$

*Proof.* From (Holderrieth et al., 2025, Proposition 5.1.) via reciprocating the RND we have that:

$$\frac{\mathrm{d}\overrightarrow{\mathbb{P}}^\mu}{\mathrm{d}\overleftarrow{\mathbb{P}}^\nu}(Y_{[t,1]}) = \frac{p_t(Y_t)}{p_1(Y_1)} R_Q^{Q'}(Y_{[t,1]})^{-1} \quad (45)$$

Then since $\mathrm{d}\overrightarrow{\mathbb{P}}^\mu/\mathrm{d}\overleftarrow{\mathbb{P}}^\nu(Y_{[t,1]}) = 1$, rearranging gives

$$p_t(Y_t) = p_1(Y_1) R_Q^{Q'}(Y_{[t,1]}). \quad (46)$$

$\square$

Notice that in discrete diffusion, one can integrate the Kolmogorov forward equation in backward time numerically

$$\partial_t p_t = -{Q'}_t^\top p_t \tag{47}$$

and then index into the vector $p_t$ with $Y_t$ to obtain $p(Y_t)$, however this has the computational cost of $\mathcal{O}(\text{Number of Steps} \times (\text{Vocabulary Size})^2)$. Instead, our estimator can be run online whilst generating samples at a cost of $\mathcal{O}(\text{Number of Steps} \times \text{Vocabulary Size})$, making our RNDE likelihood computation for general CTMCs much more tractable. We highlight that our speed gain holds even in settings where the concrete score is time-independent, and we are able to obtain a matrix exponential solution to the Kolmogorov equation (Ou et al., 2024a). This is because the closed-form solution does not scale for large vocabularies, as it requires diagonalising $Q'_t$.

Unlike the Kolmogorov equation solvers, our RNDE introduces bias in practice due to the time reversal being approximate. To mitigate this, we can similarly derive an RNDE-IS-based estimator, which should coincide with the marginal density relation in (Campbell et al., 2024, Section C.1.1., Page 23) used to derive the ELBO objective in discrete diffusions from a continuous time setting.

### D.1 RNC FOR DISCRETE DIFFUSION

Now that we have defined $R$ for CTMC, our RN Corrector can be readily applied, yielding similar estimators to Lee et al. (2025), but generalising to more tasks such as annealing and reward-tilting.

As with RNC for SDEs, we have the same setup,

1. **a backward sampling process:** $\partial_t \rho_t = -A_t^\top \rho_t$

2. **a forward target process:** $\partial_t h_t = B_t^\top h_t$

3. **intermediate target marginal densities:** $q_t$, which are a function of $p_t$ satisfying $\partial_t p_t = Q_t^\top p_t$.

To give a concrete example let us write the weight for product:

$$w_{[\tau,\tau']} \propto \left[ R_{Q'^{(1)}}^{Q^{(1)}}(X_{[\tau,\tau']}) \right]^\alpha \left[ R_{Q'^{(2)}}^{Q^{(2)}}(X_{[\tau,\tau']}) \right]^\beta \left[ R_B^A(X_{[\tau,\tau']}) \right]^{-1} \tag{50}$$

Where $Q'^{(i)}, Q^{(i)}$ are the rate matrices of a CTMC and its reversal and $B$ is the rate matrix corresponding to a target process CTMC, and $A$ is the rate matrix of sampling process/proposal.

### D.2 DISCRETISATION OF $R$ FOR CTMC

As in the continuous state case, we can approximate $R$ as a product of discrete-time kernels.

$$R_Q^{Q'}(Y_{[\tau,\tau']}) \approx \frac{\prod_{n=1}^{N-1} p_{n|n+1}^{Q'}(Y_{t_n}|Y_{t_{n+1}})}{\prod_{n=1}^{N-1} p_{n+1|n}^{Q}(Y_{t_{n+1}}|Y_{t_n})}. \tag{51}$$

this can be seen formalised in Holderrieth et al. (2025, Appendix A), and the discrete kernels can be approximated using Eulers method (Campbell et al., 2022)

$$p_{n+1|n}^{Q}(Y_{t+\Delta t} \mid Y_t) = \delta_{Y_{t+\Delta t}, Y_t} + Q_t(Y_t, Y_{t+\Delta t})\,\Delta t + o(\Delta t) \tag{52}$$

$$p_{n|n+1}^{Q'}(Y_t \mid Y_{t+\Delta t}) = \delta_{Y_t, Y_{t+\Delta t}} + Q'_{t+\Delta t}(Y_{t+\Delta t}, Y_t)\,\Delta t + o(\Delta t) \tag{53}$$

# E    CONNECTION BETWEEN RNE AND OTHER APPROACHES

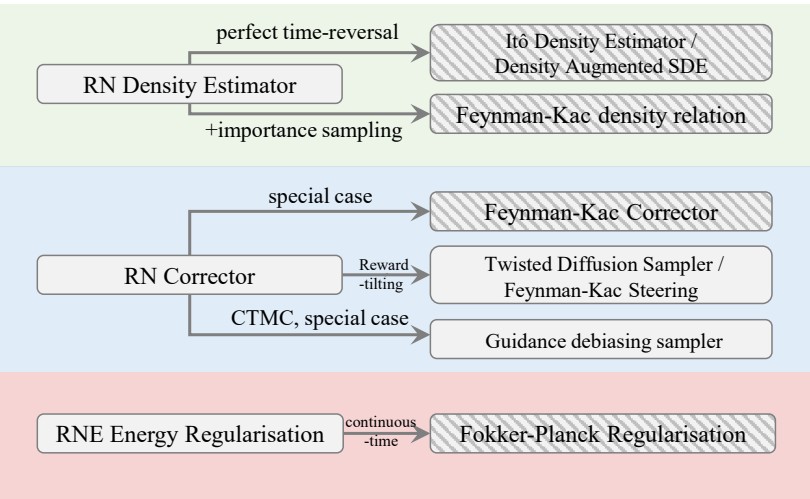

Fig 10: Connection between RNE and other density estimation & inference-time control & energy regularisation approaches. Methods with a grey striped background generally require divergence computation/estimation, and divergence-free options are available only in specific cases.

One of the key contributions of RNE is that it provides a unifying perspective, connecting several previously proposed approaches within a single framework. While we have highlighted these connections throughout the method description, in this section, we present a concise summary:

- For inference-time control, RNE recovers the Feynman–Kac corrector (Skreta et al., 2025) as a special case in continuous-time, the Twisted Diffusion Sampler (Wu et al., 2023) and Feynman–Kac steering (Singhal et al., 2025) for reward tilting, as well as the debiasing method by Lee et al. (2025) for guidance in CTMC.

- For density estimation, RNE is equivalent to the density-augmented SDE approach (Karczewski et al., 2024) and its concurrent work, Itô's density estimator (Skreta et al., 2024) in continuous-time. When coupled with importance sampling, RNE further recovers the Feynman–Kac density relation as well as the diffusion density estimators proposed by Huang et al. (2021) and Premkumar (2024).

- For energy-based training, RNE recovers the Fokker-Planck regularisation (Plainer et al., 2025), with a simpler interpretation and cheaper calculation.

We summarise these connections in Fig. 10.

# F    ADDITIONAL EXPERIMENTS AND ANALYSIS

## F.1    RNDE AND ABLATION ON REFERENCE PROCESS

To assess the effectiveness of RN Density Estimator (RNDE) proposed in Appendices C.1 and C.2, we choose a 10-D Mixture-of-Gaussian target with 40 modes, which was initially used by Midgley et al. (2022) to evaluate the performance of Boltzmann generators. Since we can access the analytical marginal density at any diffusion time step $t$, it is ideally suited for comparing different density estimation methods. In Fig. 11, we compare four estimators: RNDE with reference, RNDE without reference, importance-sampling-based RNDE, and Itô density estimator (Skreta et al., 2024). We use the variance-exploding (VE) diffusion with the exact score function, and we follow the discretisation schedule of Karras et al. (2022). For the reference process, we adopt the same VE process starting from a standard Gaussian. The vanilla RNDE underperforms the Itô estimator; however, incorporating the reference process leads to substantially better density estimates. Incorporating IS further improves

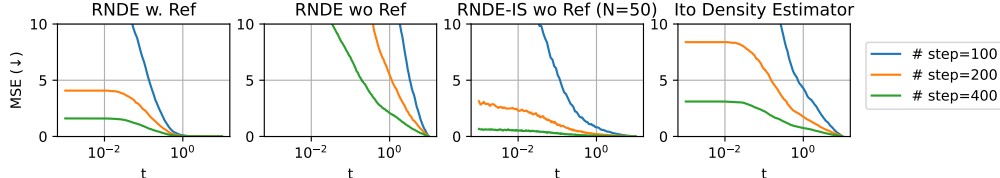

Fig 11: MSE of the log diffusion density $\log p_t$, against diffusion time $t$. Different colour represents different number of discretisation steps. We compare 4 different approaches: RNDE with reference, RNDE without reference, RNDE-IS with 50 samples, and RNE Itô density estimator (Skreta et al., 2024). We use a VE process following Karras et al. (2022) where $t \in [0, 10]$, $p_{t=10} \to N(0, 10^2 I)$ and $p_0 = p_{\text{data}}$. Hence, the error increases when $t$ gets closer to 0.

performance, albeit at the expense of increased computational cost. A more detailed analysis of importance sampling is provided in the next section.

## F.2 ANALYSIS ON RNDE-IS

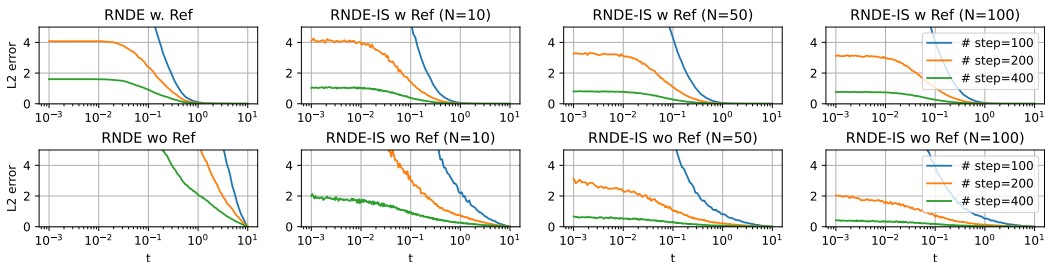

Fig 12: MSE of density estimation by RNDE and RNDE-IS across varying sample sizes, shown with and without a reference process. Different colour represents different number of discretisation steps.

In this section, we provide a more comprehensive analysis of the performance of RNDE-IS and the influence of the reference process. In Fig. 12, we show the MSE of density estimation by RNDE and RNDE-IS across varying sample sizes, both with and without a reference process. As we can see,

- when sample size is small (or without IS), using reference can significantly boost the performance;
- when the sample size is large enough, the reference will negatively influence the performance.

This behaviour is as expected: when the sample size is small, as we motivated in Section 3, the reference is used to address the instability of RNE. This instability is eliminated when the sample size is large enough. On the other hand, this reference will bring in its own discretisation error. This error is negligible compared to its benefits when the sample size is small, while becoming significant when the sample size is large enough. However, we stress that in the application of RNDE and RNC, we usually rely on estimation using just one sample, and hence reference is always favourable there.

## F.3 INFERENCE-TIME ANNEALING: MORE ANALYSIS

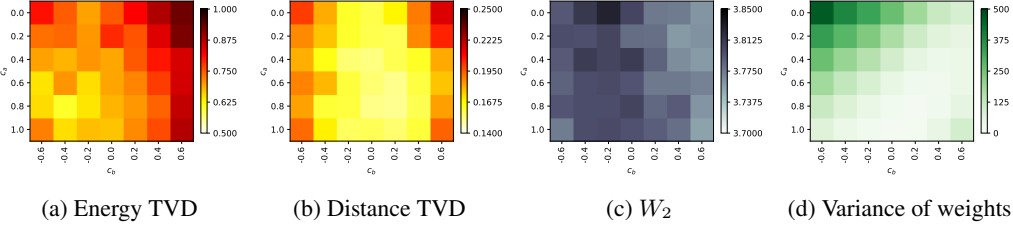

(a) Energy TVD     (b) Distance TVD     (c) $W_2$     (d) Variance of weights

Fig 13: Inference-time annealing for LJ-13.

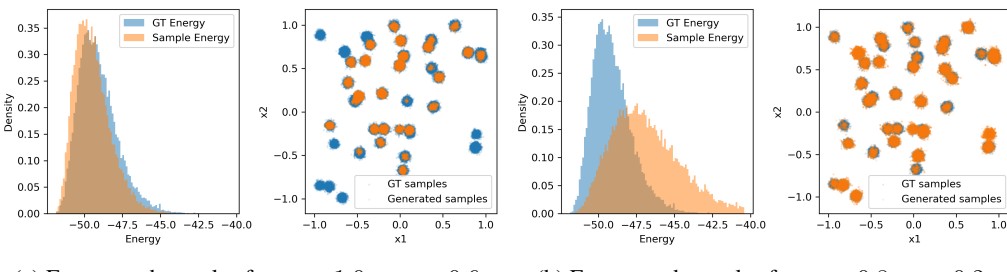

(a) Energy and samples for $c_a = 1.0, c_b = -0.6$.     (b) Energy and samples for $c_a = 0.8, c_b = 0.2$.

Fig 14: Visualisation of inference-time annealing on 10D Mixture of Gaussian target.

In this section, we provide a more comprehensive analysis of RNC with different $c_a$ and $c_b$.

We first consider the Lennard-Jones (LJ) system with 13 particles. We first train a diffusion model for temperature $T_{\text{high}} = 2.0$ and anneal it down to $T_{\text{low}} = 1.0$. Similar to ALDP, we sample using a batch size of 500, and repeat this 50 times to collect all data. We chose this system as it only has one peaked mode, showing different properties compared to the ALDP in the main text.

In Fig. 13, we show the TVD between the energy histogram, the interatomic distance histogram, the $W_2$ distance and the variance of accumulated weights. As we can see, the variance of accumulated weights features the same pattern as ALDP in Fig. 2, showing that $c_a + c_b$ within $1 \pm 0.2$ typically achieves a better effective sample size (ESS) compared to other choices. However, energy TVD exhibits the opposite trend: it actually improves in regions where the weight variance is large. This can be explained by the property of the LJ-potential. As the distribution has a very peaked mode, we generally do not need high diversity in the samples—even though ESS is so low that all 500 draws in a batch collapse to the same particle, the SMC weights will still single out the sample that best aligns with the mode. By contrast, when ESS is high, the algorithm may occasionally select a suboptimal configuration, potentially due to the intrinsic SMC bias associated with a finite sample size, and can be amplified by discretisation error and model imperfections.

This can also be observed in a toy Gaussian Mixture target. Specifically, we consider anneal a 10D GMM target from $T_{\text{high}} = 1$ to $T_{\text{low}} = 1/3$. We run RNC with the analytical score from the GMM target, using a batch size of 500, and collect 100 batches in total. In Fig. 14, we visualise two settings (a) $c_a = 1.0, c_b = -0.6$, resulting in a low ESS; and (b) $c_a = 0.8, c_b = 0.2$, resulting in a higher ESS. In case (a), it exhibits significant mode collapse, with the samples being closer to the centre of each mode. However, the energy histogram shows a good alignment with the ground truth energy. By contrast, in case (b) the samples almost cover all the modes, but a few particles diffuse slightly, and hence the energy histogram deviates more from the ground truth.

**In summary**, SMC intrinsically struggles with sample diversity. In terms of RNC, different choices of $c_a, c_b$ yield different ESS values. This ESS (weight's variance) pattern is highly consistent across different targets. In general, a larger ESS is preferable to ensure greater diversity. However, for certain targets, a lower ESS can also be advantageous. Although there is no universally optimal choice, we argue that if preserving the entire distribution is the goal, one should aim for the highest possible ESS; conversely, if the objective is merely to select the single best sample, it may be beneficial to pick $c_a, c_b$ that leads to a slightly smaller ESS.

### F.4 EMPIRICAL ANALYSIS ON IMPERFECT SCORE AND DISCRETISATION ERROR

In Proposition 3.2, we discussed the theoretical guarantee of the SMC weight calculated by RNE when the discretisation error and score network error are present. In this section, we evaluate the influence of these two errors empirically, taking inference-time annealing on ALDP, for example. To analyse the influence of discretisation error, we choose to run RNC with 20, 100, 200, 400 discretisation steps in the diffusion generation process; to analyse the influence of imperfect score, we choose to early stop the diffusion network training stage after different numbers of iterations. For comparison, we report the heuristic choice by simply annealing the score without SMC. As we can see from Figs. 15 and 16, the performance of RNC increases w.r.t. the number of steps and the number of training iterations, as expected from Proposition 3.2. Also, while RNC performs worse

when using a small number of steps and when the score has a larger error, it still presents empirical performance gain compared to the heuristic choice.

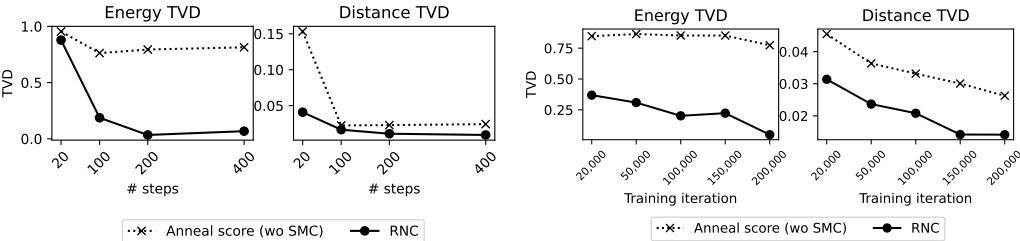

Fig 15: Influence of discretisation error.  Fig 16: Influence of score error.

### F.5 ABLATION ON REFERENCE PROCESS FOR BETTER DENOISING KERNELS

One line of work (Bao et al., 2022; Ou et al., 2024b) aims to estimate the variance of the denoising kernel, resulting in a denoising kernel that is more accurate than the one obtained from simple EM discretisation. A natural question is whether our proposed reference process still provides gains when used together with such improved denoising kernels. In this section, we investigate this empirically. Specifically, we evaluate the performance of RNE for density estimation (RNDE) on a 10D GMM with 40 modes, for which the marginal density is available in closed form, allowing us to directly assess the accuracy of our RNE estimator.

We visualise the results in Fig. 17. In Fig. 17(a), we show the results obtained using EM discretisation, while in Fig. 17(b), we estimate the variance $\text{Var}[x_{t_{n-1}} \mid x_{t_n}]$ following Ou et al. (2024b, Theorem 1). We can see that using the estimated variance substantially improves the accuracy of the RNE estimator. Furthermore, even in this setting, incorporating our reference process still provides an additional boost, further reducing the error and yielding highly accurate estimates.

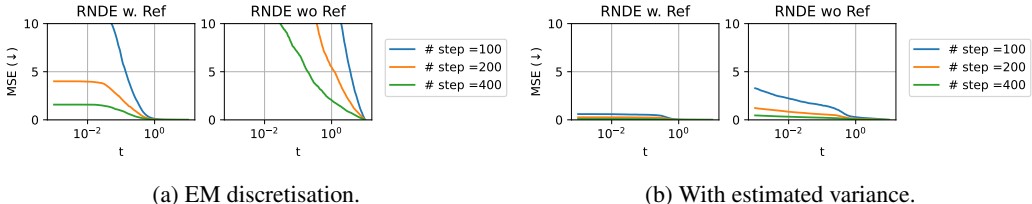

(a) EM discretisation.  (b) With estimated variance.

Fig 17: MSE of the RNE-estimated log diffusion density $\log p_t$, against diffusion time $t$.

## G DISCRETISATION, STABILITY AND CONVERGENCE GUARANTEES

In this section, we provide additional details and discussion on the discretisation error and convergence guarantees for RNE with reference.

### G.1 INTUITION FOR INSTABILITY WITHOUT REFERENCE

To understand the instability without reference process, let's consider the following example: given a diffusion model with a forward VE-SDE $dX_t = \epsilon_t \overrightarrow{dW_t}$, and a backward score SDE $dX_t = -\epsilon_t^2 s_t(X_t)dt + \epsilon_t \overleftarrow{dW_t}$, the contribution to $R$ from the final denoising step (from $\Delta t$ to 0) is

$$\frac{\mathcal{N}(X_0|X_{\Delta t} + \epsilon_{\Delta t}^2 s_{\Delta t}(X_1)\Delta t, \epsilon_{\Delta t}^2 \Delta t)}{\mathcal{N}(X_{\Delta t}|X_0, \epsilon_0^2 \Delta t)} \approx \exp\left(\xi^2 \left(\frac{\epsilon_{\Delta t}^2}{2\epsilon_0^2} - \frac{1}{2}\right)\right), \ \xi \sim \mathcal{N}(0, I). \quad (54)$$

If $\epsilon_t$ decreases quickly as with many noise schedulers (Nichol & Dhariwal, 2021; Karras et al., 2022) and if $\epsilon_0$ is small, the term $\epsilon_{\Delta t}^2/\epsilon_0^2$ becomes large and unstable. At a high level, this issue arises because, at each discretisation step, the variances of the forward and backward kernels are misaligned. In fact, this misalignment also introduces accumulated error, as discussed in Appendix G.2.

## G.2 CONTINUOUS FORMULATION VS DISCRETE GAUSSIAN KERNELS FOR $R$

In the main text, we introduced RNE in the form of a limiting ratio between sequences of Gaussian kernels, as described in Eq. (10). Another equivalent formulation, as we discussed in Eq. (12), directly expresses RNE in continuous time in terms of stochastic integrals.

In practice, for a finite number of steps $N$, these two formulations have very different behaviours. The Gaussian kernel formulation (without reference) typically suffers from higher accumulated error when the diffusion coefficient $\epsilon_t$ is not constant; while applying Euler-Maruyama to Eq. (12) will not have this issue. Specifically, we have the following conclusion:

> Denote the result obtained by applying Euler-Maruyama to Eq. (12) with $N$ steps as $R_N$, and the result obtained by Gaussian kernel formulation (without reference) as $G_N$, then
>
> $$\Delta_N = \log R_N - \log G_N \approx \sum_n d \frac{\epsilon_{t_n}^2 - \epsilon_{t_{n+1}}^2}{2\epsilon_{t_n}^2} - d \log \frac{\epsilon_1}{\epsilon_0}, \qquad (55)$$
>
> where $d$ is the dimensionality.

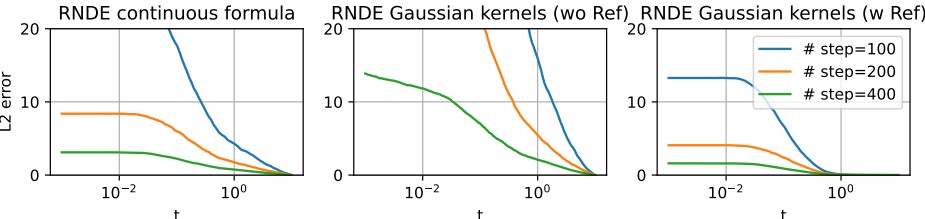

Fig 18: RN Density estimator calculated by three different ways: Left: Euler-Maruyama to continuous path integral (i.e., Eq. (12)); Middle: Gaussian kernels (i.e., Eq. (10)); Right: Gaussian kernels with reference (i.e., Eq. (24)). Different colour represents different number of discretisation steps.

We empirically verify this in Fig. 18. As we can see, directly using Gaussian kernels will result in a significant error, echoing our earlier discussion. However, fortunately, by adding the reference process as described in Section 3, we successfully address this issue and achieve the best performance out of the different estimators and discretisations we consider.

In what follows, we will analyse and discuss this error in more detail:

We first consider Eq. (12):

$$\log R_\mu^\nu(Y_{[\tau,\tau']}) = \int_\tau^{\tau'} \frac{1}{\epsilon_t^2} \nu_t \cdot \overleftarrow{dY_t} - \int_\tau^{\tau'} \frac{1}{\epsilon_t^2} \mu_t \cdot \overrightarrow{dY_t} + \frac{1}{2} \int_\tau^{\tau'} \frac{1}{\epsilon_t^2}(||\mu_t||^2 - ||\nu_t||^2)dt, \qquad (56)$$

For simplicity, we consider discretising $R$ using an equidistant step size ($\Delta t$ is constant). Discretise using Euler-Maruyama, and denote its discrete version as $R_N$, we obtain

$$\log R_N = \sum_n \frac{1}{\epsilon_{t_{n+1}}^2} \nu_{t_{n+1}} \cdot (Y_{t_{n+1}} - Y_{t_n}) - \sum_n \frac{1}{\epsilon_{t_n}^2} \mu_{t_n} \cdot (Y_{t_{n+1}} - Y_{t_n})$$
$$\underbrace{- \sum_n \frac{1}{2\epsilon_{t_{n+1}}^2} ||\nu_{t_{n+1}}||^2 \Delta t_n}_{(1)} + \sum_n \frac{1}{2\epsilon_{t_n}^2} ||\mu_{t_n}||^2 \Delta t_n, \qquad (57)$$

Note that since (1) is a standard Riemann integral without a stochastic integrator, we can discretise it by evaluating the integrand anywhere in the interval i.e. $\sum_n \frac{1}{2\epsilon_{\tau_n^*}^2} ||\nu_{\tau_n^*}||^2 \Delta t_n, \forall \tau_n^* \in [t_n, t_{n+1}]$. This can be done since the upper and lower Darboux integrals are equal to each other. Following Vargas et al. (2023b) we choose $\sum_n \frac{1}{2\epsilon_{t_{n+1}}^2} ||\nu_{t_{n+1}}||^2 \Delta t_n$ to be more consistent with how we discretised the backwards integral and to better align with the Gaussian density ratio below.

Now, let's consider the discrete Gaussian estimator of the RND (which we denote by $G_N$):

$$\log G_N = \log \frac{\prod_n \mathcal{N}\left(Y_{t_{n+1}}; Y_{t_n} + \mu_{t_n}(Y_{t_n})\Delta t,\ \epsilon_{t_n}^2 \Delta t I\right)}{\prod_n \mathcal{N}\left(Y_{t_n}; Y_{t_{n+1}} - \nu_{t_{n+1}}(Y_{t_{n+1}})\Delta t,\ \epsilon_{t_{n+1}}^2 \Delta t I\right)}, \tag{58}$$

expanding (where $d$ is the dimensionality):

$$-\sum_n \frac{||Y_{t_{n+1}} - Y_{t_n} - \mu_{t_k}(Y_{t_n})\Delta t||^2}{2\Delta t \epsilon_{t_n}^2} + \frac{||Y_{t_{n+1}} - Y_{t_n} + \nu_{t_{n+1}}(Y_{t_{n+1}})\Delta t||^2}{2\Delta t \epsilon_t^2} + d\log \frac{\epsilon_{t_{n+1}}}{\epsilon_{t_k}}, \tag{59}$$

expanding again:

$$\sum_n -\frac{||Y_{t_{n+1}} - Y_{t_n}||^2}{2\Delta t \epsilon_{t_n}} + \frac{(Y_{t_{n+1}} - Y_{t_n}) \cdot \mu_{t_n}}{\epsilon_{t_n}^2} - \frac{||\mu_{t_n}||^2 \Delta t}{2\epsilon_{t_n}^2} \tag{60}$$

$$+ \sum_n \frac{||Y_{t_{n+1}} - Y_{t_n}||^2}{2\Delta t \epsilon_{t_{n+1}}^2} - \frac{(Y_{t_{n+1}} - Y_{t_n}) \cdot \nu_{t_{n+1}}}{\epsilon_{t_{n+1}}^2} + \frac{||\nu_{t_{n+1}}||^2 \Delta t}{2\epsilon_{t_{n+1}}^2} + d\log \frac{\epsilon_{t_{n+1}}}{\epsilon_{t_n}}, \tag{61}$$

we can then see that:

$$\Delta_N = \log R_N - \log G_N = \sum_n \underbrace{\frac{||Y_{t_{n+1}} - Y_{t_n}||^2}{2\Delta t \epsilon_{t_{n+1}}^2}}_{(a)} - \underbrace{\frac{||Y_{t_{n+1}} - Y_{t_n}||^2}{2\Delta t \epsilon_{t_n}}}_{(b)} - d\log \frac{\epsilon_{t_{n+1}}}{\epsilon_{t_n}}, \tag{62}$$

From the convergence rates of the total variation and the Euler-Maruyama discretisation, we know that (up to an error in $\sqrt{\Delta t}$),

$$(Y_{t_{n+1},i} - Y_{t_n,i})^2 \approx \epsilon_{t_{n+1}}^2 \Delta t, \tag{63}$$

where $i$ is the i-th dimension index, and the approximation can be understood as a.s. or in $L_2$ and can be formalised as an upper bound, but for ease of presentation, we chose $\approx$. Substituting this back into $(a)$, we see:

$$\sum_n \frac{||Y_{t_{n+1}} - Y_{t_n}||^2}{2\Delta t \epsilon_{t_{n+1}}^2} \approx \sum_k \frac{d\epsilon_{t_{n+1}}^2 \Delta t}{2\Delta t \epsilon_{t_{n+1}}^2} = \frac{dN}{2}, \tag{64}$$

and for $(b)$, we have

$$\sum_n \frac{||Y_{t_{n+1}} - Y_{t_n}||^2}{2\Delta t \epsilon_{t_n}^2} \approx \frac{d}{2} \sum_n \frac{\epsilon_{t_{n+1}}^2}{\epsilon_{t_n}^2}, \tag{65}$$

combining the two

$$\Delta_N \approx \sum_n d \frac{\epsilon_{t_n}^2 - \epsilon_{t_{n+1}}^2}{2\epsilon_{t_n}^2} - d\log \frac{\epsilon_1}{\epsilon_0}, \tag{66}$$

Indicating that the deviation between Eq. (12) and Eq. (10) can be large in practice when $N$ is not large.

In the limit (Berner et al., 2025, Lemma B.7), we do note that this vanishes as

$$\Delta_N \approx -\frac{d}{2}\int_0^1 \epsilon_t^{-2}\mathrm{d}\epsilon_t^2 - d\log \frac{\epsilon_1}{\epsilon_0} = d(\ln \epsilon_1 - \ln \epsilon_0) - d\log \frac{\epsilon_1}{\epsilon_0} = 0, \tag{67}$$

To have more intuition on the magnitude of this error, we consider a VE-SDE with $\epsilon_t = a^t$ where $a^2 = \mathrm{var}_{max}/\mathrm{var}_{min}$ into Eq. (66).

$$\sum_n d \frac{\epsilon_{t_n}^2 - \epsilon_{t_{n+1}}^2}{2\epsilon_{t_n}^2} = \sum_n d \frac{a^{2t_n} - a^{2(t_n + \Delta t)}}{2a^{2t_n}} = \frac{d}{2}N - \frac{dN}{2}\left(\frac{\mathrm{var}_{max}}{\mathrm{var}_{min}}\right)^{1/N}, \tag{68}$$

and thus

$$\Delta_N \approx \frac{d}{2}N - \frac{dN}{2}\left(\frac{\mathrm{var}_{max}}{\mathrm{var}_{min}}\right)^{1/N} + \frac{d}{2}\log \frac{\mathrm{var}_{max}}{\mathrm{var}_{min}}, \tag{69}$$

If we choose $\mathrm{var}_{max} = 10^2$, $\mathrm{var}_{min} = 0.001^2$ and $N = 100$, then $|\Delta_N| \approx 0.87d$ leading to large deviations from $\log R_N$ when in high dimensions.

### G.3 RNE WITH REFERENCE - CONVERGENCE RATE

#### G.3.1 CONVERGENCE RATE FOR RNE

We now show our reference-based RNE's convergence rate. Furthermore the analytic reference based estimator we use allows us proof such a rate without needing to bound the error of estimating forward integrals with backwards samples or vice versa, as the reference cancels the mismatch in integral directions.

Whilst a proof for the non-asymptotic guarantees of the continuous-time RNE estimator without reference should be possible, it will require a lot more technical effort, which we will leave for future discussion.

Let's consider the case where we use the diffusion model defined in Eqs. (1) and (2), where $\mu_t = f_t$ is a linear function, and $\nu_t = \mu_t - \sigma_t^2 \nabla \log p_t$ is the backward drift. We now choose a reference process whose forward drift $\phi_t = \mu_t$, and the initial marginal to be Gaussian. In this case, we can analytically solve the marginal $\pi_t$ at any time step $t$ and hence can access the analytical time-reversal of the reference process. Let's denote the drift of this time-reversal as $\psi_t = \mu_t - \sigma_t^2 \nabla \log \pi_t$.

**Proposition G.1.** *Let us consider the case where $\mu_t$ corresponds to a linear forward drift, as is the case in diffusion models and half-sided interpolants. Then we chose an analytic reference by setting its drift as $\mu_t$ and $\pi_1$ to be Gaussian, and denote its time-reversal drift as $\psi_t$. Assuming $Y$ has bounded $L^p$ moments, and calculating $R$ with reference, it follows that:*

$$|| \log R_\mu^\nu(Y_{[\tau,\tau']}) - \log R^N(\hat{Y}_{[\tau,\tau']})||_{L^2} \leq \mathcal{O}(\sqrt{\Delta t}) \tag{70}$$

*Where $||.||_{L^2}$ denotes the $L^2$ norm, $\hat{Y}$ is the Euler-Maruyama discretisation of $Y$, and $\log R^N(\hat{Y}_{[\tau,\tau']})$ is our discretised RNE estimator*

$$R^N(\hat{Y}_{[\tau,\tau']}) = \frac{\pi_\tau(\hat{Y}_\tau)}{\pi_{\tau'}(\hat{Y}_{\tau'})} \frac{\prod_{n=1}^{N-1} p_{n|n+1}^\nu(\hat{Y}_{t_n}|\hat{Y}_{t_{n+1}})}{\prod_{n=1}^{N-1} p_{n|n+1}^\psi(\hat{Y}_{t_n}|\hat{Y}_{t_{n+1}})} \frac{\prod_{n=1}^{N-1} p_{n+1|n}^\phi(\hat{Y}_{t_{n+1}}|\hat{Y}_{t_n})}{\prod_{n=1}^{N-1} p_{n+1|n}^\mu(\hat{Y}_{t_{n+1}}|\hat{Y}_{t_n})}. \tag{71}$$

*Proof.* In this setting, the reference has the same forward drift as our diffusion and hence will be cancelled, leaving only two backward processes: one for diffusion, one for the reference. Therefore:

$$\log R_\mu^\nu(Y_{[\tau,\tau']}) = \log \frac{\pi_\tau(Y_\tau)}{\pi_{\tau'}(Y_{\tau'})} + \int_\tau^{\tau'} \frac{1}{\sigma_t^2}(\nu_t - \psi_t) \cdot \overleftarrow{\mathrm{d}Y_t} - \frac{1}{2}\int_\tau^{\tau'} \frac{1}{\sigma_t^2}(||\nu_t||^2 - ||\psi_t||^2)\mathrm{d}t, \tag{72}$$

$$= \log \frac{\pi_\tau(Y_\tau)}{\pi_{\tau'}(Y_{\tau'})} + \frac{1}{2}\int_\tau^{\tau'} \frac{1}{\sigma_t^2}||\nu_t - \psi_t||^2\mathrm{d}t + \int_\tau^{\tau'} \frac{1}{\sigma_t}(\nu_t - \psi_t) \cdot \overleftarrow{\mathrm{d}W_t}, \tag{73}$$

First, let us consider the EM discretisation of $Y_t$,

$$\hat{Y}_{t_n} = \hat{Y}_{t_{n+1}} - \nu_{t_{n+1}}(\hat{Y}_{t_{n+1}})\Delta t - \sigma_{t_{n+1}}(W_{t_{n+1}} - W_{t_n}), \tag{74}$$

$$\hat{Y}_{t_n} = \hat{Y}_{t_{n+1}} - \int_{t_k}^{t_{k+1}} \nu_{t_{n+1}}(\hat{Y}_{t_{n+1}})\mathrm{d}t - \int_{t_k}^{t_{k+1}} \sigma_{t_{n+1}} \overleftarrow{\mathrm{d}W}_t, \tag{75}$$

Now we can embed this discretisation into continuous time $\bar{Y}_t = \sum_{n=1}^N \mathbb{1}_{s \in [t_n, t_{n+1}]} \hat{Y}_{t_n}$, which we can express in integral form as,

$$\bar{Y}_t = \bar{Y}_T - \int_\tau^{\tau'} \bar{\nu}_s(\bar{Y}_s)\mathrm{d}s - \int_\tau^{\tau'} \bar{\sigma}_s \overleftarrow{\mathrm{d}W_s} \tag{76}$$

where $\bar{\nu}_s(\bar{Y}_s) = \sum_{n=1}^N \mathbb{1}_{s \in [t_n, t_{n+1}]} \nu_{t_{n+1}}(\bar{Y}_{t_{n+1}})$ and $\bar{\sigma}_s = \sum_{n=1}^N \mathbb{1}_{s \in [t_n, t_{n+1}]} \sigma_{t_{n+1}}$. We also define $\bar{\psi}$ in the same way, now by Lemma G.2 we have that

$$\log R^N(\hat{Y}_{[\tau,\tau']}) = \log \frac{\pi_\tau(\hat{Y}_\tau)}{\pi_{\tau'}(\hat{Y}_{\tau'})} + \sum_n \frac{1}{\sigma_{t_{n+1}}}(\nu_{t_{n+1}} - \psi_{t_{n+1}})(\hat{Y}_s)(W_{t_{n+1}} - W_{t_n})$$
$$- \frac{1}{2}\sum_n \frac{1}{\sigma_{t_{n+1}}^2}||\nu_{t_n} - \psi_{t_n}||^2(\hat{Y}_{t_{n+1}})\Delta t, \tag{77}$$

Then, via Remark G.3 we have

$$|| \log R^\nu_\mu(Y_{[\tau,\tau']}) - \log R^N(\hat{Y}_{[\tau,\tau']})||^2_{L^2} = || \log R^\nu_\mu(Y_{[\tau,\tau']}) - \log R^N(\bar{Y}_{[\tau,\tau']})||^2_{L^2}$$

where via Jensens inequality (i.e. $||\frac{3a}{3} + \frac{3b}{3} + \frac{3c}{3}||^2 \leq \frac{1}{3}(9||a||^2 + 9||b||^2 + 9||c||^2)$) we have that

$$|| \log R^\nu_\mu(Y_{[\tau,\tau']}) - \log R^N(\bar{Y}_{[\tau,\tau']})||^2_{L^2} \leq 3(A + B + C), \tag{78}$$

where (for brevity in this step we have assumed a time homogenous $\sigma_s$ however its easy to see how this extends to the time dependent setting)

$$A = \mathbb{E}\left[\left|\left|\int_\tau^{\tau'} \frac{1}{\sigma_s}((\bar{\nu}_s - \bar{\psi}_s)(\bar{Y}_s) - (\nu_s - \psi_s)(Y_s))\overleftarrow{\mathrm{d}W}_s\right|\right|^2\right] \tag{79}$$

$$B = K_0\mathbb{E}\left[\int_\tau^{\tau'} \frac{1}{\sigma_s^2}||(\bar{\nu}_s - \bar{\psi}_s)^2(\bar{Y}_s) - (\nu_s - \psi_s)^2(Y_s)||^2\mathrm{d}s\right] \tag{80}$$

$$C = \mathbb{E}\left[|| \log \pi_\tau(Y_\tau) - \log \pi_\tau(\bar{Y}_\tau)||^2\right] + \mathbb{E}\left[|| \log \pi_{\tau'}(Y_{\tau'}) - \log \pi_{\tau'}(\bar{Y}_{\tau'})||^2\right] \tag{81}$$

now applying Itô's isometry:

$$A = \mathbb{E}\left[\left|\left|\int_\tau^{\tau'} \frac{1}{2\sigma_s^2}((\bar{\nu}_s - \bar{\psi}_s)(\bar{Y}_s) - (\nu_s - \psi_s)(Y_s))\overleftarrow{\mathrm{d}W}_s\right|\right|^2\right]$$

$$= \mathbb{E}\left[\int_\tau^{\tau'} \frac{1}{2\sigma_s}||(\bar{\nu}_s - \bar{\psi}_s)(\bar{Y}_s) - (\nu_s - \psi_s)(Y_s)||^2\mathrm{d}s\right], \tag{82}$$

then by the Lip property of the SDEs coeficients, the strong convergence of the EM scheme (Kloeden et al., 1992), and that $\sigma_t > 0$ (i.e. we bound $\int(1/\sigma_s)h_s\mathrm{d}s \leq \max_s \frac{1}{\sigma_s} \cdot \int h_s\mathrm{d}s$)

$$A \leq L \sum_n \int_{t_n}^{t_{n+1}} \mathbb{E}||\bar{Y}_s - Y_s||^2\mathrm{d}s \leq L\Delta t, \tag{83}$$

For $B$, let's label $f = (\nu_s - \psi_s), \bar{f} = (\bar{\nu}_s - \bar{\psi}_s)$:

$$||\bar{f}^2 - f^2||^2 = ||(\bar{f} - f)(\bar{f} + f)||^2. \tag{84}$$

Then we can use Cauchy–Schwarz inequality:

$$\mathbb{E}[\int_\tau^{\tau'} \frac{1}{\sigma_s^4}||(\bar{f} - f)(\bar{f} + f)||^2\mathrm{d}s] \leq K_1 \left(\int_\tau^{\tau'} \mathbb{E}[||(\bar{f} - f)||^4]\mathrm{d}s\right)^{1/2} \left(\int_\tau^{\tau'} \mathbb{E}[||(\bar{f} + f)||^4]\mathrm{d}s\right)^{1/2}. \tag{85}$$

Via the Lipschitz property of $f$, we have that

$$\int_\tau^{\tau'} \mathbb{E}[||(\bar{f} - f)||^4]\mathrm{d}s \leq K_2 \int_\tau^{\tau'} \mathbb{E}[||\bar{Y}_s - Y_s||^4]\mathrm{d}s, \tag{86}$$

$$\leq K_1 \sum_n \int_{t_n}^{t_{n+1}} \mathbb{E}[||\bar{Y}_s - Y_s||^4]\mathrm{d}s. \tag{87}$$

Remark 1.2 in Gyöngy & Rásonyi (2011) ($L^p$ convergence of EM scheme) implies that:

$$\int_{t_n}^{t_{n+1}} \mathbb{E}[||\bar{Y}_s - Y_s||^4]\mathrm{d}s \leq K_3\Delta t^2, \tag{88}$$

and thus

$$\left(\int_\tau^{\tau'} \mathbb{E}[||(\bar{f} - f)||^4]\right)^{1/2} \mathrm{d}s \leq K_4\Delta t. \tag{89}$$

Also, $(\int_\tau^{\tau'} \mathbb{E}[||(\bar{f}+f)||^4]ds)^{1/2}$ is bounded from Novikov's condition (which we assume all throughout) + Jensensen's inequality. Thus,

$$B \le K'\Delta t. \tag{90}$$

Given log concavity for $\pi_\tau$ (as it is the density of the analytic reference, which is Gaussian), we have that

$$\log \pi_\tau(Y_\tau) - \log \pi_\tau(\bar{Y}_\tau) \le \nabla \log \pi_\tau(Y_\tau) \cdot (Y_\tau - \bar{Y}_\tau), \tag{91}$$

thus by Cauchy-Schwartz

$$\mathbb{E}[|\log \pi_\tau(Y_\tau) - \log \pi_\tau(\bar{Y}_\tau)|^2] \le \mathbb{E}[||\nabla \log \pi_\tau(Y_\tau)||^2||Y_\tau - \bar{Y}_\tau||^2]. \tag{92}$$

Applying Cauchy-Schwarz inequality, we have

$$\mathbb{E}[||\nabla \log \pi_\tau(Y_\tau)||^2||Y_\tau - \bar{Y}_\tau||^2] \le \mathbb{E}[||\nabla \log \pi_\tau(Y_\tau)||^4]^{1/2}\mathbb{E}[||Y_\tau - \bar{Y}_\tau||^4]^{1/2} \tag{93}$$

then since $\nabla \log \pi_\tau$ is linear, it follows that $\mathbb{E}[||\nabla \log \pi_\tau(Y_\tau)||^4]^{1/2}$ is bounded and thus

$$C \le H\Delta t. \tag{94}$$

Here $L$, $K_0$, $K_1$, $K_2$, $K_3$, $K_4$, $K'$ and $H$ are constants. □

Now we introduce a few auxiliary results we needed to derive our convergence rate.

**Lemma G.2.** *In this setting, the discrete RNE estimator with reference,*

$$R^N(\hat{Y}_{[\tau,\tau']}) = \frac{\pi_\tau(\hat{Y}_\tau)}{\pi_{\tau'}(\hat{Y}_{\tau'})} \frac{\prod_{n=1}^{N-1} p_{n|n+1}^\nu(\hat{Y}_{t_n}|\hat{Y}_{t_{n+1}})}{\prod_{n=1}^{N-1} p_{n|n+1}^\psi(\hat{Y}_{t_n}|\hat{Y}_{t_{n+1}})} \frac{\prod_{n=1}^{N-1} p_{n+1|n}^\phi(\hat{Y}_{t_{n+1}}|\hat{Y}_{t_n})}{\prod_{n=1}^{N-1} p_{n+1|n}^\mu(\hat{Y}_{t_{n+1}}|\hat{Y}_{t_n})}. \tag{95}$$

*Simplifies to*

$$\log R^N(\hat{Y}_{[\tau,\tau']}) = \log \frac{\pi_\tau(\hat{Y}_\tau)}{\pi_{\tau'}(\hat{Y}_{\tau'})} + \sum_n \frac{1}{\sigma_{t_{n+1}}^2}(\nu_{t_{n+1}} - \psi_{t_{n+1}})(\hat{Y}_s)(\hat{Y}_{t_{n+1}} - \hat{Y}_{t_n})$$
$$- \frac{1}{2}\sum_n \frac{1}{\sigma_{t_{n+1}}^2}(\nu_{t_n}^2(\hat{Y}_{t_{n+1}}) - \psi_{t_n}^2(\hat{Y}_{t_{n+1}}))\Delta t, \tag{96}$$

*Proof.* First note that, as the reference has the same forward drift as the diffusion process, we have

$$R^N(\hat{Y}_{[\tau,\tau']}) = \frac{\pi_\tau(\hat{Y}_\tau)}{\pi_{\tau'}(\hat{Y}_{\tau'})} \frac{\prod_{n=1}^{N-1} p_{n|n+1}^\nu(\hat{Y}_{t_n}|\hat{Y}_{t_{n+1}})}{\prod_{n=1}^{N-1} p_{n|n+1}^\psi(\hat{Y}_{t_n}|\hat{Y}_{t_{n+1}})} \tag{97}$$

For brevity, we will use $\log R^N$, by substituting in the Gaussian kernels we have that:

$$\log R^N = \log \frac{\pi_\tau(\hat{Y}_\tau)}{\pi_{\tau'}(\hat{Y}_{\tau'})} + \sum_n \frac{||\hat{Y}_{t_{n+1}} - \hat{Y}_{t_n} - \psi_{t_k}(\hat{Y}_{t_n})\Delta t||^2}{2\Delta t\sigma_{t_n}^2} - \frac{||\hat{Y}_{t_{n+1}} - \hat{Y}_{t_n} + \nu_{t_{n+1}}(\hat{Y}_{t_{n+1}})\Delta t||^2}{2\Delta t\sigma_t^2}. \tag{98}$$

Expanding squares and collecting terms yields

$$\log R^N = \log \frac{\pi_\tau(\hat{Y}_\tau)}{\pi_{\tau'}(\hat{Y}_{\tau'})} + \sum_n \frac{1}{\sigma_{t_{n+1}}^2}(\nu_{t_{n+1}} - \psi_{t_{n+1}})(\hat{Y}_s)(\hat{Y}_{t_{n+1}} - \hat{Y}_{t_n})$$
$$- \frac{1}{2}\sum_n \frac{1}{\sigma_{t_{n+1}}^2}(\nu_{t_n}^2(\hat{Y}_{t_{n+1}}) - \psi_{t_n}^2(\hat{Y}_{t_{n+1}}))\Delta t. \tag{99}$$

□

*Remark* G.3. By embedding the discretised $\hat{Y}$ into continuous time $\bar{Y}$, we will not change the value of $R^N$, i.e., $R^N(\bar{Y}_{[\tau,\tau']}) = R^N(\hat{Y}_{[\tau,\tau']})$.

*Proof.* Let's consider the continous time embedding formula of the RNDE estimator:

$$\log R^N(\bar{Y}_{[\tau,\tau']}) = \log \frac{\pi_\tau(\bar{Y}_\tau)}{\pi_{\tau'}(\bar{Y}_{\tau'})} + \sum_n \int_{t_n}^{t_{n+1}} \frac{1}{\bar{\sigma}_s^2}(\bar{\nu}_s - \bar{\psi}_s)(\bar{Y}_s)\overleftrightarrow{\mathrm{d}\bar{Y}}_s \tag{100}$$

$$- \sum_n \frac{1}{2} \int_{t_n}^{t_{n+1}} \frac{1}{\bar{\sigma}_s^2}(\bar{\nu}_s^2(\bar{Y}_s) - \bar{\psi}_s^2(\bar{Y}_s))\mathrm{d}s, \tag{101}$$

now by construction inside the interval $[t_n, t_{n+1}]$ the integrands are constant and thus

$$\log R^N(\bar{Y}_{[\tau,\tau']}) = \log \frac{\pi_\tau(\bar{Y}_\tau)}{\pi_{\tau'}(\bar{Y}_{\tau'})} + \sum_n \frac{1}{\bar{\sigma}_{t_{n+1}}^2}(\bar{\nu}_{t_{n+1}} - \bar{\psi}_{t_{n+1}})(\bar{Y}_s) \int_{t_n}^{t_{n+1}} \overleftrightarrow{\mathrm{d}\bar{Y}}_s \tag{102}$$

$$- \sum_n \frac{1}{\bar{\sigma}_{t_{n+1}}^2}(\bar{\nu}_{t_{n+1}}^2(\bar{Y}_s) - \bar{\psi}_{t_{n+1}}^2(\bar{Y}_s))\frac{1}{2} \int_{t_n}^{t_{n+1}} \mathrm{d}s, \tag{103}$$

and,

$$\log R^N(\bar{Y}_{[\tau,\tau']}) = \log \frac{\pi_\tau(\bar{Y}_\tau)}{\pi_{\tau'}(\bar{Y}_{\tau'})} + \sum_n \frac{1}{\sigma_{t_{n+1}}^2}(\nu_{t_{n+1}} - \psi_{t_{n+1}})(\bar{Y}_s)(\bar{Y}_{t_{n+1}} - \bar{Y}_{t_n}) \tag{104}$$

$$- \sum_n \frac{1}{\sigma_{t_{n+1}}^2}(\nu_{t_{n+1}}^2(\bar{Y}_s) - \psi_{t_{n+1}}^2(\bar{Y}_s))\Delta t, \tag{105}$$

where by construction $(\bar{Y}_{t_{n+1}} - \bar{Y}_{t_n}) = (\hat{Y}_{t_{n+1}} - \hat{Y}_{t_n})$ thus we have embedded such that

$$\log R^N(\bar{Y}_{[\tau,\tau']}) = \log \frac{\pi_\tau(\hat{Y}_\tau)}{\pi_{\tau'}(\hat{Y}_{\tau'})} + \sum_n \frac{1}{\sigma_{t_{n+1}}^2}(\nu_{t_{n+1}} - \psi_{t_{n+1}})(\hat{Y}_s)(\hat{Y}_{t_{n+1}} - \hat{Y}_{t_n})$$

$$- \frac{1}{2} \sum_n \frac{1}{\sigma_{t_{n+1}}^2}(\nu_{t_n}^2(\hat{Y}_{t_{n+1}}) - \psi_{t_n}^2(\hat{Y}_{t_{n+1}}))\Delta t = \log R^N(\hat{Y}_{[\tau,\tau']}). \tag{106}$$

$\square$

**Proposition G.4.** *(RNC weights non-asymptotic error) We can bound the error between exact discrete-time SMC weights (using exact likelihoods) and our RNC estimator,*

$$||\log w^{\mathrm{exact}}(\hat{Y}_{[\tau,\tau']}) - \log w^{\mathrm{RNC}}(\hat{Y}_{[\tau,\tau']})||_{L^2} \leq \mathcal{O}(\Delta t^{1/2}), \tag{107}$$

*where*

$$w^{\mathrm{exact}}(\hat{Y}_{[\tau,\tau']}) = \frac{q_\tau(\hat{Y}_\tau)}{q_{\tau'}(\hat{Y}_{\tau'})} \frac{\prod_{n=1}^{N-1} p_{n+1|n}^b(\hat{Y}_{t_{n+1}}|\hat{Y}_{t_n})}{\prod_{n=1}^{N-1} p_{n|n+1}^a(\hat{Y}_{t_n}|\hat{Y}_{t_{n+1}})} \tag{108}$$

*and $w^{\mathrm{RNC}}(\hat{Y}_{[\tau,\tau']})$ is the weight estimated by $R^N(\hat{Y}_{[\tau,\tau']})$ (assuming a perfect score).*

*Proof.* Given the Lip assumption on $\log p_\tau$ we have:

$$||\log p_\tau(\hat{Y}_\tau)/p_{\tau'}(\hat{Y}_{\tau'}) - \log p_\tau(Y_\tau)/p_{\tau'}(Y_{\tau'})||_{L^2} \leq P||\hat{Y}_{\tau'} - Y_{\tau'}||_{L^2} \tag{109}$$

Now combining with Proposition 3.1 via triangle inequality we have that

$$||\log p_\tau(\hat{Y}_\tau)/p_{\tau'}(\hat{Y}_{\tau'}) - \log R^N(\hat{Y}_{[\tau,\tau']})^{-1}||_{L^2}$$
$$\leq ||\log p_\tau(Y_\tau)/p_{\tau'}(Y_{\tau'}) - \log R^N(\hat{Y}_{[\tau,\tau']})^{-1}||_{L^2} + ||\log p_\tau(\hat{Y}_\tau)/p_{\tau'}(\hat{Y}_{\tau'}) - \log p_\tau(Y_\tau)/p_{\tau'}(Y_{\tau'})||_{L^2}$$
$$= ||\log R_\mu^\nu(Y_{[\tau,\tau']}) - \log R^N(\hat{Y}_{[\tau,\tau']})||_{L^2} + P||\hat{Y}_{\tau'} - Y_{\tau'}||_{L^2},$$
$$\leq P'\sqrt{\Delta t} \tag{110}$$

Now moving onto the weights (for simplicity we assume $\frac{q_\tau(\hat{Y}_\tau)}{q_{\tau'}(\hat{Y}_{\tau'})} = g\left(\frac{p_\tau(\hat{Y}_\tau)}{p_{\tau'}(\hat{Y}_{\tau'})}\right)$ where $\log g$ is Lipchitz. Note that this assumption covers all cases but the reward-tilting one. However, for the

reward-tilting one, we can still get a similar bound assuming our chosen intermediate reward is bounded or Lipchitz.)

$$
\begin{aligned}
|| \log w^{\text{exact}}(\hat{Y}_{[\tau,\tau']}) - \log w^{\text{RNC}}(\hat{Y}_{[\tau,\tau']})||_{L^2} &\leq \left|\left| \log \frac{q_\tau(\hat{Y}_\tau)}{q_{\tau'}(\hat{Y}_{\tau'})} - \log g(R^N(\hat{Y}_{[\tau,\tau']})^{-1}) \right|\right|_{L^2} \\
&\leq L \left|\left| \log \frac{p_\tau(\hat{Y}_\tau)}{p_{\tau'}(\hat{Y}_{\tau'})} - \log R^N(\hat{Y}_{[\tau,\tau']})^{-1} \right|\right|_{L^2} \leq P'\sqrt{\Delta t}
\end{aligned}
$$

$\square$

### G.3.2 CONVERGENCE RATE FOR SMC WEIGHTS IN DISCRETE TIME AND APPROXIMATED SCORE

**Proposition G.5.** *(Discrete time and approximate score bound) Following (Lee et al., 2023; Chen et al., 2022) we assume:*

$$
||\nabla \log p_\tau(\hat{Y}_\tau) - s_\tau^\theta(\hat{Y}_\tau)||_{L^2} \leq \epsilon_{\text{score}}, \tag{111}
$$

*then*

$$
|| \log w^{\text{exact}}(\hat{Y}_{[\tau,\tau']}) - \log w_\theta^{\text{RNC}}(\hat{Y}_{[\tau,\tau']})||_{L^2} \leq E\epsilon_{\text{score}} + P'\sqrt{\Delta t} \tag{112}
$$

*where*

$$
w^{\text{exact}}(\hat{Y}_{[\tau,\tau']}) = \frac{q_\tau(\hat{Y}_\tau)}{q_{\tau'}(\hat{Y}_{\tau'})} \frac{\prod_{n=1}^{N-1} p_{n+1|n}^b(\hat{Y}_{t_{n+1}}|\hat{Y}_{t_n})}{\prod_{n=1}^{N-1} p_{n|n+1}^a(\hat{Y}_{t_n}|\hat{Y}_{t_{n+1}})} \tag{113}
$$

*and $w_\theta^{\text{RNC}}(\hat{Y}_{[\tau,\tau']})$ is the discrete time weight estimated using $R^N(\hat{Y}_{[\tau,\tau']})$ with the learned score $s_\tau^\theta$.*

*Proof.* To prove this conclusion, we separate the error contributed by discretisation and by imperfect score first, and then bound them separately:

$$
|| \log w_\theta^{\text{RNC}}(\hat{Y}_{[\tau,\tau']}) - \log w^{\text{exact}}(\hat{Y}_{[\tau,\tau']})||_{L^2} \tag{114}
$$

$$
= || \log R_\theta^N(\hat{Y}_{[\tau,\tau']})^{-1} - \log p_\tau(\hat{Y}_\tau)/p_{\tau'}(\hat{Y}_{\tau'})||_{L^2} \tag{115}
$$

$$
\leq || \log R_\theta^N(\hat{Y}_{[\tau,\tau']}) - \log R^N(\hat{Y}_{[\tau,\tau']})||_{L^2} + || \log p_\tau(\hat{Y}_\tau)/p_{\tau'}(\hat{Y}_{\tau'}) - \log R^N(\hat{Y}_{[\tau,\tau']})^{-1}||_{L^2} \tag{116}
$$

$$
= || \log R_\theta^N(\hat{Y}_{[\tau,\tau']}) - \log R^N(\hat{Y}_{[\tau,\tau']})||_{L^2} + P'\sqrt{\Delta t} \tag{117}
$$

Where the last line follows from Proposition G.4.

Following the time embedding of Remark G.3 and Lemma G.2 we have

$$
\log R^N(\bar{Y}_{[\tau,\tau']}) = A^N + B^N + \Delta_\theta^N + C^N \tag{118}
$$

$$
= \log \frac{\pi_\tau(\bar{Y}_\tau)}{\pi_{\tau'}(\bar{Y}_{\tau'})} + \int_\tau^{\tau'} \frac{1}{\bar{\sigma}_s}(\bar{\nu}_s - \bar{\psi}_s)(\bar{Y}_s)\overleftarrow{dW}_s \tag{119}
$$

$$
+ \int_\tau^{\tau'} \frac{1}{\bar{\sigma}_s^2}(\bar{\nu}_s(\bar{Y}_s) - \bar{\psi}_s(\bar{Y}_s)) \cdot (\bar{\nu}_s^\theta(\bar{Y}_s) - \bar{\nu}_s(\bar{Y}_s))ds - \frac{1}{2}\int_\tau^{\tau'} \frac{1}{\bar{\sigma}_s^2}||\bar{\nu}_s(\bar{Y}_s) - \bar{\psi}_s(\bar{Y}_s)||^2 ds \tag{120}
$$

and

$$
\log R_\theta^N(\bar{Y}_{[\tau,\tau']}) = A^N + B_\theta^N + C_\theta^N \tag{121}
$$

$$
= \log \frac{\pi_\tau(\bar{Y}_\tau)}{\pi_{\tau'}(\bar{Y}_{\tau'})} + \int_\tau^{\tau'} \frac{1}{\bar{\sigma}_s}(\bar{\nu}_s^\theta - \bar{\psi}_s)(\bar{Y}_s)\overleftarrow{dW}_s - \frac{1}{2}\int_\tau^{\tau'} \frac{1}{\bar{\sigma}_s^2}||\bar{\nu}_s^\theta(\bar{Y}_s) - \bar{\psi}_s(\bar{Y}_s)||^2 ds \tag{122}
$$

Therefore,

$$
|| \log R_\theta^N(\hat{Y}_{[\tau,\tau']}) - \log R^N(\hat{Y}_{[\tau,\tau']})||_{L^2} \leq ||B_\theta^N - B^N||_{L^2} + ||C_\theta^N - C^N||_{L^2} + ||\Delta_\theta^N||_{L^2}. \tag{123}
$$

Let's focus on the first term first:

$$\|B_\theta^N - B^N\|_{L^2} = \left\|\int_\tau^{\tau'} \frac{1}{\bar{\sigma}_s}(\bar{\nu}_s^\theta - \bar{\nu}_s)(\bar{Y}_s)\overleftarrow{\mathrm{d}W}_s\right\|_{L^2} \tag{124}$$

$$= \mathbb{E}\left[\int_\tau^{\tau'} \|\bar{\nu}_s^\theta - \bar{\nu}_s\|^2 \mathrm{d}s\right]^{1/2} \tag{125}$$

$$\leq \epsilon_{\mathrm{score}} \tag{126}$$

For the second term, let us use the shorthands $\bar{f}_s^\theta = \bar{\nu}_s^\theta - \bar{\psi}_s$ and $\bar{f} = \bar{\nu}_s - \bar{\psi}_s$ then

$$\|C_\theta^N - C^N\|_{L^2} = \left\|\frac{1}{2}\int_\tau^{\tau'}\frac{1}{\bar{\sigma}_s^2}(\|\bar{f}_s^\theta\|^2 - \|\bar{f}_s\|^2)\mathrm{d}s\right\| \tag{127}$$

$$= \left\|\frac{1}{2}\int_\tau^{\tau'}\frac{1}{\bar{\sigma}_s^2}(\bar{f}_s^\theta + \bar{f}_s)\cdot(\bar{f}_s^\theta - \bar{f}_s)\mathrm{d}s\right\| \tag{128}$$

$$\leq \mathbb{E}\left[\frac{1}{2}\int_\tau^{\tau'}\frac{1}{\bar{\sigma}_s^2}\|\bar{f}_s^\theta - \bar{f}_s\|^2\mathrm{d}s\right]^{1/2} \mathbb{E}\left[\frac{1}{2}\int_\tau^{\tau'}\frac{1}{\bar{\sigma}_s^2}\|\bar{f}_s^\theta + \bar{f}_s\|^2\mathrm{d}s\right]^{1/2} \tag{129}$$

$$\leq F\epsilon_{\mathrm{score}} \tag{130}$$

Where $\mathbb{E}\left[\frac{1}{2}\int_\tau^{\tau'}\frac{1}{\bar{\sigma}_s^2}\|\bar{f}_s^\theta + \bar{f}_s\|^2\mathrm{d}s\right]^{1/2} \leq F$ is bounded from Novikov's condition (which we assume all throughout) + Jensen's inequality.

Finally, use the exact same arguments as Equations 128-130 we can see that $\|\Delta_\theta^N\| \leq D'\epsilon_{\mathrm{score}}$. Therefore, in summary, we have $\|\log R_\theta^N(\hat{Y}_{[\tau,\tau']}) - \log R^N(\hat{Y}_{[\tau,\tau']})\|_{L^2} \leq E\epsilon_{\mathrm{score}}$. □

# H    PROOFS

## H.1    CORRECTNESS OF SMC WEIGHTS IN EQ. (16)

Eq. (16) is correct SMC weight for marginal $q_\tau$, as for a measurable function $h$ on $X_\tau$:

$$\mathbb{E}_{X_{[\tau,\tau']}\sim\overleftarrow{\mathbb{Q}}_{[\tau,\tau']}^a}\left[w_{[\tau,\tau']}(X_{[\tau,\tau']})h(X_\tau)\right] = \mathbb{E}_{X_{[\tau,\tau']}\sim\overrightarrow{\mathbb{Q}}_{[\tau,\tau']}^b}\left[h(X_\tau)\right] = \mathbb{E}_{X_\tau\sim q_\tau}[h(X_\tau)] \tag{131}$$

Running SMC with this weight in Eq. (16) has the following convergence guarantee:

**Proposition H.1.** *Let $\{X_0^{(m)}, w^{(w)}(X_{[0,\tau_0]})\}_{m=1}^M$ be the particles and their (unnormalised) weights returned at the last iteration (denoise from $\tau_0 \to 0$) of the sequential Monte Carlo algorithm with $M$ particles described in Section 2.1.1. If the weighting function $w_{[0,\tau_0]}$ is positive and $\mathbb{E}_{\overleftarrow{\mathbb{Q}}^a}[w_{[0,\tau_0]}^2|X_{\tau_0}]$ is bounded, then for a bounded, $q_0$-measurable function $h$, we have*

$$\mathbb{E}\left[\left\|\int h(X_0)q_0(X_0)\mathrm{d}X_0 - \sum_{m=1}^M \frac{w^{(m)}}{\sum_{j=1}^M w^{(j)}}h(X_0^{(m)})\right\|^2\right] \leq C'M^{-1} \tag{132}$$

*If $\mathbb{E}_{\overleftarrow{\mathbb{Q}}^a}[w_{[0,\tau_0]}^4|X_{\tau_0}]$ is bounded, we have*

$$\lim_{M\to\infty}\frac{w^{(m)}}{\sum_{j=1}^M w^{(j)}}h(X_0^{(m)}) \stackrel{a.s.}{=} \int h(X_0)q_0(X_0)\mathrm{d}X_0. \tag{133}$$

*Proof.* Eq. (132) is a direct Corollary of (Mbalawata & Särkkä, 2016, Theorem 3.4). To apply Theorem 3.4 of Mbalawata & Särkkä (2016), we need to verify Assumption 3.1-3.3. Assumption 3.1 is satisfied as we define the marginal $q_t$ to have bounded density. Assumption 3.2 is satisfied as our scheme ensures their Eq. (7) to be 0. Assumption 3.3 is satisfied according to our assumption. Similarly, Eq. (133) is a direct Corollary of (Mbalawata & Särkkä, 2016, Theorem 4.7). □

**Discussion on Assumptions**: For convergence in $L^2$ we require the following 2nd Order Novikov like assumptions to be satisfied

$$\mathbb{E}\left[\exp\left(\int_\tau^{\tau'} ||u^\pm(X_t)||^2 \mathrm{d}t\right)\Big| X_{\tau'}\right] < C \tag{134}$$

here $u^\pm$ represents all drift terms we use in our algorithm ($a_t$, $b_t$, $\psi$, $\phi$, $\mu$, $\nu$, etc.). For the a.s. convergence, we additionally require $\mathbb{E}\left[\exp\left(2\int_\tau^{\tau'} ||u^\pm(X_t)||^2 \mathrm{d}t\right)\Big| X_{\tau'}\right] < C$. Note that, for applying Girsanov's Theorem, we already require the following assumption:

$$\mathbb{E}\left[\exp\left(\frac{1}{2}\int_\tau^{\tau'} ||u^\pm(X_t)||^2 \mathrm{d}t\right)\Big| X_{\tau'}\right] < C. \tag{135}$$

We note that assuming $\mathbb{E}\left[\exp\left(\int_\tau^{\tau'} ||u^\pm(X_t)||^2 \mathrm{d}t\right)\Big| X_\tau\right]$ $<$ $C$ or $\mathbb{E}\left[\exp\left(\int_\tau^{\tau'} 2||u^\pm(X_t)||^2 \mathrm{d}t\right)\Big| X_\tau\right] < C$ implies the first-order Novikov condition in Eq. (135), and hence Proposition H.1 requires a stronger assumption than the standard Novikov required for Girsanov theorem.

## H.2 IS PERSPECTIVES FOR RNE, CONNECTIONS TO HUANG ET AL. (2021)

> **Proposition C.1.** let $p_t(x_t)$ be the marginal density of $X_t = x_t$ satisfying the backwards SDE:
>
> $$\mathrm{d}X_t = \nu_t(X_t)\mathrm{d}t + \epsilon_t \overleftarrow{\mathrm{d}W_t}, \ \ X_1 \sim p_1. \tag{136}$$
>
> Consider a forward process $Y_t$ with drift $u_t$, and define $R_u^\nu$ as Eq. (10), we have
>
> $$p_t(x_t) = \mathbb{E}\left[p_1(Y_1)R_u^\nu(Y_{[t:1]})\big|Y_t = x_t\right], \tag{137}$$
>
> where the expectation is taken over the forward process within the time horizon $[t, 1]$ conditional on $Y_t = x_t$. When discretised with $t = t_1 < t_2 < \cdots < t_N = 1$:
>
> $$p_{t_1}(x_{t_1}) \approx \mathbb{E}\left[p_{t_N}(Y_{t_N})\frac{\prod_{n=1}^{N-1} p_{n|n+1}^\nu(Y_{t_n}|Y_{t_{n+1}})}{\prod_{n=1}^{N} p_{n+1|n}^u(Y_{t_{n+1}}|Y_{t_n})}\Bigg|Y_{t_1} = x_{t_1}\right]. \tag{138}$$

*Proof.* Let's define the path measure of Eq. (136) as $\overleftarrow{\mathbb{P}}$. We also denote the path measure of the forward process $Y_t$ with drift $u_t$, starting from some $q_t$, as $\overrightarrow{\mathbb{Q}}$. Recall the definition of $R_u^\nu$ in Eq. (10), we can see that

$$R_u^\nu(Y_{[t,1]}) = \frac{\mathrm{d}\overleftarrow{\mathbb{P}}}{\mathrm{d}\overrightarrow{\mathbb{Q}}}(Y_{[t,1]})\frac{q_t(Y_t)}{p_1(Y_1)} \tag{139}$$

Hence we have

$$\mathbb{E}\left[p_1(Y_1)R_u^\nu(Y_{[t:1]})\big|Y_t = x_t\right] = \mathbb{E}\left[p_1(Y_1)\frac{\mathrm{d}\overleftarrow{\mathbb{P}}}{\mathrm{d}\overrightarrow{\mathbb{Q}}}(Y_{[t,1]})\frac{q_t(Y_t)}{p_1(Y_1)}\Bigg|Y_t = x_t\right] \tag{140}$$

$$= \mathbb{E}\left[\frac{1}{1/q_t(Y_t)}\frac{\mathrm{d}\overleftarrow{\mathbb{P}}}{\mathrm{d}\overrightarrow{\mathbb{Q}}}(Y_{[t,1]})\Bigg|Y_t = x_t\right] \tag{141}$$

$$= \mathbb{E}\left[\frac{p_t(Y_t)}{q_t(Y_t)/q_t(Y_t)}\frac{\mathrm{d}\overleftarrow{\mathbb{P}}_{(t,1]|t}}{\mathrm{d}\overrightarrow{\mathbb{Q}}_{(t,1]|t}}(Y_{(t,1]})\Bigg|Y_t = x_t\right] \tag{142}$$

$$= p_t(x_t)\int \mathrm{d}\overleftarrow{\mathbb{P}}_{(t,1]|t} \tag{143}$$

$$= p_t(x_t). \tag{144}$$

Here we use $\overleftarrow{\mathbb{P}}_{(t,1]|t}$ to represent the path measure $\overleftarrow{\mathbb{P}}$ conditional on values $X_t = x_t$ at $t$. $\qquad\square$

**Corollary C.2.** The relation in Eq. (30) can be simplified to

$$p_t(x_t) = \mathbb{E}_{Z_{[t,1]} \sim \overrightarrow{\mathbb{P}}^\nu} \left[ p_1(Z_1) \exp \left( \int_t^1 \nabla \cdot \nu_{t'}(Z_{t'}) \mathrm{d}t' \right) \Big| Z_t = x_t \right], \tag{145}$$

*Proof.* Using the conversion formula (Vargas et al., 2023b, Remark 3) with Eq. (12), we have that

$$\log R_u^\nu(Y_{[t:1]}) = \int \frac{1}{\epsilon_{t'}^2} \nu_{t'} \cdot \overleftarrow{\mathrm{d}Y_{t'}} - \int \frac{1}{\epsilon_{t'}^2} u_{t'} \cdot \overrightarrow{\mathrm{d}Y_{t'}} + \frac{1}{2} \int \frac{1}{\epsilon_{t'}^2} (||u_{t'}||^2 - ||\nu_{t'}||^2) \mathrm{d}t' \tag{146}$$

$$= \int \frac{1}{\epsilon_{t'}^2} \nu_{t'} \cdot \overrightarrow{\mathrm{d}Y_{t'}} - \int \frac{1}{\epsilon_{t'}^2} u_{t'} \cdot \overrightarrow{\mathrm{d}Y_{t'}} + \frac{1}{2} \int \frac{1}{\epsilon_{t'}^2} (||u_{t'}||^2 - ||\nu_{t'}||^2 + \epsilon_{t'}^2 \nabla \cdot \nu_{t'}) \mathrm{d}t' \tag{147}$$

which for $\overrightarrow{\mathrm{d}Y}_t = u_t \mathrm{d}t + \epsilon_t \overrightarrow{\mathrm{d}W}_t$ can be re-expressed as:

$$\log R_u^\nu(Y_{[t:1]}) = \int \frac{1}{\epsilon_{t'}} (\nu_{t'} - u_{t'}) \cdot \overrightarrow{\mathrm{d}W} - \int \frac{1}{\epsilon_{t'}^2} (\frac{1}{2} ||u_{t'} - \nu_{t'}||^2 + \epsilon_{t'}^2 \nabla \cdot \nu_{t'}) \mathrm{d}t'. \tag{148}$$

Notice that, by Girsanov theorem, we have $\int \frac{1}{\epsilon} (\nu - u) \cdot \overrightarrow{\mathrm{d}W} - \int \frac{1}{\epsilon^2} (\frac{1}{2} ||u - \nu||^2) \mathrm{d}t' = \log \frac{\mathrm{d}\overrightarrow{\mathbb{P}}^\nu}{\mathrm{d}\overrightarrow{\mathbb{Q}}^u}$ thus

$$p_t(x_t) = \mathbb{E}_{Z_{[t,1]} \sim \overrightarrow{\mathbb{P}}^\nu} \left[ p_1(Z_1) \exp \left( \int_t^1 \nabla \cdot \nu_{t'}(Z_{t'}) \mathrm{d}t' \right) \Big| Z_t = x_t \right], \tag{149}$$

$\square$

**Corollary H.2.** *The RNE-IS relation can also be simplified to*

$$p_t(x_t) = \mathbb{E}_{\overrightarrow{\mathbb{P}}^u} \left[ p_1(Y_1) \exp \left( \int_t^1 \frac{1}{\epsilon_{t'}^2} (\nu_{t'} - u_{t'}) \cdot \overrightarrow{\mathrm{d}W} - \int_t^1 (\frac{1}{2\epsilon_{t'}^2} ||u_{t'} - \nu_{t'}||^2 + \nabla \cdot \nu_{t'}) \mathrm{d}t' \right) \Big| Y_t = x_t \right] \tag{150}$$

$$\geq \exp \mathbb{E}_{\overrightarrow{\mathbb{P}}^u} \left[ \log p_1(Y_1) + \left( - \int_t^1 (\frac{1}{2\epsilon_{t'}^2} ||u_{t'} - \nu_{t'}||^2 + \nabla \cdot \nu_{t'}) \mathrm{d}t' \right) \Big| Y_t = x_t \right] \tag{151}$$

*which recovers the density estimator of (Premkumar, 2024, Eq. 3.5) and the Variational objective of (Huang et al., 2021, Eq 16). Additionally, we note*

$$\mathbb{E}_{\overrightarrow{\mathbb{P}}^u} \left[ \int_t^1 \nabla \cdot \nu_{t'} \mathrm{d}t' \Big| Y_t = x_t \right] = \int_t^1 -\mathbb{E}_{p_{t'}^u(\cdot|Y_t = x_t)} [\nabla \log p_{t'}^u(\cdot|Y_t = x_t) \cdot \nu_{t'}] \mathrm{d}t' \tag{152}$$

*and we will recover the divergence-free density estimator of (Premkumar, 2024, Eq. 3.8).*

*Proof.* Eq. (152) follows Eq. (148) with Jensen's inequality. For the divergence-free estimator:

$$\mathbb{E}_{\overrightarrow{\mathbb{P}}^u} \left[ \int_t^1 \nabla \cdot \nu_{t'} \mathrm{d}t' \Big| Y_t = x_t \right] = \int_t^1 \mathbb{E}_{p_{t'}^u(\cdot|Y_t = x_t)} [\nabla \cdot \nu_{t'}] \mathrm{d}t' \tag{153}$$

$$= \int_t^1 \int p_{t'}^u(y|Y_t = x_t) \nabla \cdot \nu_{t'}(y) \mathrm{d}y \mathrm{d}t' \tag{154}$$

$$= \int_t^1 - \int \nabla p_s^u(y|Y_t = x_t) \cdot \nu_{t'} \mathrm{d}y \mathrm{d}t' \tag{155}$$

$$= \int_t^1 - \int p_{t'}^u(y|Y_t = x_t) \nabla \log p_{t'}^u(y|Y_t = x_t) \cdot \nu_{t'}(y) \mathrm{d}y \mathrm{d}t' \tag{156}$$

$$= \int_t^1 -\mathbb{E}_{p_{t'}^u(\cdot|Y_t = x_t)} [\nabla \log p_{t'}^u(\cdot|Y_t = x_t) \cdot \nu_{t'}] \mathrm{d}t' \tag{157}$$

$\square$

Additionally, if we access the unnormalised version of $p_t$ and $p_1$, taking the expectation over $p_t$, we will obtain Jarzynski equality (Jarzynski, 1997) and Escorted Jarzynski equality (Vaikuntanathan & Jarzynski, 2008), which can be used to estimate the free energy difference between two states with learned transports (He et al., 2025a).

## H.3 Equivalence to Itô density estimator

The RNDE in Eq. (27) is equivalent to Itô density estimator in continuous time:

$$\mathrm{d}\log p_t(Y_t) = -\Big(\nabla \cdot f_t(Y_t) + \nabla \log p_t(Y_t) \cdot (f_t(Y_t) - \frac{\sigma_t^2}{2}\nabla \log p_t(Y_t))\Big)\mathrm{d}t + \nabla \log p_t(Y_t) \cdot \overleftarrow{\mathrm{d}Y_t}. \quad (158)$$

*Proof.* With expression of $R$ in Eq. (12) and the conversion rule (Vargas et al., 2023b), we have

$$\log p_t(Y_t) = \log p_1(Y_1) + \log R_\mu^\nu(Y_{[t,1]}) \quad (159)$$

$$= \log p_1(Y_1) - \int_t^1 \frac{1}{\epsilon_{t'}^2}\mu_{t'} \cdot \overrightarrow{\mathrm{d}Y_{t'}} + \frac{1}{2}\int_t^1 \frac{1}{\epsilon_{t'}^2}\|\mu_{t'}\|^2\mathrm{d}t' + \int_t^1 \frac{1}{\epsilon_{t'}^2}\nu_{t'} \cdot \overleftarrow{\mathrm{d}Y_{t'}} - \frac{1}{2}\int_t^1 \frac{1}{\epsilon_{t'}^2}\|\nu_{t'}\|^2\mathrm{d}t' \quad (160)$$

$$= \log p_1(Y_1) - \int_t^1 \frac{1}{\epsilon_{t'}^2}\mu_{t'} \cdot \overleftarrow{\mathrm{d}Y_{t'}} + \frac{1}{2}\int_t^1 \frac{1}{\epsilon_{t'}^2}\|\mu_{t'}\|^2\mathrm{d}t'$$
$$+ \int_t^1 \frac{1}{\epsilon_{t'}^2}\nu_{t'} \cdot \overleftarrow{\mathrm{d}Y_{t'}} - \frac{1}{2}\int_t^1 \frac{1}{\epsilon_{t'}^2}\|\nu_{t'}\|^2\mathrm{d}t' + \int_t^1 \nabla \cdot \mu_{t'}\mathrm{d}t' \quad (161)$$

For a pretrained diffusion model considered in (Karczewski et al., 2024; Skreta et al., 2024), we set $\epsilon_t = \sigma_t$, $\mu_t = f_t$, and $\nu_t = f_t - \sigma_t^2\nabla \log p_t$. Therefore,

$$\log p_t(Y_t) = \log p_1(Y_1) - \int_t^1 \nabla \log p_{t'}(Y_{t'}) \cdot \overleftarrow{\mathrm{d}Y_{t'}}$$
$$+ \int_t^1 \nabla \log p_{t'}(Y_{t'}) \cdot \Big(f_{t'}(Y_{t'}) - \frac{\sigma_{t'}^2}{2}\nabla \log p_{t'}(Y_{t'})\Big)\mathrm{d}t' + \int_t^1 \nabla \cdot f_{t'}(Y_{t'})\mathrm{d}t' \quad (162)$$

$\square$

## H.4 Connections to Keynman-Kac Corrector (Skreta et al., 2025)

As stated in Eq. (37), for some special cases, our proposed RNC is theoretically equivalent to FKC. In this section, we will prove these connections. The proof directly applies Eq. (12) and the conversion formula (Vargas et al., 2023b) to the importance weights in Eqs. (17) to (19).

**Before showing the equivalence between FKC and RNC in detail, we need to clarify one important concept.** In our importance weight calculation, as shown in Eq. (16), we have made a few important details implicit for the sake of brevity. In particular, notice the following more explicit notation for the RND:

$$w_{[\tau,\tau']}(X_{[\tau,\tau']}) = \frac{\mathrm{d}\overrightarrow{\mathbb{Q}}^b}{\mathrm{d}\overleftarrow{\mathbb{Q}}^a}(X_{[\tau,\tau']}) = \frac{\mathrm{d}\overrightarrow{\mathbb{Q}}_{[\tau,\tau']}^{b,q_\tau}}{\mathrm{d}\overleftarrow{\mathbb{Q}}_{[\tau,\tau']}^{a,q_{\tau'}}}(X_{[\tau,\tau']}) \quad (163)$$

where $\overrightarrow{\mathbb{Q}}_{[\tau,\tau']}^{b,q_\tau}$ is used to denote that the target process has as its initial distribution $q_\tau$ and moves forward in time from $\tau$ to $\tau'$, e.g.

$$\mathrm{d}Y_t = b_t(Y_t)\,\mathrm{d}t + \epsilon_t\,\overrightarrow{\mathrm{d}W_t}, \quad Y_\tau \sim q_\tau \text{ (e.g } q_\tau = p_\tau^\beta), \quad t \in [\tau, \tau'] \quad (164)$$

and similarly $\overleftarrow{\mathbb{Q}}_{[\tau,\tau']}^{a,q_{\tau'}}$ is used to denote that the proposal process has as its initial distribution $q_{\tau'}$ and moves backward in time from $\tau'$ to $\tau$. **It is important to notice, when simulating the target process $\overrightarrow{\mathbb{Q}}_{[\tau,\tau']}^{b,q_\tau}$ from $\tau$ to $\tau'$, it will not necessarily result in samples following $q_{\tau'}$. In other word, assuming two adjacent steps $[s,\tau]$ and $[\tau,\tau']$, then $\overrightarrow{\mathbb{Q}}_{[\tau,\tau']}^{b,q_\tau}$ is not the same as continuing $\overrightarrow{\mathbb{Q}}_{[s,\tau]}^{b,q_s}$**

to $\tau'$. Note this clarification needs highlighting as we abuse $\mathrm{d}\overrightarrow{\mathbb{Q}}^b/\mathrm{d}\overleftarrow{\mathbb{Q}}^a(X_{[t,\tau']})$ to denote RNDs at different time intervals without providing a time index on $\mathbb{Q}$ (only on $X$). **In fact, we use it to represent a sequence of path measures indexed by time as opposed to the path measure of the same SDE simulated within different time horizons.**

In what follows, we will demonstrate which choices of $a_t$ and $b_t$ recover the FKC weights from the RNC weights, thus reinterpreting these weights as the RND between two SDEs.

The proof follows the same principle: (1) we first express the importance weight of RNC using a continuous formulation defined by Eq. (12), and (2) apply the conversion formula (Vargas et al., 2023b) to convert forward and backward Itô's integrals in the same direction. (3) If there are additional terms, such as a reward, we will apply Itô's Lemma to further simplify it.

### H.4.1 ANNEAL FKC

> **Proposition H.3.** *Anneal-FKC states that, for a perfect diffusion model (as defined in Eqs. (1) and (2) or Eqs. (3) and (4)), one can implement the following backward sampling SDE:*
>
> $$\mathrm{d}X_t = \left(f_t(X_t) - \eta\sigma_t^2 \nabla \log p_t(X_t)\right)\mathrm{d}t + \zeta\sigma_t\overleftarrow{\mathrm{d}W_t}, \qquad (165)$$
>
> *and the importance weight for Sequential Monte Carlo satisfies the backward ODE:*
>
> $$\mathrm{d}\log w_t = -(\beta - 1)\left(\nabla \cdot f_t(X_t) + \frac{\sigma_t^2}{2}\beta\|\nabla \log p_t(X_t)\|^2\right)\mathrm{d}t, \qquad (166)$$
>
> *where*
>
> $$\eta = \beta + (1 - \beta)a, \quad \zeta = \sqrt{1 + (1 - \beta)2a/\beta}, \quad \forall a \in [0, 1/2]. \qquad (167)$$
>
> *This is equivalent to our proposed RNC (Eq. (17)), when*
>
> $$a_t = f_t - \eta\sigma_t^2 \nabla \log p_t, \quad b_t = f_t - (\eta\sigma_t^2 - \beta\epsilon_t^2)\nabla \log p_t, \quad \epsilon_t = \zeta\sigma_t, \qquad (168)$$

*Proof.* We consider the SI characterisation of the diffusion model, as defined in Eqs. (3) and (4). Therefore, we have

$$\mu_t = \mathrm{v}_t + \frac{\epsilon_t^2}{2}\nabla \log p_t = f_t + \frac{\epsilon_t^2 - \sigma_t^2}{2}\nabla \log p_t \qquad (169)$$

$$\nu_t = \mathrm{v}_t - \frac{\epsilon_t^2}{2}\nabla \log p_t = f_t - \frac{\epsilon_t^2 + \sigma_t^2}{2}\nabla \log p_t \qquad (170)$$

Therefore, with the continuous-time expression of $R$ in Eq. (12) and the conversion formula (Vargas et al., 2023b):

$$\mathrm{d}\log R_\mu^\nu = -\frac{\nu_t - \mu_t}{\epsilon_t^2}\cdot\overleftarrow{\mathrm{d}X_t} - \nabla\cdot\mu_t\mathrm{d}t + \frac{1}{2\epsilon_t^2}(\nu_t - \mu_t)(\nu_t + \mu_t)\mathrm{d}t \qquad (171)$$

$$= \nabla\log p_t\cdot\overleftarrow{\mathrm{d}X_t} - \nabla\cdot\left(f_t + \frac{\epsilon_t^2 - \sigma_t^2}{2}\nabla\log p_t\right)\mathrm{d}t - \nabla\log p_t\cdot(f_t - \frac{\sigma_t^2}{2}\nabla\log p_t)\mathrm{d}t \qquad (172)$$

Similarly, we have

$$\mathrm{d}\log R_b^a = -\frac{a_t - b_t}{\epsilon_t^2}\cdot\overleftarrow{\mathrm{d}X_t} - \nabla\cdot b_t\mathrm{d}t + \frac{1}{2\epsilon_t^2}(a_t - b_t)(a_t + b_t)\mathrm{d}t \qquad (173)$$

$$= \beta\nabla\log p_t\cdot\overleftarrow{\mathrm{d}X_t} - \nabla\cdot\left(f_t - (\eta\sigma_t^2 - \beta\epsilon_t^2)\nabla\log p_t\right)\mathrm{d}t$$

$$- \beta\nabla\log p_t\cdot\left(f_t - (\eta\sigma_t^2 - \frac{\beta\epsilon_t^2}{2})\nabla\log p_t\right)\mathrm{d}t \qquad (174)$$

Then, according to the RNC weight given by Eq. (17), we have

$$\mathrm{d}\log w_t = \beta \mathrm{d}\log R_\mu^\nu - \mathrm{d}\log R_b^a \tag{175}$$

$$= -\nabla \cdot \underbrace{\left(\beta f_t + \beta \frac{\epsilon_t^2 - \sigma_t^2}{2}\nabla \log p_t - f_t + (\eta\sigma_t^2 - \beta\epsilon_t^2)\nabla \log p_t\right)}_{(1)} \mathrm{d}t$$

$$- \beta \nabla \log p_t \cdot \underbrace{\left(f_t - \frac{\sigma_t^2}{2}\nabla \log p_t - f_t + (\eta\sigma_t^2 - \frac{\beta\epsilon_t^2}{2})\nabla \log p_t\right)}_{(2)} \mathrm{d}t \tag{176}$$

To compare with FKC, we set:

$$\epsilon_t = \zeta\sigma_t, \quad \eta = \beta + (1-\beta)a, \quad \zeta = \sqrt{1 + (1-\beta)2a/\beta} \tag{177}$$

We first look at (1):

$$\beta f_t + \beta\frac{\epsilon_t^2 - \sigma_t^2}{2}\nabla \log p_t - f_t + (\eta\sigma_t^2 - \beta\epsilon_t^2)\nabla \log p_t \tag{178}$$

$$= (\beta - 1)f_t - \frac{\beta\zeta^2\sigma_t^2}{2}\nabla \log p_t + \left(\eta\sigma_t^2 - \frac{\beta\sigma_t^2}{2}\right)\nabla \log p_t \tag{179}$$

$$= (\beta - 1)f_t - \frac{\beta(1 + (1-\beta)2a/\beta)\sigma_t^2}{2}\nabla \log p_t + \left((\beta + (1-\beta)a)\sigma_t^2 - \frac{\beta\sigma_t^2}{2}\right)\nabla \log p_t \tag{180}$$

$$= (\beta - 1)f_t \tag{181}$$

We then look at term (2):

$$f_t - \frac{\sigma_t^2}{2}\nabla \log p_t - f_t + (\eta\sigma_t^2 - \frac{\beta\epsilon_t^2}{2})\nabla \log p_t \tag{182}$$

$$= -\frac{\sigma_t^2}{2}\nabla \log p_t + (\eta\sigma_t^2 - \frac{\beta\zeta^2\sigma_t^2}{2})\nabla \log p_t \tag{183}$$

$$= -\frac{\sigma_t^2}{2}\nabla \log p_t + ((\beta + (1-\beta)a)\sigma_t^2 - \frac{(\beta + (1-\beta)2a)\sigma_t^2}{2})\nabla \log p_t \tag{184}$$

$$= (\beta - 1)\frac{\sigma_t^2}{2}\nabla \log p_t \tag{185}$$

Putting things together, we have

$$\mathrm{d}\log w_t = -(\beta - 1)\left(\frac{1}{2}\beta\sigma_t^2\|\nabla \log p_t\|^2 + \nabla \cdot f_t\right)\mathrm{d}t \tag{186}$$

which coincides with the expression by FKC. $\qquad\square$

### H.4.2 PRODUCT FKC

**Proposition H.4.** *Product-FKC states that, for two perfect diffusion models, one can implement the following backward sampling SDE:*

$$\mathrm{d}X_t = \left(f_t(X_t) - \eta\sigma_t^2\nabla \log p_t^{(1)}(X_t) - \eta\sigma_t^2\nabla \log p_t^{(2)}(X_t)\right)\mathrm{d}t + \zeta\sigma_t\overleftarrow{\mathrm{d}W_t}, \tag{187}$$

*and the importance weight for Sequential Monte Carlo satisfies the backward ODE:*

$$\mathrm{d}\log w_t = -\Big((2\beta - 1)\nabla \cdot f_t(X_t) + \sigma_t^2\beta\nabla \log p_t^{(1)}(X_t) \cdot \nabla \log p_t^{(2)}(X_t)$$

$$+ \frac{\sigma_t^2}{2}\beta(\beta - 1)\|\nabla \log p_t^{(1)}(X_t) + \nabla \log p_t^{(2)}(X_t)\|^2\Big)\mathrm{d}t, \tag{188}$$

*where*

$$\eta = \beta + (1-\beta)a, \quad \zeta = \sqrt{1 + (1-\beta)2a/\beta}, \quad \forall a \in [0, 1/2]. \tag{189}$$

> *This is equivalent to our proposed RNC (Eq. (19)), when*
>
> $$a_t = f_t - \eta\sigma_t^2 \left( \nabla \log p_t^{(1)} + \nabla \log p_t^{(2)} \right),$$
>
> $$b_t = f_t - \left( \eta\sigma_t^2 - \epsilon_t^2 \beta \right) \left( \nabla \log p_t^{(1)} + \nabla \log p_t^{(2)} \right), \tag{190}$$
>
> $$\epsilon_t = \zeta\sigma_t.$$

*Proof.* Similar to the anneal case, for $i \in \{1, 2\}$, we have

$$\mathrm{d}\log R_{\mu^{(i)}}^{\nu^{(i)}} = -\frac{\nu_t^{(i)} - \mu_t^{(i)}}{\epsilon_t^2} \cdot \overleftarrow{\mathrm{d}X_t} - \nabla \cdot \mu_t^{(i)}\mathrm{d}t + \frac{1}{2\epsilon_t^2}(\nu_t^{(i)} - \mu_t^{(i)})(\nu_t^{(i)} + \mu_t^{(i)})\mathrm{d}t \tag{191}$$

$$= \nabla \log p_t^{(i)} \cdot \overleftarrow{\mathrm{d}X_t} - \nabla \cdot \left( f_t + \frac{\epsilon_t^2 - \sigma_t^2}{2}\nabla \log p_t^{(i)} \right) \mathrm{d}t$$

$$- \nabla \log p_t^{(i)} \cdot (f_t - \frac{\sigma_t^2}{2}\nabla \log p_t^{(i)})\mathrm{d}t \tag{192}$$

and

$$\mathrm{d}\log R_b^a = -\frac{a_t - b_t}{\epsilon_t^2} \cdot \overleftarrow{\mathrm{d}X_t} - \nabla \cdot b_t\mathrm{d}t + \frac{1}{2\epsilon_t^2}(a_t - b_t)(a_t + b_t)\mathrm{d}t \tag{193}$$

$$= \beta(\nabla \log p_t^{(1)} + \nabla \log p_t^{(2)}) \cdot \overleftarrow{\mathrm{d}X_t} \tag{194}$$

$$- \nabla \cdot \left( f_t - (\eta\sigma_t^2 - \beta\epsilon_t^2)(\nabla \log p_t^{(1)} + \nabla \log p_t^{(2)}) \right) \mathrm{d}t$$

$$- \beta(\nabla \log p_t^{(1)} + \nabla \log p_t^{(2)}) \cdot \left( f_t - (\eta\sigma_t^2 - \frac{\beta\epsilon_t^2}{2})(\nabla \log p_t^{(1)} + \nabla \log p_t^{(2)}) \right) \mathrm{d}t \tag{195}$$

Then, according to the RNC weight given by Eq. (19), we have

$$\mathrm{d}\log w_t = \beta \sum_{i=\{1,2\}} \mathrm{d}\log R_{\mu^{(i)}}^{\nu^{(i)}} - \mathrm{d}\log R_b^a \tag{196}$$

$$= -\nabla \cdot \underbrace{\left( 2\beta f_t + \beta\frac{\epsilon_t^2 - \sigma_t^2}{2}(\nabla \log p_t^{(1)} + \nabla \log p_t^{(2)}) - f_t + (\eta\sigma_t^2 - \beta\epsilon_t^2)(\nabla \log p_t^{(1)} + \nabla \log p_t^{(2)}) \right)}_{(1)} \mathrm{d}t$$

$$- \beta\nabla \log p_t^{(1)} \cdot \underbrace{\left( f_t - \frac{\sigma_t^2}{2}\nabla \log p_t^{(1)} - f_t + (\eta\sigma_t^2 - \frac{\beta\epsilon_t^2}{2})(\nabla \log p_t^{(1)} + \nabla \log p_t^{(2)}) \right)}_{(2)} \mathrm{d}t$$

$$- \beta\nabla \log p_t^{(2)} \cdot \underbrace{\left( f_t - \frac{\sigma_t^2}{2}\nabla \log p_t^{(2)} - f_t + (\eta\sigma_t^2 - \frac{\beta\epsilon_t^2}{2})(\nabla \log p_t^{(1)} + \nabla \log p_t^{(2)}) \right)}_{(3)} \mathrm{d}t \tag{197}$$

First, let's look at term (1), same as Eq. (178), we obtain $(1) = (2\beta - 1)f_t$. We now turn to term (2):

$$f_t - \frac{\sigma_t^2}{2}\nabla \log p_t^{(1)} - f_t + (\eta\sigma_t^2 - \frac{\beta\epsilon_t^2}{2})(\nabla \log p_t^{(1)} + \nabla \log p_t^{(2)}) \tag{198}$$

$$= -\frac{\sigma_t^2}{2}\nabla \log p_t^{(1)} + (\eta\sigma_t^2 - \frac{\beta\epsilon_t^2}{2})(\nabla \log p_t^{(1)} + \nabla \log p_t^{(2)}) \tag{199}$$

$$= (\beta - 1)\frac{\sigma_t^2}{2}\nabla \log p_t^{(1)} + \beta\frac{\sigma_t^2}{2}\nabla \log p_t^{(2)} \tag{200}$$

Similarly, for term (3), we have

$$f_t - \frac{\sigma_t^2}{2}\nabla \log p_t^{(2)} - f_t + (\eta\sigma_t^2 - \frac{\beta\epsilon_t^2}{2})(\nabla \log p_t^{(1)} + \nabla \log p_t^{(2)}) \tag{201}$$

$$= (\beta - 1)\frac{\sigma_t^2}{2}\nabla \log p_t^{(2)} + \beta\frac{\sigma_t^2}{2}\nabla \log p_t^{(1)} \tag{202}$$

Therefore, putting them together, we have

$$d\log w_t = \beta \sum_{i=\{1,2\}} d\log R^{\nu^{(i)}}_{\mu^{(i)}} - d\log R^a_b \tag{203}$$

$$= -\nabla \cdot (2\beta - 1)f_t dt \tag{204}$$

$$- \beta\nabla\log p_t^{(1)} \cdot \left((\beta-1)\frac{\sigma_t^2}{2}\nabla\log p_t^{(1)} + \beta\frac{\sigma_t^2}{2}\nabla\log p_t^{(2)}\right)dt$$

$$- \beta\nabla\log p_t^{(2)} \cdot \left((\beta-1)\frac{\sigma_t^2}{2}\nabla\log p_t^{(2)} + \beta\frac{\sigma_t^2}{2}\nabla\log p_t^{(1)}\right)dt \tag{205}$$

$$= -\nabla \cdot (2\beta - 1)f_t dt \tag{206}$$

$$- \beta\nabla\log p_t^{(1)} \cdot \left((\beta-1)\frac{\sigma_t^2}{2}\nabla\log p_t^{(1)} + (\beta-1)\frac{\sigma_t^2}{2}\nabla\log p_t^{(2)}\right)dt$$

$$- \beta\nabla\log p_t^{(2)} \cdot \left((\beta-1)\frac{\sigma_t^2}{2}\nabla\log p_t^{(2)} + (\beta-1)\frac{\sigma_t^2}{2}\nabla\log p_t^{(1)}\right)dt$$

$$- \beta\sigma_t^2\nabla\log p_t^{(1)} \cdot \nabla\log p_t^{(2)} \tag{207}$$

$$= -\nabla \cdot (2\beta-1)f_t dt - \beta(\beta-1)\frac{\sigma_t^2}{2}\|\nabla\log p_t^{(1)} + \nabla\log p_t^{(2)}\|^2 dt$$

$$- \beta\sigma_t^2\nabla\log p_t^{(1)} \cdot \nabla\log p_t^{(2)} dt \tag{208}$$

$$\square$$

### H.4.3 CFG FKC

> **Proposition H.5.** *CFG-FKC states that, for two perfect diffusion models, one can implement the following backward sampling SDE:*
>
> $$dX_t = \left(f_t(X_t) - (1-\beta)\sigma_t^2\nabla\log p_t^{(1)}(X_t) - \beta\sigma_t^2\nabla\log p_t^{(2)}(X_t)\right)dt + \sigma_t\overleftarrow{dW_t}, \tag{209}$$
>
> *and the importance weight for Sequential Monte Carlo satisfies the backward ODE:*
>
> $$d\log w_t = -\frac{\sigma_t^2}{2}\beta(\beta-1)\|\nabla\log p_t^{(1)}(X_t) - \nabla\log p_t^{(2)}(X_t)\|^2 dt \tag{210}$$
>
> *This is equivalent to our proposed RNC (Eq. (19)), when*
>
> $$a_t = f_t - \sigma_t^2\left((1-\beta)\nabla\log p_t^{(1)} + \beta\nabla\log p_t^{(2)}\right), \quad b_t = f_t, \quad \epsilon_t = \sigma_t \tag{211}$$

*Proof.* We directly consider $\epsilon_t = \sigma_t$.

Similar to the anneal and product case, for $i \in \{1, 2\}$, we have

$$d\log R^{\nu^{(i)}}_{\mu^{(i)}} = -\frac{\nu_t^{(i)} - \mu_t^{(i)}}{\sigma_t^2} \cdot \overleftarrow{dX_t} - \nabla \cdot \mu_t^{(i)} dt + \frac{1}{2\sigma_t^2}(\nu_t^{(i)} - \mu_t^{(i)})(\nu_t^{(i)} + \mu_t^{(i)})dt \tag{212}$$

$$= \nabla\log p_t^{(i)} \cdot \overleftarrow{dX_t} - \nabla \cdot f_t dt - \nabla\log p_t^{(i)} \cdot (f_t - \frac{\sigma_t^2}{2}\nabla\log p_t^{(i)})dt \tag{213}$$

and

$$d\log R^a_b = -\frac{a_t - b_t}{\sigma_t^2} \cdot \overleftarrow{dX_t} - \nabla \cdot b_t dt + \frac{1}{2\sigma_t^2}(a_t - b_t)(a_t + b_t)dt \tag{214}$$

$$= ((1-\beta)\nabla\log p_t^{(1)} + \beta\nabla\log p_t^{(2)}) \cdot \overleftarrow{dX_t} - \nabla \cdot f_t dt$$

$$-((1-\beta)\nabla\log p_t^{(1)} + \beta\nabla\log p_t^{(2)}) \cdot \left(f_t - \frac{\sigma_t^2}{2}\left((1-\beta)\nabla\log p_t^{(1)} + \beta\nabla\log p_t^{(2)}\right)\right)dt \tag{215}$$

Then, according to the RNC weight given by Eq. (19), we have

$$\mathrm{d}\log w_t = (1-\beta)\mathrm{d}\log R^{\nu^{(1)}}_{\mu^{(1)}} + \beta\mathrm{d}\log R^{\nu^{(2)}}_{\mu^{(2)}} - \mathrm{d}\log R^a_b \tag{216}$$

$$= -(1-\beta)\nabla\log p_t^{(1)} \cdot \left( f_t - \frac{\sigma_t^2}{2}\nabla\log p_t^{(1)} - f_t + \frac{\sigma_t^2}{2}((1-\beta)\nabla\log p_t^{(1)} + \beta\nabla\log p_t^{(2)}) \right)\mathrm{d}t$$

$$\quad - \beta\nabla\log p_t^{(2)} \cdot \left( f_t - \frac{\sigma_t^2}{2}\nabla\log p_t^{(2)} - f_t + \frac{\sigma_t^2}{2}((1-\beta)\nabla\log p_t^{(1)} + \beta\nabla\log p_t^{(2)}) \right)\mathrm{d}t \tag{217}$$

$$= (1-\beta)\beta\frac{\sigma_t^2}{2}\left( \nabla\log p^{(1)} - \nabla\log p^{(2)} \right) \cdot \left( \nabla\log p^{(1)} - \nabla\log p^{(2)} \right)\mathrm{d}t \tag{218}$$

$$\square$$

### H.4.4 REWARD-TILTING FKC

FKC also derive the reward-tilting formulas. Due to the update of their arXiv, they have two versions. In this section, we discuss that both are special cases of RNC.

**Version 1** In the appendix of FKC (V1[5], Skreta et al., 2025, Proposition D.5), the authors derived FKC for reward-tilting. However, their conclusion requires the reward model to be twice differentiable, and it necessitates computing the Laplacian of the reward model in order to form the importance weight. We note that this reward-tilting formulation can be derived as a special case of our RNC framework which in contrast is Laplacian free.

---

**Proposition H.6.** *Reward-FKC states that, for the following backward SDE*

$$\mathrm{d}X_t = u_t(X_t)\mathrm{d}t + \sigma_t\overleftarrow{\mathrm{d}W_t} \tag{219}$$

*and the importance weight for Sequential Monte Carlo satisfies the backward ODE:*

$$\mathrm{d}\log w_t = \left[ \beta_t\nabla r(X_t) \cdot \left( u_t(X_t) + \sigma_t^2\nabla\log p_t(X_t) + \frac{\sigma_t^2}{2}\beta_t\nabla r(X_t) \right) + \beta_t\frac{\sigma_t^2}{2}\Delta r(X_t) - \frac{\partial\beta_t}{\partial t}r(X_t) \right]\mathrm{d}t \tag{220}$$

*This is equivalent to our proposed RNC (Eq. (18)), when*

$$a_t = u_t, \quad b_t = u_t(X_t) + \sigma_t^2\nabla\log p_t(X_t) + \beta_t\sigma_t^2\nabla r(X_t), \quad \epsilon_t = \sigma_t \tag{221}$$

*with the intermediate reward $r_t = \beta_t r$.*

---

*Proof.* For the processes considered in Eq. (219), $\nu_t = u_t$ and $\mu_t = u_t + \sigma_t^2\nabla\log p_t$. Similar to the anneal, product and CFG cases:

$$\mathrm{d}\log R^\nu_\mu = -\frac{\nu_t - \mu_t}{\sigma_t^2} \cdot \overleftarrow{\mathrm{d}X_t} - \nabla\cdot\mu_t\mathrm{d}t + \frac{1}{2\sigma_t^2}(\nu_t - \mu_t)(\nu_t + \mu_t)\mathrm{d}t \tag{222}$$

$$= \nabla\log p_t \cdot \overleftarrow{\mathrm{d}X_t} - \nabla\cdot(u_t + \sigma_t^2\nabla\log p_t)\mathrm{d}t - \nabla\log p_t \cdot (u_t + \frac{\sigma_t^2}{2}\nabla\log p_t)\mathrm{d}t \tag{223}$$

and

$$\mathrm{d}\log R^a_b = -\frac{a_t - b_t}{\sigma_t^2} \cdot \overleftarrow{\mathrm{d}X_t} - \nabla\cdot b_t\mathrm{d}t + \frac{1}{2\sigma_t^2}(a_t - b_t)(a_t + b_t)\mathrm{d}t \tag{224}$$

$$= (\nabla\log p_t + \beta_t\nabla r) \cdot \overleftarrow{\mathrm{d}X_t} - \nabla\cdot(u_t + \sigma_t^2\nabla\log p_t + \beta_t\sigma_t^2\nabla r)\mathrm{d}t$$

$$\quad - (\nabla\log p_t + \beta_t\nabla r) \cdot \left( u_t + \frac{\sigma_t^2}{2}\nabla\log p_t + \beta_t\frac{\sigma_t^2}{2}\nabla r \right)\mathrm{d}t \tag{225}$$

Additionally, applying Itô's Lemma to $r_t = \beta_t r$, we have

$$\mathrm{d}r_t(X_t) = -\left( \partial_t\beta_t r(X_t) + \beta_t\frac{\sigma_t^2}{2}\Delta r(X_t) \right)\mathrm{d}t + \beta_t\nabla r(X_t) \cdot \overleftarrow{\mathrm{d}X_t} \tag{226}$$

---

[5]https://arxiv.org/abs/2503.02819v1

Therefore:

$$\mathrm{d}\log w_t = \mathrm{d}(r_t(X_t)) + \mathrm{d}\log R_\mu^\nu - \mathrm{d}\log R_b^a \tag{227}$$

$$= -\left(\partial_t\beta_t r(X_t) + \beta_t\frac{\sigma_t^2}{2}\Delta r(X_t)\right)\mathrm{d}t + \cancel{\beta_t\nabla r(X_t)\cdot\overleftarrow{\mathrm{d}X_t}} \tag{228}$$

$$+ \cancel{\nabla\log p_t\cdot\overleftarrow{\mathrm{d}X_t}} - \nabla\cdot(u_t + \sigma_t^2\nabla\log p_t)\mathrm{d}t - \nabla\log p_t\cdot(u_t + \frac{\sigma_t^2}{2}\nabla\log p_t)\mathrm{d}t \tag{229}$$

$$- \cancel{(\nabla\log p_t + \beta_t\nabla r)\cdot\overleftarrow{\mathrm{d}X_t}} + \nabla\cdot(u_t + \sigma_t^2\nabla\log p_t + \beta_t\sigma_t^2\nabla r)\mathrm{d}t \tag{230}$$

$$+ (\nabla\log p_t + \beta_t\nabla r)\cdot\left(u_t + \frac{\sigma_t^2}{2}\nabla\log p_t + \beta_t\frac{\sigma_t^2}{2}\nabla r\right)\mathrm{d}t \tag{231}$$

$$= -\left(\partial_t\beta_t r(X_t) + \beta_t\frac{\sigma_t^2}{2}\Delta r(X_t)\right)\mathrm{d}t \tag{232}$$

$$\textcolor{orange}{-\nabla\cdot(u_t + \sigma_t^2\nabla\log p_t)\mathrm{d}t - \nabla\log p_t\cdot(u_t + \frac{\sigma_t^2}{2}\nabla\log p_t)\mathrm{d}t} \tag{233}$$

$$\textcolor{orange}{+\nabla\cdot(u_t + \sigma_t^2\nabla\log p_t + \beta_t\sigma_t^2\nabla r)\mathrm{d}t} \tag{234}$$

$$\textcolor{orange}{+ (\nabla\log p_t + \beta_t\nabla r)\cdot\left(u_t + \frac{\sigma_t^2}{2}\nabla\log p_t + \beta_t\frac{\sigma_t^2}{2}\nabla r\right)\mathrm{d}t} \tag{235}$$

$$= \textcolor{orange}{\left(-\partial_t\beta_t r(X_t) + \beta_t\frac{\sigma_t^2}{2}\Delta r(X_t)\right)\mathrm{d}t} \tag{236}$$

$$\textcolor{orange}{+ \nabla\log p_t\cdot\beta_t\frac{\sigma_t^2}{2}\nabla r + \beta_t\nabla r\cdot\left(u_t + \frac{\sigma_t^2}{2}\nabla\log p_t + \beta_t\frac{\sigma_t^2}{2}\nabla r\right)\mathrm{d}t} \tag{237}$$

$$= \textcolor{orange}{\left(-\partial_t\beta_t r(X_t) + \beta_t\frac{\sigma_t^2}{2}\Delta r(X_t)\right)\mathrm{d}t + \beta_t\nabla r\cdot\left(u_t + \sigma_t^2\nabla\log p_t + \beta_t\frac{\sigma_t^2}{2}\nabla r\right)\mathrm{d}t} \tag{238}$$

$$\square$$

**Version 2**    In a recent work, Chen et al. (2025) proposed and empirically explored a formula for solving inverse problems without the need for the Laplacian. Shortly after, in the updated version of FKC (V2[6], Skreta et al., 2025, Proposition D.6), the authors included a similar result for reward-tilting. By carefully designing the sampling process, they can cancel the Laplacian of the reward model. As with Version 1 we now show how this reward-tilting formulation can be derived as a special case of RNC.

**Notably, comparing the two FKC variants highlights RNC's greater design flexibility**: FKC requires a special design to eliminate the Laplacian term, while RNC relies on a single, unified formula that does not require the Laplacian, and hence supports a wider range of sampling processes, including the heuristic choices proposed by Chung et al. (2023); Song et al. (2023b).

---

**Proposition H.7.** *for a perfect diffusion model (as defined in Eqs. (1) and (2)), one can implement the following backward sampling SDE:*

$$\mathrm{d}X_t = (f_t(X_t) - \sigma_t^2\nabla\log p_t(X_t) - \beta_t\frac{\sigma^2}{2}\nabla r(X_t))\mathrm{d}t + \sigma_t\overleftarrow{\mathrm{d}W_t} \tag{239}$$

*and the importance weight for Sequential Monte Carlo satisfies the backward ODE:*

$$\mathrm{d}\log w_t = \left[\beta_t\nabla r(X_t)\cdot\left(f_t(X_t) - \frac{\sigma_t^2}{2}\nabla\log p_t(X_t)\right) - \frac{\partial\beta_t}{\partial t}r(X_t)\right]\mathrm{d}t \tag{240}$$

*This is equivalent to our proposed RNC (Eq. (18)), when*

$$a_t = f_t - \sigma_t^2\nabla\log p_t - \beta_t\frac{\sigma_t^2}{2}\nabla r, \quad b_t = f_t + \beta_t\frac{\sigma_t^2}{2}\nabla r, \quad \epsilon_t = \sigma_t \tag{241}$$

---

[6]https://arxiv.org/abs/2503.02819v2

> *with the intermediate reward $r_t = \beta_t r$.*

*Proof.* Similar to the anneal, product and CFG case, for the diffusion model, we have

$$\mathrm{d}\log R_\mu^\nu = \nabla\log p_t \cdot \overleftarrow{\mathrm{d}X_t} - \nabla\cdot f_t\mathrm{d}t - \nabla\log p_t \cdot (f_t - \frac{\sigma_t^2}{2}\nabla\log p_t)\mathrm{d}t \tag{242}$$

and for the sampling & target processes:

$$\mathrm{d}\log R_b^a = -\frac{a_t - b_t}{\sigma_t^2} \cdot \overleftarrow{\mathrm{d}X_t} - \nabla\cdot b_t\mathrm{d}t + \frac{1}{2\sigma_t^2}(a_t - b_t)(a_t + b_t)\mathrm{d}t \tag{243}$$

$$= (\nabla\log p_t + \beta_t\nabla r) \cdot \overleftarrow{\mathrm{d}X_t} - \nabla\cdot(f_t + \beta_t\frac{\sigma_t^2}{2}\nabla r)\mathrm{d}t$$

$$-(\nabla\log p_t + \beta_t\nabla r) \cdot \left(f_t - \frac{\sigma_t^2}{2}\nabla\log p_t\right)\mathrm{d}t \tag{244}$$

Again, applying Itô's Lemma to $r_t = \beta_t r$, we have

$$\mathrm{d}r_t(X_t) = -\left(\partial_t\beta_t r(X_t) + \beta_t\frac{\sigma_t^2}{2}\Delta r(X_t)\right)\mathrm{d}t + \beta_t\nabla r(X_t) \cdot \overleftarrow{\mathrm{d}X_t} \tag{245}$$

Therefore:

$$\mathrm{d}\log w_t = \mathrm{d}(r_t(X_t)) + \mathrm{d}\log R_\mu^\nu - \mathrm{d}\log R_b^a \tag{246}$$

$$= -\left(\partial_t\beta_t r(X_t) + \beta_t\frac{\sigma_t^2}{2}\Delta r(X_t)\right)\mathrm{d}t + \cancel{\beta_t\nabla r(X_t) \cdot \overleftarrow{\mathrm{d}X_t}} \tag{247}$$

$$+ \cancel{\nabla\log p_t \cdot \overleftarrow{\mathrm{d}X_t}} - \nabla\cdot f_t\mathrm{d}t - \nabla\log p_t \cdot (f_t - \frac{\sigma_t^2}{2}\nabla\log p_t)\mathrm{d}t \tag{248}$$

$$- \cancel{(\nabla\log p_t + \beta_t\nabla r) \cdot \overleftarrow{\mathrm{d}X_t}} + \nabla\cdot(f_t + \beta_t\frac{\sigma_t^2}{2}\nabla r)\mathrm{d}t \tag{249}$$

$$+ (\nabla\log p_t + \beta_t\nabla r) \cdot \left(f_t - \frac{\sigma_t^2}{2}\nabla\log p_t\right)\mathrm{d}t \tag{250}$$

$$= -\left(\partial_t\beta_t r(X_t) + \beta_t\frac{\sigma_t^2}{2}\Delta r(X_t)\right)\mathrm{d}t \tag{251}$$

$$-\nabla\cdot f_t\mathrm{d}t - \nabla\log p_t \cdot (f_t - \frac{\sigma_t^2}{2}\nabla\log p_t)\mathrm{d}t \tag{252}$$

$$+\nabla\cdot(f_t + \beta_t\frac{\sigma_t^2}{2}\nabla r)\mathrm{d}t \tag{253}$$

$$+ (\nabla\log p_t + \beta_t\nabla r) \cdot \left(f_t - \frac{\sigma_t^2}{2}\nabla\log p_t\right)\mathrm{d}t \tag{254}$$

$$= (-\partial_t\beta_t r(X_t))\,\mathrm{d}t + \beta_t\nabla r \cdot \left(f_t - \frac{\sigma_t^2}{2}\nabla\log p_t\right)\mathrm{d}t \tag{255}$$

$$\square$$

## H.5 Connecting RNE Energy Regularisation with FPE Regularisation

Our proposed RNE regularisation is connected to the Fokker-Planck Equation (FPE) regularisation (Plainer et al., 2025) in the limit. We assume the diffusion model's noising drift is $f_t$ and the denoising drift is $f_t - \sigma_t^2\nabla\log p_t$. To make the discussion easier, we now swap the diffusion direction, so that $p_0$ corresponds to the Gaussian side, and $p_1$ corresponds to the data side. Therefore, the diffusion's noising and denoising processes are given by

$$\mathrm{d}X_t = -f_t(X_t)\mathrm{d}t + \sigma_t\overleftarrow{\mathrm{d}W_t}, \quad X_1 \sim p_1 \tag{256}$$

$$\mathrm{d}X_t = -f_t(X_t)\mathrm{d}t + \sigma_t^2\nabla\log p_t(X_t)\mathrm{d}t + \sigma_t\overrightarrow{\mathrm{d}W_t}, \quad X_0 \sim p_0 \tag{257}$$

We first recall the FPE in log-space:

$$\partial \log p_t(X_t) - \nabla \cdot f_t - \nabla \log p_t(X_t) \cdot f_t + \frac{\sigma_t^2}{2} \|\nabla \log p_t\|^2 + \frac{\sigma_t^2}{2} \Delta \log p_t = 0 \tag{258}$$

We then look at RNE:

$$\log p_{t+\Delta t}(X_{t+\Delta t}) - \log p_t(X_t) = \int_t^{t+\Delta t} \frac{1}{\sigma_s^2} \sigma_s^2 \nabla \log p_s \cdot \overrightarrow{dX_s} - \int_t^{t+\Delta t} f_s(X_s) \cdot \overrightarrow{dX_s}$$

$$+ \int_t^{t+\Delta t} f_s(X_s) \cdot \overleftarrow{dX_s} - \int_t^{t+\Delta t} \frac{1}{2\sigma_s^2} \|\sigma_s^2 \nabla \log p_s - f_s\|^2 ds + \int_t^{t+\Delta t} \frac{1}{2\sigma_s^2} \|f_s\|^2 ds \tag{259}$$

Using the conversion rule (Vargas et al., 2023b), we have:

$$\log p_{t+\Delta t}(X_{t+\Delta t}) - \log p_t(X_t) = \int_t^{t+\Delta t} \frac{1}{\sigma_s^2} \sigma_s^2 \nabla \log p_s \cdot \overrightarrow{dX_s} + \int_t^{t+\Delta t} \nabla \cdot f_s(X_s) ds \tag{260}$$

$$- \int_t^{t+\Delta t} \frac{1}{2\sigma_s^2} \|\sigma_s^2 \nabla \log p_s - f_s\|^2 ds + \int_t^{t+\Delta t} \frac{1}{2\sigma_s^2} \|f_s\|^2 ds \tag{261}$$

When $\Delta t \to 0$, we have

$$d \log p_t(X_t) = \frac{1}{\sigma_t^2} \sigma_t^2 \nabla \log p_t \cdot \overrightarrow{dX_t} + \nabla \cdot f_s(X_s) dt - \frac{1}{2\sigma_t^2} \|\sigma_t^2 \nabla \log p_t\|^2 dt + \nabla \log p_t \cdot f_t dt \tag{262}$$

Due to Itô's Lemma, we have

$$d \log p_t(X_t) = \partial_t \log p_t(X_t) dt + \frac{\sigma_t^2}{2} \Delta \log p_t dt + \nabla \log p_t(X_t) \cdot \overrightarrow{dX_t} \tag{263}$$

We can hence write the RNE relation as

$$\partial_t \log p_t(X_t) dt + \frac{\sigma_t^2}{2} \Delta \log p_t dt + \cancel{\nabla \log p_t(X_t) \cdot \overrightarrow{dX_t}}$$

$$- \cancel{\frac{1}{\sigma_t^2} \sigma_t^2 \nabla \log p_t \cdot \overrightarrow{dX_t}} - \nabla \cdot f_s(X_s) dt + \frac{1}{2\sigma_t^2} \|\sigma_t^2 \nabla \log p_t\|^2 dt - \nabla \log p_t \cdot f_t dt = 0 \tag{264}$$

which gives us the same expression as the FPE relation.

# I  ADDITIONAL EXPERIMENTAL DETAILS

## I.1  ADDITIONAL DETAILS FOR INFERENCE-TIME ANNEALING

**Network and Diffusion Hyperparameters.** For ALDP and LJ-13, we use the EGNN (Hoogeboom et al., 2022) with 4 layers and 64 hidden units. Following Karras et al. (2022), we parametrise the network as "denoiser" to output the mean value given noisy samples. We also rescale the input by $c_{in}$ and add skip connections following Karras et al. (2022). For GMM, we calculate the analytical score instead of training diffusion models.

We choose a VE-SDE: $dX_t = \sqrt{2t} \overrightarrow{dW_t}$, where $t \in [0.001, 10]$. We discretise the time horizon according to Karras et al. (2022) with $N = 200$ steps, i.e.,

$$t_n = \left(t_{\min}^{1/\rho} + \frac{n}{N}(t_{\max}^{1/\rho} - t_{\min}^{1/\rho})\right)^\rho, \quad n = 1, \cdots, N \tag{265}$$

**Dataset.** Alanine Dipeptide (ALDP) is a target with 22 atoms, each of which has 3 dimensions. The target is defined in implicit solvent, with the AMBER ff96 classical force field. Following He et al. (2025a), we gather samples from a 5-microsecond simulation under 300K with Generalised Born implicit solvent implemented in openmmtools Chodera et al. (2025). The Langevin middle integrator implemented by Eastman et al. (2023) with a friction of 1/picosecond and a step size of 2 femtoseconds was used to harvest a total of 250,000 samples.

Lennard-Jones (LJ)-13 is a system with 13 particles, with Lennard-Jones potential between all pairs of particles $i$ and $j$. Concretely, the entire potential of the system is defined as:

$$U = \sum_{i \neq j} U_{\text{LJ}}(\|X_i - X_j\|) + \frac{1}{2} \sum_{n=1}^{N} \left\| X_n - \frac{1}{N} \sum_{n'=1}^{N} X'_n \right\|^2 \tag{266}$$

where

$$U_{\text{LJ}}(r) = 4\epsilon \left[ \left( \frac{\sigma}{r} \right)^{12} - \left( \frac{\sigma}{r} \right)^{6} \right] \tag{267}$$

Here $\frac{1}{2} \sum_{n=1}^{N} \left\| X_n - \frac{1}{N} \sum_{n'=1}^{N} X'_n \right\|^2$ is a harmonic oscillator.

**FKC Details.** Skreta et al. (2025) discussed two choices for annealing: target score simulation, where one rescales the score by the temperature, and tempered noise, where one rescales the diffusion coefficient. In our experiment for ALDP, we report the former, as we found the latter achieves a significantly worse performance with severe mode collapsing. This is in line with their observation for GMM (see Figure 2 in Skreta et al. (2025)).

**RNE Details.** For all experiments, we use an analytical reference with the VE process where $\pi_0$ is a standard Gaussian.

**Resample Details.** For ALDP and LJ-13, we run SMC with a batch size of 500, and collect samples by repeating 50 batches. For GMM, to have a clearer visualisation, we collect 100 batches. We accumulate the weight along the generation process, and calculate Effective sample size (ESS). If ESS is smaller than 75%, we will perform resampling and reset the weight of all particles to 0.

**Computational Resources**. All experiments are run on a single NVIDIA H100 GPU.

## I.2 ADDITIONAL DETAILS FOR MULTI-TARGET SBDD

**Introduction.** Structure-based drug design (SBDD) (Blundell, 1996) is a main paradigm in drug discovery—given a protein target (i.e. pocket), we aim to design (small-molecule) ligands that bind to it. Recently, multi-target SBDD has attracted increasing attention to design ligands that bind to more than one target (Bolognesi & Cavalli, 2016). This problem can be formulated as sampling from the product of multiple diffusion models because of the inaccessibility to ligands that bind to multiple targets (Skreta et al., 2025). We take the pre-trained diffusion model from Guan et al. (2023), which is trained conditioning on each different protein pocket and generates ligands conditioned on a single target. We consider the dual target scenario with 20 pairs of protein targets, randomly sampled from the setting by Zhou et al. (2024). We validate the performance mainly based on the docking score calculated by Autodock Vina, with additional basic statistics and physicochemical properties (Eberhardt et al., 2021).

**Dataset**. We randomly sampled 20 pairs of protein targets from the dataset provided in (Zhou et al., 2024) with indices: (356, 233), (186, 341), (36, 333), (84, 41), (406, 169), (255, 39), (423, 45), (277, 262), (21, 334), (36, 121), (378, 143), (274, 307), (16, 143), (36, 345), (421, 420), (264, 26), (230, 70), (350, 137), (324, 423), (110, 39).

**Statistics**. The diversity is calculated as the pairwise distance between any two generated ligands (1 - Tanimoto similarity of their Morgan fingerprints). The quality is evaluated by the percentage of ligands that have QED $\geq 0.6$ and normalized SA score $\geq 0.67$.

**Experiment Details**. For each target, we sample 32 ligands with size 23 following (Skreta et al., 2025). We take pre-trained diffusion models conditioned on each protein target in (Guan et al., 2023).

**RNE Details.** For all experiments, we use an analytical reference with VP process at stationarity, following Vargas et al. (2023a).

**Product Details**. We consider the product of two diffusion models, i.e., $q_0 \propto \left( p_0^{(1)} p_0^{(2)} \right)^\beta$. In our experiments, we select $\beta = 2$ as it shows the best performance according to Skreta et al. (2025).

**Resampling Details**. For each protein target, we run SMC with a batch size of 32. Following Skreta et al. (2025), the resampling is performed when $t \in [0.4T, T]$.

**Computational Resources**. All experiments are run on a single NVIDIA H100 GPU.

## I.3 ADDITIONAL DETAILS FOR TRAJECTORY STITCHING EXPERIMENTS

**Network and Diffusion Hyperparameters.** We use an MLP of 5 layers and 512 hidden units. We use the VE (EDM) schedule with preconditioning $(c_{in}, c_{out}, c_{skip})$ following Karras et al. (2022). Following Luo et al. (2025), when training the network, we normalise the data to [-1, 1].

**Dataset.** We use the dataset `pointmaze-medium-stitch-v0` (Park et al., 2024) This is a dataset of short trajectories of length 64.

**Choice of Reward Function** Our reward function need to impose (1) first trajectory starts from the initial position and (2) the last trajectory ends at the target, and (3) consecutive trajectories are connected. Here, we follow He et al. (2025c) to define our reward function. For easy reference, we describe the design choice below. We will first define the reward $r$ for the clean space ($t = 0$), followed by the reward $r_t$ when $t > 0$. For each of the 3 constraints, we impose a combination of $L^2$ distance and $L^1$ distance. For now, we use $X^{j,i}$ to represent the $i$-th point in the $j$-th short trajectory. We use $-1$ to represent the last element, following the index convention of Python. We use $O$ and $P$ to represent the initial and target points.

$$\text{Reward for initial point: } r^O = -\lambda_O(\lambda_{L^2}||X^{0,0} - O||_2^2 + \lambda_{L^1}||X^{0,0} - O||_1^1) \tag{268}$$

$$\text{Reward for target point: } r^P = -\lambda_P(\lambda_{L^2}||X^{-1,-1} - P||_2^2 + \lambda_{L^1}||X^{-1,-1} - P||_1^1) \tag{269}$$

$$\text{Reward for neighboring trajectories:} \tag{270}$$

$$r^N = -\sum_j \lambda_N(\lambda_{L^2}||X^{j,-1} - X^{j+1,0}||_2^2 + \lambda_{L^1}||X^{j,-1} - X^{j+1,0}||_1^1) \tag{271}$$

where we set $\lambda_O = \lambda_P = 100 \times J$, $\lambda_N = 100$, $\lambda_{L^2} = 1$ and $\lambda_{L^1} = 10$. and the final reward is:

$$r = r^O + r^P + r^N \tag{272}$$

For the intermediate reward $r_t$, we define it following:

$$r_t(X_t^0, X_t^1, \cdots, X_t^J) = \beta_t \cdot r(\mathbb{E}[X_0|X_t^0], \mathbb{E}[X_0|X_t^1], \cdots, \mathbb{E}[X_0|X_t^J]) \tag{273}$$

We use $X_t^j$ to represent the sample at diffusion time step $t$ in the $j$-th short trajectory. $\mathbb{E}[X_0|X_t^j]$ is calculated by Tweedie's formula. $\beta_t$ is a smooth function between $\beta_1 = 0$ and $\beta_0 = 1$. We set $\beta_{t_n} = [\beta_1^{1/\rho} + \frac{n}{N}(\beta_0^{1/\rho} - \beta_1^{1/\rho})]^\rho$ and $\rho = 10$. When sampling, we discrete the diffusion process into $N = 600$ steps and apply resample at every step.

**RNE Details.** We choose $b_t$ to the standard noising process, and $a_t$ to be reward-guided process. More precisely, we add an extra term $\nabla r_t$ to the original score. We apply SMC with a batch size of 10,000. The SMC weight is calculated easily by RNC:

$$w_{[\tau,\tau']} \propto \frac{\exp(r_\tau([X_\tau^{(1)}, \cdots, X_\tau^{(L)}]))}{\exp(r_{\tau'}([X_{\tau'}^{(1)}, \cdots, X_{\tau'}^{(L)}]))} \left(\prod_l R_{\mu^{(l)}}^{\nu^{(l)}}(X_{[\tau,\tau']}^{(l)})\right) \left[R_b^a([X_{[\tau,\tau']}^{(1)}, \cdots, X_{[\tau,\tau']}^{(L)}])\right]^{-1}. \tag{274}$$

**Computational Resources**. All experiments are run on a single NVIDIA RTX 4090 GPU.

## I.4 ADDITIONAL DETAILS FOR CTMC-RNE

**Network and Diffusion Setups.** We use the MaskGIT model (Chang et al., 2022) as our network. This model is reproduced in PyTorch by Besnier et al. (2025) and we use their pretrained model weight. This is a model with two components, a VQ-GAN and a latent model to predict masked token value conditional on the unmasked region. This model on its own is not a diffusion model. However, similar to Ren et al. (2025), we can turn MaskGIT into a masked discrete diffusion by introducing a stochastic masking schedule. More precisely, following Shi et al. (2024), we introduce a stochastic masking schedule $\alpha_t$. Following the notation of Shi et al. (2024) where we use $e_m$ to represent a one-hot vector where element at the mask index is 1, the forward (masking) process is defined by

$$p(x_t|x_s) = \text{Cat}\left(x_t \left| \left(\frac{\alpha_t}{\alpha_s}I + (1 - \frac{\alpha_t}{\alpha_s})\mathbf{1}e_m^\top\right)^\top x_s\right.\right) \tag{275}$$

and the backward (unmasking) process is defined as

$$p(x_s|x_t) = \text{Cat}\left(x_s \middle| \left(I + \frac{\alpha_s - \alpha_t}{1 - \alpha_t} e_m(\text{MaskGIT}(x_t) - e_m)^\top\right)^\top x_t\right) \tag{276}$$

where $\text{MaskGIT}(x_t)$ predicts the probability of tokens given input $x_t$.

**Choice of Reward Function.** We define the reward function $r$ by ImageReward (Xu et al., 2023). It takes a text prompt and an image, outputting a score reflecting the alignment between the image and the prompt. We amplify the IR value by 10 to amplify the reward strength. Note that IR is defined over images, while our masked diffusion is in the latent discrete space. Therefore, every time we evaluate the reward, we need to first reconstruct the image by the decoder of the VQ-GAN. We define the intermediate reward $r_t(x_t) = r(\hat{x}_0)$, where $\hat{x}_0$ is obtained by directly input $x_t$ into MaskGIT, and taking the token with the largest probability.

**RNE Details.** When generation, we use a CFG strength of 2. We choose $b_t$ to be standard unmasking process (with CFG), and $a_t$ to be the standard masking process. We discretisation the unmasking process with 128 steps. We apply a batch size of 32, and resample at every time step except the first and the last one to aviod numerical issue. We apply both CFG debiasing and reward-tilting together, and hence the SMC weight is given by the combination of Eqs. (18) and (19):

$$w_{[\tau,\tau']} \propto \frac{\exp(r_\tau(X_\tau))}{\exp(r_{\tau'}(X_{\tau'}))} \left[R_{\mu^{(1)}}^{\nu^{(1)}}(X_{[\tau,\tau']})\right]^\alpha \left[R_{\mu^{(2)}}^{\nu^{(2)}}(X_{[\tau,\tau']})\right]^\beta \left[R_b^a(X_{[\tau,\tau']})\right]^{-1}. \tag{277}$$

### I.5 Additional Details for Training Energy-based Diffusion Models

To obtain $X_t$ and $X_{t+\Delta t}$ for the objective Eq. (22), we simply add noise to the training data. Plugging in the definition of $R$, we obtain the regularisation term:

$$\mathcal{R}_1 = \mathbb{E}_{x_{t+\Delta t}, x_t, x_0, t}\|\text{sg}(\log p^\nu(x_t|x_{t+\Delta t}) - \log p^\mu(x_{t+\Delta t}|x_t)$$
$$+ \log p_{t+\Delta t}(x_{t+\Delta t}) - \log p_t(x_t)\|^2 \tag{278}$$

We can also use the reference process as introduced in Section 3, leading to the following objective:

$$\mathcal{R}_2 = \mathbb{E}_{x_{t+\Delta t}, x_t, x_0, t}\|\text{sg}\left[\log p^\nu(x_t|x_{t+\Delta t}) - \log p^\psi(x_t|x_{t+\Delta t})\right.$$
$$\left. - \log p^\mu(x_{t+\Delta t}|x_t) + \log p^\phi(x_{t+\Delta t}|x_t) + \log \pi_t(x_t) - \log \pi_{t+\Delta t}(x_{t+\Delta t})\right]$$
$$+ \log p_{t+\Delta t}(x_{t+\Delta t}) - \log p_t(x_t)\|^2 \tag{279}$$

The entire training loss is

$$\mathcal{L} = \mathbb{E}_t \mathbb{E}_{x_0} \mathbb{E}_{x_t|x_0} \mathbb{E}_{x_{t+\Delta t}|x_t} \left[\ell_{\text{DSM}} + \lambda_\mathcal{R} \mathcal{R}_{1 \text{ or } 2}\right], \tag{280}$$

where we choose the strength $\lambda_\mathcal{R} = 10^3$ and $\Delta t = 10^{-4}$ and found these hyperparameters generalise well across difference targets.

For both GMM and ALDP, we use a VE process following Karras et al. (2022) ($\mathrm{d}X_t = \sqrt{2t}\overrightarrow{\mathrm{d}W_t}$, where $t \in [0.001, 10]$), and take the inner product between the input and network output to obtain a scalar value as the (negative) energy, i.e., $\log p_t(x_t) \approx g_\theta(x_t, t) = NN(c_{\text{in}}x_t, t) \cdot x_t$, where $NN$ is the neural network, $c_{\text{in}}$ is the rescaling factor used by Karras et al. (2022). For GMM in 100D, we found it is beneficial to add another scalar network: $\log p_t(x_t) \approx g_\theta(x_t, t) = NN(c_{\text{in}}x_t, t) \cdot x_t + NN_2(c_{\text{in}}x_t, t)$ where $NN_2$ outputs a scalar. We use a standard MLP for GMM and an EGNN for ALDP. We scale up the RNE regularisation by $\lambda_\mathcal{R}$ as it is generally close to 0, which results in the following loss function:

$$\mathcal{L} = \mathbb{E}_t \mathbb{E}_{x_0} \mathbb{E}_{x_t|x_0} \mathbb{E}_{x_{t+\Delta t}|x_t} \left[t^2 \ell_{\text{DSM}} + \lambda_\mathcal{R} \mathcal{R}_2\right], \quad \ell_{\text{DSM}} = \|\nabla \log \mathcal{N}(x_t|x_0, t^2) - \nabla g_\theta(x_t, t)\|^2 \tag{281}$$

where we choose $\lambda_\mathcal{R} = 10^3$ and $\Delta t = 10^{-4}$, and we sample $t$ by $\log t \sim \mathcal{N}(-1.2, 1.2)$ following Karras et al. (2022). We found this set of hyperparameters works well for both 2D GMM and ALDP.

**Details on Dual SM Baseline.** Yu et al. (2025) proposed Time Score Matching loss and was later adopted by Dual SM (Guth et al., 2025) as a regularisation term:

$$\ell_{\text{TimeSM}} = \|\partial_t g_\theta(x_t, t) - \partial_t \log p(x_t|x_0, t)\|^2 \tag{282}$$

We use the same VE process following Karras et al. (2022), and hence $\log p(x_t|x_0, t) = \mathcal{N}(x_t|x_0, t^2)$. We also reweight the Time SM and DSM term following (Guth et al., 2025), leading to the following loss:

$$\mathcal{L} = \mathbb{E}_t \mathbb{E}_{x_0} \mathbb{E}_{x_t|x_0} \left[ \frac{t^2}{d} \ell_{\text{DSM}} + \frac{t^2}{d^2} \ell_{\text{TimeSM}} \right] \tag{283}$$

where $d$ is the dimensionality. Note that the exact form of the objective is different from that used by Guth et al. (2025) because they assumed $\log p(x_t|x_0, t) = \mathcal{N}(x_t|x_0, t)$. However, we follow their principle to ensure unitless of both Time SM and DSM terms.

### I.6 BACKGROUND AND DETAILS FOR FREE ENERGY ESTIMATION WITH THERMODYNAMIC INTEGRATION

#### I.6.1 BACKGROUND ON FREE ENERGY

We first provide a brief introduction to the background on free energy, following the discussion of He et al. (2025a). For a more comprehensive treatment, we refer the reader to He et al. (2025a). Precisely, the free energy is expressed as:

$$F = -\log Z, \qquad Z = \int_\Omega \exp(-U(x))\mathrm{d}x \tag{284}$$

where $\Omega \subseteq \mathbb{R}^d$, $U : \Omega \to \mathbb{R}$ is the energy function, assumed to be such that $Z < \infty$. In many cases, rather than calculating $F$ directly, one may be interested in the free energy difference between systems (or states) $S_a$ and $S_b$ with energies $U_a$ and $U_b$. This is important for biological conformational changes, ligand-macromolecule binding, or chemical reaction mechanisms (Wang et al., 2015):

$$\Delta F = F_b - F_a = -\log(Z_b/Z_a) \tag{285}$$

Zwanzig (1954) reformulated the problem as *importance sampling*, where one system serves as the proposal and the free energy difference is estimated via Monte Carlo sampling. This is known as the free energy perturbation (FEP) method:

$$\Delta F = -\log(Z_b/Z_a) = -\log \mathbb{E}_a \left[ \exp(U_a - U_b) \right], \tag{286}$$

where we use $\mathbb{E}_a$ to denote the expectation with respect to the equilibrium distribution $\mu_a(\mathrm{d}x) = Z_a^{-1} e^{-U_a(x)} \mathrm{d}x$ of system $S_a$.

On the other hand, the Thermodynamic Integration (TI) approach introduces a sequence of distributions that connects the two marginal distributions and estimates free energy difference as follows:

$$\Delta F = \int_0^1 \frac{\partial F_t}{\partial t} \mathrm{d}t \tag{287}$$

$$= -\int_0^1 \frac{\frac{\partial Z_t}{\partial t}}{Z_t} \mathrm{d}t \tag{288}$$

$$= -\int_0^1 \frac{\int \exp(-U_t)(-\frac{\partial U_t}{\partial t})\mathrm{d}x}{\int \exp(-U_t)\mathrm{d}x} \mathrm{d}t \tag{289}$$

$$= \int_0^1 \mathbb{E}_{p_t} \left[ \frac{\partial U_t}{\partial t} \right] \mathrm{d}t \tag{290}$$

In our experiment, we will aim to estimate the free-energy difference using the TI formula with a learned energy path $U_t$, similar to neural TI (Máté et al., 2025).

### I.6.2 EXPERIMENTAL DETAILS

**System Details.** We estimate the solvation free energy for alanine dipeptide following He et al. (2025a). Concretely, we consider the free energy difference between ALDP in the vacuum environment and with implicit solvent, defined with AMBER ff96 classical force field. We train our model with samples used by He et al. (2025a). The author gathered the training set from a 5 microsecond simulation under 300K with Generalized Born implicit solvent implemented in `openmmtools` (Chodera et al., 2025). The Langevin middle integrator implemented in Eastman et al. (2023) with a friction of 1/picosecond and a step size of 2 femtoseconds was used to harvest a total of 250,000 samples.

Below are settings for $S_a$ and $S_b$:

- $S_a$: ALDP in the vacuum environment;
- $S_b$: ALDP in implicit solvent.

Similar to He et al. (2025a), when training the network, we rescale each target scale by 20, i.e., we define the energy as $U\left(\frac{x}{20}\right)$. Note that this will only change the scale of input and the score, with no influence on the free energy difference as long as we apply the same scaling to both targets.

**Stochastic Interpolant Training Details.** We train a stochastic interpolant model bridging between $S_a$ and $S_b$. The model has two networks, a vector field and an energy network:

Given pairs of samples $(x_a, x_b)$ from systems $S_a$ and $S_b$, we first define an interpolant:

$$I_t = \alpha_t x_a + \beta_t x_b + \gamma_t \epsilon, \quad \epsilon \sim \mathcal{N}(0, \mathrm{Id}) \tag{291}$$

where $\alpha_0 = 1, \alpha_1 = 0$; $\beta_0 = 0, \beta_1 = 1$; and $\gamma_0 = \gamma_1 = 0$ ensure proper boundary conditions: $I_{t=0} = x_a$ and $I_{t=1} = x_b$. By Albergo et al. (2023), the vector field and energy path is defined as

$$v_t(x) = \mathbb{E}[\dot{I}_t | I_t = x], \qquad \nabla U_t(x) = \gamma_t^{-1} \mathbb{E}[\epsilon | I_t = x] \tag{292}$$

where the dot denotes the time derivative and $\mathbb{E}[\cdot | I_t = x]$ denotes expectation over the law of $I_t$ conditional on $I_t = x$. Using the $L^2$ formulation of the conditional expectation, we can write objective functions for the function $v_t$ and $\nabla U_t$ defined in Eq. (292); if we parametrize these functions as neural networks $v_t^\psi(x)$ and $U_t^\theta(x)$, depending on both $t$ and $x$, this leads to the losses:

$$\mathcal{L}_v(\psi) = \mathbb{E}_{t \sim \mathcal{U}(0,1)} \mathbb{E}_{x_a, x_b, \epsilon} \left[ \lambda_t | v_t^\psi(I_t) - \dot{I}_t |^2 \right] \tag{293}$$

$$\mathcal{L}_U(\theta) = \mathbb{E}_{t \sim \mathcal{U}(0,1)} \mathbb{E}_{x_a, x_b, \epsilon} \left[ \eta_t | \nabla U_t^\theta(I_t) - \gamma_t^{-1} \epsilon |^2 \right] \tag{294}$$

where $\lambda_t$ and $\eta_t$ are weighting functions to balance optimisation across different times. In practice, we follow He et al. (2025a) to set $\lambda_t = 1$ and $\eta_t = \gamma_t$.

Additionally, we also use target score matching (TSM, De Bortoli et al., 2024) to enhance the energy learning following (Máté et al., 2025; He et al., 2025a):

$$\mathcal{L}_U^{\mathrm{TSM},0}(\theta) = \mathbb{E}_{t \sim \mathcal{U}(0,0.5)} \mathbb{E}_{x_a, x_b, \epsilon} \left[ | \nabla U_t^\theta(I_t) - \alpha_t^{-1} \nabla U_a(x_a) |^2 \right] \tag{295}$$

$$\mathcal{L}_U^{\mathrm{TSM},1}(\theta) = \mathbb{E}_{t \sim \mathcal{U}(0.5,1)} \mathbb{E}_{x_a, x_b, \epsilon} \left[ | \nabla U_t^\theta(I_t) - \beta_t^{-1} \nabla U_b(x_b) |^2 \right] \tag{296}$$

At optimality, the following two SDEs become time reversals of each other ($\forall \sigma_t \geq 0$):

$$dX_t = -\frac{1}{2} \sigma_t^2 \nabla U_t^\theta(X_t) dt + v_t^\psi(X_t) dt + \sigma_t \overrightarrow{dB_t}, \quad X_0 \sim \mu_a, \tag{297}$$

$$dX_t = \frac{1}{2} \sigma_t^2 \nabla U_t^\theta(X_t) dt + v_t^\psi(X_t) dt + \sigma_t \overleftarrow{dB_t}, \quad X_1 \sim \mu_b, \tag{298}$$

where $\mu_a$ and $\mu_b$ are the distribution defined via energy $U_a$ and $U_b$.

We train the model with a batch size of 20. Also, to improve results and to accelerate convergence, we apply mini-batch Optimal Transport Tong et al. (2024) when sampling the pair $(x_a, x_b)$. We also follow He et al. (2025a) to use a different batch size for OT (500) and for training the network (20). We train both methods for 40,000 iterations.

We parametrise the energy network with inner product in the same way as Appendix I.5, i.e., $U^\theta(x_t, t) = NN(c_{\mathrm{in}} x_t, t) \cdot x_t$. Note that in Máté et al. (2025), the author add preconditioning to

ensure $U^\theta(\cdot, 0) = U_a(\cdot)$ and $U^\theta(\cdot, 1) = U_b(\cdot)$. However, this will require calling the target energy during training and will be less efficient (He et al., 2025b). Also, this requires designing a smoother parameter for $t \in (0, 1)$, which is easier for the LJ system considered by Máté et al. (2025) but non-trivial for ALDP. Therefore, we drop this preconditioning and fully parametrise the energy with a neural network.

For the baseline without RNE regularisation, we only train the energy network using $\mathcal{L}_U + \mathcal{L}_U^{\text{TSM, 0}} + \mathcal{L}_U^{\text{TSM, 1}}$. For the results with RNE regularisation, we train both the vector field and the energy network, with an additional RNE regularisation: $\mathcal{L}_U + \mathcal{L}_U^{\text{TSM, 0}} + \mathcal{L}_U^{\text{TSM, 1}} + \lambda_{\mathcal{R}} \mathcal{R}$, where

$$\mathcal{R} = \mathbb{E}_{x_{t+\Delta t}, x_t, x_0, t} \| \text{sg}(\log p^\mu(x_t | x_{t+\Delta t}) - \log p^\nu(x_{t+\Delta t} | x_t)) \\ - U_{t+\Delta t}^\theta(x_{t+\Delta t}) + U_t^\theta(x_t) \|^2 \quad (299)$$

where $\mu = \frac{1}{2}\sigma_t^2 \nabla U_t^\theta(X_t) + v_t^\psi(X_t)$, and $\nu = -\frac{1}{2}\sigma_t^2 \nabla U_t^\theta(X_t) + v_t^\psi(X_t)$. We found the hyperparameters used in the diffusion setting generalise well here: $\Delta t = 10^{-4}$ and $\lambda_{\mathcal{R}} = 10^3$. Different from the diffusion setting, we have the freedom to choose any $\sigma_t \geq 0$ for the forward and backwards pair of SDE. We hence choose $\sigma_t = \sqrt{0.2}, \forall t$. We did not find that this choice had a significant influence on the results. Also, as $\sigma_t$ is a constant for any time step $t$, the instability issue discussed in Section 3 does not happen in this setting, and hence we do not use the analytical reference here.

**Estimation Details.** After training the network, we estimate the free-energy difference using the TI formula. This estimation is identical for both the baseline and the RNE regularisation. One caveat, however, is that when training without preconditioning, the boundary conditions are not satisfied. This not only makes the TI estimation inaccurate but also can introduce a constant shift at the boundaries. In other words, the free energies of $U_a$ and $U_0^\theta$ differ, and the same holds for $U_b$ and $U_1^\theta$.

To account for this mismatch, we estimate the free energy difference between $U_a$ and $U_0^\theta$ using FEP formulation as described in Eq. (286) with samples from $\mu_a \propto \exp(-U_a)$. Similarly, we estimate the free energy difference between $U_b$ and $U_1^\theta$ using FEP with samples from $\mu_b \propto \exp(-U_b)$.

We then estimate the free energy difference between $U_0^\theta$ and $U_1^\theta$ using the TI formulation in Eq. (290). The final free energy difference between $U_a$ and $U_b$ will be the summation of these three estimates:

$$\Delta F_{U_a, U_b} = \Delta F_{U_a, U_0^\theta} + \Delta F_{U_0^\theta, U_1^\theta} + \Delta F_{U_1^\theta, U_b} \quad (300)$$

Note that Eq. (290) requires the sample from $p_t \propto \exp(-U_t^\theta)$. However, we only have samples from $\mu_a$ and $\mu_b$. Therefore, we take the assumption that $I_t \sim p_t$, where $I_t$ is defined as Eq. (291), similar to Máté et al. (2025). When the energy path is learned poorly, this assumption breaks down severely, resulting in inaccurate free-energy estimates. Conversely, a well-learned energy network improves the accuracy of the estimation. Hence, this serves as a good metric for assessing whether our proposed RNE regularisation provides improvements in learning a more accurate energy network.

We estimate $\Delta F_{U_a, U_0^\theta}$ and $\Delta F_{U_1^\theta, U_b}$ using 5,000 samples each. For $\Delta F_{U_0^\theta, U_1^\theta}$, we use 5,000 samples with 1,000 steps, uniformly discretising the interval $(0, 1)$. We repeat baseline and RNE regularisation 3 times, and report the mean and standard deviation in Tab. 5. The reference value is taken from He et al. (2025a), which was obtained with MBAR (Shirts & Chodera, 2008).

We also provide a visualisation comparing $U_a$ with the learned $U_0^\theta$, and $U_b$ with the learned $U_1^\theta$ in Fig. 19. In summary, the TI estimates reported in Tab. 5 show that RNE improves the energy along the path, while Fig. 19 shows RNE improves the energy at two ends.

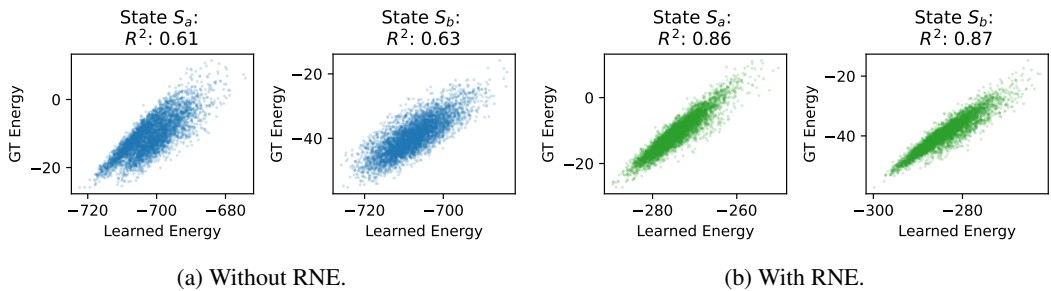

(a) Without RNE.  (b) With RNE.

Fig 19: Comparing $U_a$ and $U_b$ with the learned $U_0^\theta$ and $U_1^\theta$ without / with RNE regularisation.

## J  RELATED WORKS IN SAMPLING FROM UNNORMALISED DENSITIES

It is important to highlight that in the task of sampling from unnormalised densities, we have seen a recent uptake in methods that exploit the RND between SDEs for Sequential Monte Carlo (Chen et al., 2024; Albergo & Vanden-Eijnden, 2024; Tan et al., 2025). Among these, Chen et al. (2024) is most closely aligned with our methodology, as it also uses the RND between forward and backward SDEs. However, their approach is not directly applicable to generative models, as it relies on access to the intermediate densities, which was manually designed in their case as a geometric interpolation between the prior and the target. In our case, these intermediate densities are not tractable, and this is precisely where our RNE framework comes in and provides a principled solution.

