# OpenReview forum: "RNE: plug-and-play diffusion inference-time control and energy-based training"
_ICLR.cc/2026/Conference — ICLR 2026 Poster_

### Official Review · Reviewer_zcEE · 2025-10-21

**Soundness:** 3
**Presentation:** 2
**Contribution:** 3
**Rating:** 6
**Confidence:** 3

**Summary:**

The paper proposes a novel **Radon–Nikodym Estimator (RNE)** framework that unifies marginal density estimation, inference-time control, and energy-based training in diffusion models. By leveraging time-reversal theory, RNE connects marginal densities and transition kernels without explicitly solving Fokker–Planck equations. Building on this, the **Radon–Nikodym Corrector (RNC)** enables plug-and-play inference-time control—including annealing, reward-tilting, and model composition—via sequential Monte Carlo weighting. Additionally, RNE serves as a principled regularizer for energy-based diffusion training and generalizes beyond standard diffusion models to stochastic interpolants, bridge models, and continuous-time Markov chains.

**Strengths:**

The paper presents a theoretically elegant and practically flexible framework for diffusion-based sampling. By introducing the Radon–Nikodym Estimator (RNE) and Corrector (RNC), it unifies marginal densities, transition kernels, and various correction techniques under a single probabilistic formalism. This approach enables plug-and-play inference-time control, bridges diffusion and energy-based modeling, generalizes beyond Gaussian diffusions, and demonstrates clear empirical improvements on tasks such as annealing, ligand design, and free-energy estimation, all while maintaining mathematical rigor and conceptual coherence.

**Weaknesses:**

### About the bound of the importance weight

One potential weakness of the proposed method is that the guidance weight w(t) used in the conditional or composite sampling scheme is not explicitly constrained. When w(t) becomes too large or varies significantly over time, it can cause several issues:

1. **High variance in importance weights** – In the sequential importance sampling / SMC interpretation, unbounded w(t) directly amplifies the RN derivative, which may lead to weight degeneracy, where only a few particles dominate the distribution. This reduces effective sample size and lowers approximation quality.
2. **Numerical instability or reducing diversity** – Large w(t) can excessively scale the conditional drift/score term, potentially causing gradient explosion or unstable particle trajectories in continuous-time SDE sampling. And excessive weighting favors certain modes strongly, leading to collapse of particle diversity and biased approximation of the target marginal qt.

### About CFG

In your formulation, **CFG** is considered as
$$
d \tilde X_t=\left(f_t(X_t)-(1-\beta )\sigma^2\nabla\log p_t^{(1)}-\beta \sigma^2_t\nabla\log p_t^{(2)}\right)dt+\sigma d\bar W_t,
$$
however, in some practical or generalized settings, the CFG formulation takes the form:
$$
d \tilde X_t=\left(f_t(X_t)-\sigma^2\nabla\log p_t^{(1)}-\gamma \sigma^2_t\nabla\log p_t^{(2)}\right)dt+\sigma d\bar W_t,
$$
i.e.,
$$
\alpha \neq (1-\beta).
$$
Under this more general setting, certain derivations or propositions in the appendix (e.g., **Proposition H.4**) may not hold exactly.
 Could the authors clarify:

- whether their framework can accommodate the case
  $$
  \alpha+\beta\neq 1;
  $$

- and if not, what theoretical or empirical assumptions justify enforcing
  $$
  \alpha+\beta=1
  $$
  in CFG-like processes?

### Reading difficulty / dense derivations

Some sections, especially the appendix and derivations of the CFG and RNE formulas, are mathematically dense. Readers may find it challenging to follow all steps without carefully tracing the equations. Adding more intuitive explanations or diagrams could improve accessibility.

**Questions:**

### A little typo

I believe there is an issue with the use of Girsanov's theorem at line **1763**. The formulation of the Radon-Nikodym derivative seems to be incomplete, as it is missing the standard **exponential term**.

---

> ### Author Response · Authors · 2025-11-18
> **Reply to Reviewer zcEE**
>
> Thank you for your constructive and valuable feedback. We are glad that you found our framework theoretically elegant and practically flexible. We respond to your points in detail below.
>
> ---
> > ### One potential weakness of the proposed method is that the guidance weight w(t) used in the conditional or composite sampling scheme is not explicitly constrained. When w(t) becomes too large or varies significantly over time, it can cause several issues:
>
> Thank you for pointing this out. We agree that large or highly variable guidance weights are a common issue in SMC / importance sampling.
>
> When the target and proposal differ too much, the importance weights $w$ can have large variance, leading to low ESS, and reduce the sample quality or collapsed samples. However, this limitation is **inherent to all SMC-based control approaches**, including FKC [1], TDS [2], and related methods. RNE does not worsen this situation; instead, it **provides a more flexible way to choose the proposal and target processes, which can actually help reduce weight variance** by better aligning forward and backward dynamics.
>
> [1] Skreta, Marta, et al. Feynman-Kac correctors in diffusion: Annealing, guidance, and product of experts. ICML.
> [2] Wu, Luhuan, et al. Practical and asymptotically exact conditional sampling in diffusion models. NeurIPS.
>
>
> > ### About CFG
>
> Thank you for the question and for highlighting the more general CFG setting.
>
> Our framework **can accommodate any linear combinations** of conditional and unconditional scores, i.e. any combination of $\alpha$ and $\beta$.
> Proposition H.4 (now Proposition H.5 in our updated manuscript) is used only to show that FKC arises as a special case of RNE. RNE itself does impose any restriction on $\alpha$ and $\beta$. For general $\alpha$ and $\beta$, we can simply use the RNE weight in Eq. 20 for the model product.
>
>
> > ### Reading difficulty / dense derivations
>
>
>
> We appreciate this comment and agree that some of the appendix derivations are dense and can be hard to follow on a first read.
>
> In the revised manuscript, we have added short summaries at the beginning of Appendix H.4 to explain the main ideas behind the proof. In the camera-ready version, we will also provide proof sketches for our key theoretical results to make the arguments easier to follow.
>
> > ### A little typo
>
> Thank you for pointing this out. We have fixed this typo.
>
> ---
>
> We would like to thank you again for your valuable feedback, which has helped us improve the clarity of our manuscript. We would be very happy to discuss any further points you may have; if you find our replies satisfactory, we kindly invite you to reconsider your rating.

---

> > ### Comment · Reviewer_zcEE · 2025-11-27
> >
> > Thank you for the response. My confusion has been well resolved, and I will maintain the positive rating.

---

### Official Review · Reviewer_TNy1 · 2025-10-23

**Soundness:** 2
**Presentation:** 3
**Contribution:** 2
**Rating:** 4
**Confidence:** 3

**Summary:**

The paper introduces the Radon-Nikodym Estimator (RNE), a framework designed to unify various diffusion models by connecting marginal densities with transition kernels through the density ratio between time-reversal processes.  The paper also presents RNE as generalizing existing methods like FKC, TDS, and Itô density estimators, offering increased flexibility and computational efficiency.   Empirical results validate RNE across inference control, energy training, and density estimation on molecular dynamics, drug design, and Gaussian mixtures.

**Strengths:**

1. The RNE framework offers a conceptually appealing and potentially unifying perspective on several recent techniques for controlling and analyzing diffusion models.

2.  The proposed RNC method appears to offer increased flexibility in designing the sampling ($a_t$) and target ($b_t$) processes compared to methods like FKC, which often have more constrained designs to avoid divergence terms.


3.  The identified potential numerical instability and the introduced reference process are validated by empirical results.

**Weaknesses:**

1. The introduced reference process, while stabilizing the RND estimator's variance, appears to increase the computational cost.

2.  How reference process approach relates conceptually to methods that aim to directly minimize the conditional variance $Var(x_{t_i}|x_{t_{i+1}})$ or reconstruction error during the reverse process itself, rather than stabilizing the estimator of a ratio?


2. While RNC offers flexibility via parameters like $(c_a, c_b)$, the practical advantage over established methods like FKC seems conditional on tuning these parameters.

3. The paper proposes extending RNE to Continuous Time Markov Chains for discrete diffusion, but this extension is discussed briefly in the appendix (Appendix D) and lacks empirical validation in the experiments section.

**Questions:**

1. Does the performance gain from RNC's flexibility (tuning $c_a, c_b$) consistently outweigh the tuning effort compared to methods like FKC? How can these parameters be chosen efficiently?

2. How does the proposed method handle the numerical instability issue arising from the schedules in diffusion models as $\sigma_t$ approaches 0?

3.  How sensitive is RNC's performance to score model inaccuracies in practice?  Are there theoretical principles governing this sensitivity or ensuring robustness, apart from the specific case noted for reward-tilting (Proposition 2.2)?

---

> ### Author Response · Authors · 2025-11-18
> **Reply to Reviewer TNy1 (part I)**
>
> We appreciate your thoughtful and constructive review. **Following your feedback, we have updated our manuscript with new empirical results and theoretical guarantees.** We address your concerns and questions below.
>
> ---
>
> > ### W1: The introduced reference process increases the computational cost.
>
> **In practice, the reference process adds almost no extra cost.** The reference process is chosen to be a simple, tractable VE/OU process with analytic Gaussian transition kernels and marginals. Computing its contribution only involves closed-form Gaussian evaluations; no additional neural network evaluations are required. Since the dominant cost in diffusion generation comes from score-network calls, the extra arithmetic for the reference process is negligible in comparison.
>
> We apologise for not making this clear in the original version. We have updated the manuscript (around line 357) to clarify this point.
>
>
> > ### W2: How reference process approach relates conceptually to methods that aim to directly minimize the conditional variance?
>
>
>
> Are you referring to the methods estimating the denoising variance like [1, 2]? This line of approach aims to optimise the conditional variance to align it better with the data.
>
> They are distinct from the reference process we propose. They try to estimate the variance of the denoising kernel, as a better estimation will reduce the number of sampling steps and boost the generation quality. This approach does not stabilise the forward-backwards RND on its own. In fact, **these methods are orthogonal to the reference we proposed, and we can also apply the reference process together with their approach.**
>
>
> To verify this, we **provide an additional experiment in our updated manuscript in Appendix F.5**. More precisely, we evaluate the performance of RNE for density estimation (RNDE) on a 10D GMM with 40 modes, for which the marginal density is available in closed form, allowing us to directly assess the accuracy of our RNE estimator.
>
> We compare the performance using EM discretisation, and the denoising kernel with Estimated Variance $Var(x_{t_i}|x_{t_{i+1}})$ following Theorem 1 in [2]. For both choices of the denoising kernel, we plot the results (MSE) obtained with and without the reference process in Fig. 17 in our updated manuscript. For easier reference, we also summarise the MSE value for the RNE density estimator at $t=0.001$ in the following table.
>
> We can see that using a better denoising kernel with estimated variance, the RNE estimator indeed gets more accurate. Furthermore, even in this setting, incorporating our reference process still provides an additional boost, further reducing the error and yielding highly accurate estimates.
>
>
>
> |  | EM discretisation | EM discretisation  | Estimated Variance | Estimated Variance |
> |---|---|---|---|---|
> |  | without reference | with reference | without reference | with reference |
> | #discretisation steps = 100 | 167.489 | 12.475 | 3.285 | 0.596 |
> | #discretisation steps = 200 | 44.235 | 4.002 | 1.224 | 0.264 |
> | #discretisation steps = 400 | 13.252 | 1.579 | 0.459 | **0.086** |
>
>
> [1] Bao, Fan, et al. "Analytic-DPM: an Analytic Estimate of the Optimal Reverse Variance in Diffusion Probabilistic Models." ICLR.
>
> [2] Ou, Zijing, et al. "Improving Probabilistic Diffusion Models With Optimal Diagonal Covariance Matching." ICLR.
>
>
> > ### W3: While RNC offers flexibility via parameters like $c_a, c_b$, the practical advantage over established methods like FKC seems conditional on tuning these parameters.
>
> We first note that RNC is flexible at the level of the full forward and backward drift $a_t$ and $b_t$ instead of just parameters $c_a, c_b$. We reduce the design space to two scalars  $c_a, c_b$ to simplify the search. In fact, one can also learn the network  $a_t$ and $b_t$ (e.g., by minimising a divergence between forward and backward kernels, or via stochastic optimal control for tilting like [3]). In all these cases, RNC still provides computationally tractable SMC weights, whereas FKC is tied to a much more constrained subclass of $a_t$ and $b_t$ where certain divergence terms cancel.
>
> Moreover, RNC supports more flexible control applications than FKC. For example, we can freely combine annealing, reward tilting, and model composition for more complex tasks, and RNC continues to apply without changing the general formula. To highlight this in the revised manuscript, we include an additional maze "stitching" task: we have a diffusion model over short trajectories and seek to *stitch* these trajectories together under a global reward. This defines a composition + reward-tilting problem over path segments rather than a simple drift modification. RNC handles this task without changing its formula and obtains a success rate of 100%  (Tab. 3, Fig. 7 in our manuscript).
>
>
> [3] Domingo-Enrich, Carles, et al. "Adjoint matching: Fine-tuning flow and diffusion generative models with memoryless stochastic optimal control." ICLR 2025.

---

> ### Author Response · Authors · 2025-11-18
> **Reply to Reviewer TNy1 (part II)**
>
> > ### W4: The paper proposes extending RNE to Continuous Time Markov Chains for discrete diffusion, but lacks empirical validation in the experiments section.
>
> Thank you for pointing this out. **Following your suggestion, we have added new experiments on CTMC. In particular, we now include an ImageNet-256 experiment with a latent masked discrete diffusion model in our updated paper**. Using RNE in the CTMC setting, we perform prompt-based reward tilting with ImageReward as the reward.
>
> As shown in Fig. 8 of the updated manuscript, RNC significantly improves the alignment between generated images and target prompts. **This demonstrates that the CTMC extension is not only theoretically valid but also practically effective in a high-dimensional, discrete diffusion setting.**
>
> > ### Q1: Does the performance gain from RNC's flexibility consistently outweigh the tuning effort compared to methods like FKC? How can these parameters be chosen efficiently?
>
>
> In our ALDP experiments (Figs. 2 and 3), we observe that a good choice of $(c_a, c_b)$ transfers well across different batch sizes. This suggests a practical strategy: we can tune $(c_a, c_b)$ using a smaller batch size and a small number of batches, and then reuse the same values for larger runs with minimal additional cost.
>
>
> Moreover, RNE does not require $c_a$ and $c_b$  to be fixed across time. This allows for a simple greedy tuning strategy along the denoising trajectory. Concretely, on ALDP, we evaluated the following procedure: every 10 denoising steps, we perform a small grid search over  $c_a \in [1.2, 1.0, 0.8]$ and $c_b \in [-0.2, 0.0, 0.2]$ and select the pair that yields the lowest SMC weight variance. The resulting performance is as follows:
>
> |  | Energy TVD | Distance TVD | Sample W2 |
> |---|---|---|---|
> | FKC | 0.338 | 0.022 | 0.289 |
> | RNC ($c_a=1,c_b=0$) | 0.386 | 0.017 | 0.282 |
> | RNC-search | 0.127 | 0.012 | 0.275 |
>
> This shows that even a lightweight tuning procedure yields visible gains over FKC and over the untuned RNC baseline, indicating that the performance benefits of RNC’s flexibility do indeed outweigh the modest tuning effort required in practice.
>
> Additionally, as discussed above, RNE’s flexibility is not limited to the choice of $c_a, c_b$.
> It also supports learnable $a_t$ and $b_t$, and enables more flexible control tasks, which FKC cannot handle in general.
>
>
> > ### Q2: How does the proposed method handle the numerical instability issue arising from the schedules in diffusion models as $\sigma_t$ approaches 0?
>
> Following the convention of diffusion models, we do not run the process all the way to $\sigma_t = 0$. For example, in the ALDP experiments, we set the final step to $t = 10^{-3}$ and empirically observe that the RNC weights remain stable in these last few steps.
>
> For experiments like SBDD, we only apply SMC in the middle part of the denoising process. This is a common strategy also used in FKC. This is justified as the most important part of the generation happens in the middle of the trajectory; when $\sigma_t$ is very small, the samples change only slightly between steps. Therefore, we focus the control on the middle part and do not resample when $\sigma_t \rightarrow 0$.
>
>
>
> > ### Q3: How sensitive is RNC's performance to score model inaccuracies in practice? Are there theoretical principles...?
>
>
>
> We answer this question from both **theoretical** and **empirical** perspectives.
>
> - **Theoretically**, we have updated our manuscript with Proposition 3.2, where we characterise a bound on the error of SMC weights in terms of the time discretisation and the score network error. Specifically, assuming  $\\| \\nabla \\log p\_\\tau (\\hat Y\_{\\tau}) - s\_\\tau^{\\theta}(\\hat Y\_{\\tau}) \\|\_{L^2} \\leq \\epsilon\_{score}$.
>
>  we obtain
>
> \begin{align}
> \\| \\log w^{exact}(\\hat Y\_{[\\tau, \\tau']})  - \log w^{\\text{RNC}}\_{\\theta}(\\hat Y\_{[\\tau, \\tau']}) \\|\_{L^2} \leq E\\epsilon\_{score} + P' \\sqrt{\\Delta t}
> \end{align}
>
>   Thus, the error in the RNC log-weights scales **linearly** with the score network error $\\epsilon\_{score}$.
>
> - **Empirically**, we study the performance of RNC for inference-time annealing in Appendix F.4 (Fig. 16). To analyse the influence of an imperfect score, we early-stop the diffusion model training at different numbers of iterations and then evaluate RNC. For comparison, we also report a heuristic baseline that simply anneals the score without SMC. We observe that, while RNC can incur a larger bias when the network is relatively poorly trained, it still provides clear empirical performance gains over the heuristic baseline  (Fig. 16).
>
>
>
>
>
> ---
>
> We would like to thank you again for your thoughtful review, which has helped us significantly improve the quality and clarity of our manuscript. We would be very happy to discuss any further points you may have; if you find our replies satisfactory, we kindly invite you to reconsider your rating.

---

### Official Review · Reviewer_hh4L · 2025-10-28

**Soundness:** 3
**Presentation:** 2
**Contribution:** 2
**Rating:** 4
**Confidence:** 3

**Summary:**

This paper introduces the Radon–Nikodym Estimator (RNE) to enhance inference-time control and energy-based training for diffusion models. RNE expresses marginal density ratios along the diffusion trajectory as products of ratios between forward and reverse transition kernels across time. The central idea is that a diffusion process and its time-reversal induce the same path measure, yielding a Radon–Nikodym derivative of one; this identity links marginal densities to transition kernels. For inference-time control, RNE supplies importance weights within Sequential Monte Carlo (SMC), providing a plug-and-play recipe that unifies several prior approaches. For energy-based diffusion, RNE serves as a lightweight regularizer that enforces consistency between model-implied marginals and transition kernels during training.

**Strengths:**

Strengths:
1. Generality and unification. The framework applies broadly across settings by leveraging Bayes’ rule and time-reversal: it covers diverse inference-time control tasks (e.g., annealing, reward tilting, model composition) and extends beyond SDEs to continuous-time Markov chains (CTMCs), demonstrating genuine plug-and-play utility.
2. Comprehensive empirical evaluation. The method is validated on inference-time annealing (ALDP, LJ-13), model product for multi-target SBDD, scaling with particle count, and energy-based diffusion training (2D and 100D GMM, ALDP). The breadth and relevance of experiments substantiate the approach’s practicality.

**Weaknesses:**

Weaknesses:
1. Limited algorithmic novelty for control design. While RNE offers a unifying lens over existing methods, the new algorithmic insights for inference-time control are modest. The experimental instantiations primarily focus on drift reweighting, leaving broader design spaces underexplored.
2. Insufficient non-asymptotic guidance. Much of the theory is asymptotic, abstracting away time discretization and finite-particle effects. Clear prescriptions for step-size selection, resampling schedules, and particle budgets (M) are limited, leaving a potential gap between theory and practice in terms of variance, bias, and weight stability.
3. Convergence guarantees are implicit. The convergence narrative relies on standard Feynman–Kac SMC theory and RNE discretization analysis, but the paper does not present a single, unified theorem asserting that, as M→∞ and Δt→0, the particle system converges to q_t and ultimately q_0 under the proposed RNE-corrected scheme. Stating assumptions and conclusions explicitly in the main text would improve readability and reduce ambiguity about the limiting operator and target-matching guarantees.

**Questions:**

Questions:
One confusing issues are in Section 2.1.1. To ensure the reverse-process marginal matches q_t, and if one interprets the importance weight as a rejection rate, why do the authors use self-normalized importance resampling (SMC) rather than enforcing exact q_t via rejection at each step under a suitable design of a_t and b_t (guarantee the rejection rate)? Under self-normalized SMC, with a finite number of particles, the empirical distribution of X_τ may deviate from q_τ, and this error can accumulate over time. How should M be chosen to control this error in practice?

---

> ### Author Response · Authors · 2025-11-18
> **Reply to Reviewer hh4L (Part I)**
>
> Thank you for your insightful and constructive feedback. We are glad that you found our method to be general and unifying, and our experimental evaluation comprehensive. **Following your suggestions and questions, we have updated our manuscript with additional theoretical and empirical results**, which we believe substantially improve the clarity and rigour of the paper. We now respond to your points in detail:
>
> ---
> > ### Limited algorithmic novelty for control design. While RNE offers a unifying lens over existing methods, the new algorithmic insights for inference-time control are modest. The experimental instantiations primarily focus on drift reweighting, leaving broader design spaces underexplored.
>
>
>
> The RNE framework not only provides a flexible design space for drift choices, but it also enables more flexible control tasks than previous methods. To highlight this broader applicability, we demonstrate a maze-navigation "stitching" task, where we have a diffusion model over short trajectories and seek to *stitch* these trajectories together under a global reward. This defines a composition + reward-tilting problem over path segments rather than a simple drift modification. RNC handles this task without changing its formula and obtains a success rate of 100% (Tab. 3, Fig. 7 in our updated manuscript).
>
> Additionally, RNE provides a unified perspective for controlling masked diffusion models. We verify this on ImageNet-256 with a latent masked discrete diffusion model in Fig. 8 of our updated manuscript. Here, RNE is applied in the CTMC setting to perform prompt-based reward tilting, yielding strong alignment between generated images and prompts in a high-dimensional image-generation regime.
>
> Therefore, RNE is not limited to "drift reweighting" in a narrow sense: it offers a single macro-style recipe for SMC weights that can be instantiated for diverse control tasks across different modalities. Beyond control, RNE also unifies and connects a variety of methods in diffusion density estimation and energy-based training, which we consider an additional conceptual novelty.
>
>
>
> > ### Insufficient non-asymptotic guidance. Much of the theory is asymptotic, abstracting away time discretisation and finite-particle effects.
>
> Thank you for this valuable suggestion! In the revised manuscript, we added explicit non-asymptotic results.
>
>
>
> 1. **Non-asymptotic bounds for SMC weight with time discretisation and score imperfection**:
>
> Proposition 3.2  provides a non-asymptotic bound on the SMC weight calculated using the RNE estimator. To make this proposition align better with the practice, we also consider the score network is imperfect with
> $\\| \\nabla \\log p\_\\tau (\\hat Y\_{\\tau}) - s\_\\tau^{\\theta}(\\hat Y\_{\\tau}) \\|\_{L^2} \\leq \\epsilon\_{score}$.
>
> In this case, we have
>
> \begin{align}
> \\| \\log w^{exact}(\\hat Y\_{[\\tau, \\tau']})  - \log w^{\\text{RNC}}\_{\\theta}(\\hat Y\_{[\\tau, \\tau']}) \\|\_{L^2} \leq E\\epsilon\_{score} + P' \\sqrt{\\Delta t}
> \end{align}
>
> Where $w^{\mathrm{exact}}$ are unbiased discrete time importance weights (computed using the exact likelihoods) that enjoy the standard SMC guarantees.
>
> We want to highlight that this bound is possible thanks to our RNE framework, since it enables a connection between the standard SMC weight using a discrete-time kernel, and the continuous RND weights. This novel result is not present in prior works such as FKC, and it's not clear to us how to achieve this bound without our RNE framework.
>
>
> 2. **Convergence of the SMC estimator in continuous time with finite M**:
>
> Proposition H.1 shows that, under mild boundedness assumptions on the weight, for a bounded, $q_0$-measurable function $h$, we have
>
> \begin{align}
>       \mathbb{E} \\left[ \\left \\| \\int h(X_0 ) q_0(X_0)d X_0 - \\sum_{m=1}^M \\frac{w^{(m)}}{\\sum_{j=1}^M w^{(j)}} h(X_0^{(m)})     \\right \\|^2 \\right] \\leq C'M^{-1}
>     \end{align}
>
> where $q_0$ is our target density at time step $0$, and $\\{X_0^{(m)}, w^{(w)}\\}_{m=1}^M$ are the particles and their (unnormalised) weights returned at the last iteration of the sequential Monte Carlo algorithm with $M$ particles.
>
>
> 3. **Non-asymptotic bounds for RNE with time discretisation**:
>
> Additionally, beyond SMC, Proposition 3.1 provides a non-asymptotic bound on the RNE estimator itself:
>
> \begin{align}
> \\|  \\log R^\\nu_\\mu(Y_{[\\tau, \\tau']}) - \log R^N(\\hat Y_{[\\tau, \\tau']})  \\|_{L^2} \\leq \\mathcal{O}(\\sqrt{\\Delta t})
> \end{align}
>
> where $R^\\nu_\\mu(Y_{[\\tau, \\tau']})$ is the ideal RNE in continuous time, and $\\log R^N(\\hat{Y}_{[\\tau, \\tau']})$ is the practical estimator with with time discretisation.
>
>
> **Together, these new results provide a non-asymptotic guidance for our proposed algorithm, reducing the gap between theory and practice.**

---

> ### Author Response · Authors · 2025-11-18
> **Reply to Reviewer hh4L (Part II)**
>
> > ### Convergence guarantees are implicit ... Stating assumptions and conclusions explicitly in the main text would improve readability and reduce ambiguity about the limiting operator and target-matching guarantees.
>
>
>
> Thank you for this suggestion; we agree that making the convergence guarantees more explicit improves readability. Propositions 3.1, 3.2, and H.1 together imply that, under the stated assumptions, our RNE-corrected SMC scheme converges to the desired target as the step size $\Delta t \to 0$, the number of particles $M \to \infty$, and the score error $ \to 0$. We will clearly highlight this joint limit and its assumptions in our camera-ready version.
>
>
>
>
>
>
> > ### Question:  To ensure the reverse-process marginal matches q_t,... why do the authors use self-normalised importance resampling (SMC) rather than enforcing exact q_t via rejection at each step under a suitable design of a_t and b_t (guarantee the rejection rate)?
>
> Thank you for your questions. However, we do not think Rejection sampling is a good alternative in this case due to the following reasons:
>
> 1. **The upper bound of the density ratio is unknown.**
> Standard rejection sampling can be applied when you know the upper bound of the density ratio between the target and the proposal. However, in our case, this upper bound is not available.
>
>
> 2. **Rejection sampling is not more efficient than importance sampling.**
> Even if such an upper bound existed, rejection sampling does not achieve better efficiency in general. Assuming the proposal is $q$ and target is $p$, if we use A* sampling (a variation of rejection sampling which can achieves accurate samples using finite number of particles on average), we need on average $\Omega(2^{KL_2[q||p]})$ number of samples to accept one sample following $p$ exactly [1]. The number of particles required to achieve a small error is about $O(2^{KL_2[q||p]})$ [2,3].
> On the other hand, for importance sampling, the number of particles required to achieve a small error is also about $O(2^{KL_2[q||p]})$ [4].
>
>
>
> 3. **Rejection sampling can be less effective with few particles.**
> With a small number of particles, rejection sampling might lead to all proposals being rejected at a given step, yielding no improved samples. By contrast, although self-normalised SMC can be biased in this regime, it still preserves and amplifies particles with higher weights, providing approximated empirical improvements.
>
> Another potential "rejection" scheme is MCMC. However, it still suffers from a slow mixing issue and can be less effective with a few particles.
>
> [1] Maddison, Chris J., Daniel Tarlow, and Tom Minka. "A* sampling." NIPS 2014.
>
> [2] Havasi, Marton, Robert Peharz, and José Miguel Hernández-Lobato. "Minimal random code learning: Getting bits back from compressed model parameters." ICLR 2019.
>
> [3] Flamich, Gergely, Stratis Markou, and José Miguel Hernández-Lobato. "Fast relative entropy coding with a* coding." ICML 2022.
>
> [4] Chatterjee, Sourav, and Persi Diaconis. "The sample size required in importance sampling." The Annals of Applied Probability 28.2 (2018): 1099-1135.
>
>
>
> > ### Under self-normalised SMC, with a finite number of particles, the empirical distribution of X_τ may deviate from q_τ, and this error can accumulate over time. How should M be chosen to control this error in practice?
>
> In practice, we choose $M$ empirically depending on the task, and we did not tune this hyperparameter heavily in our experiments.
>
> 1. Distributional tasks (ALDP annealing).
>    For tasks where accurately matching the target distribution is important, such as ALDP annealing, we use a relatively large number of particles (e.g., M = 500). As shown in Tab. 1 and Fig. 3, this already yields accurate Boltzmann statistics, and increasing M further improves performance.
>
> 2. Goal-oriented tasks (SBDD, ImageReward tilting).
>    For tasks such as multi-target SBDD or ImageReward-based tilting, the main objective is to obtain *good solutions* rather than to perfectly match the full target distribution. In these settings, even a modest M (e.g., 32 in SBDD, following FKC, and 32 as well for ImageReward tilting) already leads RNC to markedly outperform the corresponding baselines.
>
>
> ---
>
> We thank you again for your thoughtful review and helpful suggestions. We believe the additional experiments and theoretical analysis following your suggestions have largely improved our paper quality. We would be very happy to clarify any remaining questions. Should you find our reply satisfactory, we kindly invite you to reconsider your rating.

---

### Official Review · Reviewer_WESr · 2025-10-31

**Soundness:** 3
**Presentation:** 3
**Contribution:** 3
**Rating:** 8
**Confidence:** 2

**Summary:**

The paper introduces the Radon–Nikodym Estimator (RNE). The authors make the observation that in diffusion, the forward and backward processes induce the same probability measure over the paths, which translates to their Radon–Nikodym derivative or, equivalently, their ratio, being equal to one. This allows them to calculate marginal densities using just the individual transition kernels, usually learned with a neural network, and cheap to compute. Using this estimator, the authors present different applications in inference-time control and regularization for training energy-based diffusion models.

**Strengths:**

- The proposed estimator provides a clean and elegant way to translate transition kernel probabilities to marginal densities. The idea of exploiting the intrinsic properties of the diffusion process to derive the estimator is novel and does not require any additional simulations, having potentially better scaling properties than previous approaches.

- From my understanding, the advantage of the RNE over the Feynman-Kac Corrector (FKC) is that, compared to FKC, RNE provides an easier way to compute the importance weights for inference-time control. For FKC, the weights are accumulated over the backwards trajectory, which may be problematic, especially when introducing the necessary resampling in Sequential Monte Carlo.

- Since there is no simulation required, the RNE can be computed for two neighboring timesteps using the forward and backward kernels, allowing the authors to utilize it in training an energy-based diffusion model. This extends the applicability of the proposed estimator to more applications than inference-time control.

**Weaknesses:**

- The main comparison regarding inference-time control seems to be with the Feynman-Kac Corrector. Although it is established that using density-ratio estimation scales badly with the number of dimensions, the FKC paper does showcase some results on high-dimensional data (images). The proposed RNE is only applied to low-dimensional settings.

**Questions:**

- Does the RNE compute a different weighting term than the Feynman–Kac formula? When are the two equivalent?

- How does the number of steps affect the Radon-Nikodym estimator? If the sampling is performed with 10 or 20 steps, then the backwards and forwards kernels will be significantly different. If I understand this correctly, the estimator relies on the two kernels being 'similar enough'.

- Are the importance weights for RNE "better" than the weights computed with FKC? Is there any intuition of why one should be better than the other? Do better importance weights work with fewer particles during sampling?

---

> ### Author Response · Authors · 2025-11-18
> **Reply to Reviewer WESr**
>
> Thank you for your insightful and constructive review, which helps us improve the quality of our paper. We are glad that you find our proposed framework clean, elegant, and novel. We address your concerns below:
>
> ---
> > ### Weakness: results on high-dimensional data
>
> Following your suggestion on Image experiments (and that of Reviewer TNy1 on CTMC), **we have added an extra experiment on high-dimensional data using ImageNet-256** with masked diffusion. More concretely, we consider a reward-tilting setup where the rewards are given by ImageReward defined via text prompts. In Fig. 8 of the updated manuscript, we visualise samples generated with and without RNC. We observe that RNC achieves strong alignment between the generated images and the target prompts, demonstrating both scalability to larger image-generation settings and the effectiveness of RNE on CTMCs.
>
>
> > ### Q1: Does the RNE compute a different weighting term than the Feynman–Kac formula? When are the two equivalent?
>
>
> Yes. RNE allows us to compute SMC weights for any choice of denoising and forward target processes (i.e., any $a_t$ and $b_t$ in Eqs. 13 and 14 of our manuscript). In contrast, FKC is equivalent to a special case of RNE with particular choices of $a_t$ and $b_t$. We state when they are equivalent in Proposition C.4.
>
> For other choices of $a_t$ and $b_t$, FKC cannot obtain the weights cheaply, whereas RNE remains computationally tractable and applicable.
>
>
> > ### Q2: How does the number of steps affect the Radon-Nikodym estimator?  If I understand this correctly, the estimator relies on the two kernels being 'similar enough'.
>
> Your understanding is correct. Following your question, we have updated our manuscript with both theoretical and empirical results on the effect of discretisation error.
>
> **Theoretically,** Proposition 3.1 in our updated manuscript states the convergence rate for RNE with respect to the number of discretisation steps.
>
> **Empirically,** we test the performance of RNC for inference-time annealing using different numbers of steps in Appendix F.4 (Fig. 15). For comparison, we report the heuristic choice by simply annealing the score without SMC. We can see that while RNC can have a larger bias when using a small number of steps, it still presents empirical performance gain compared to the heuristic choice.
>
>
>
>
> > ### Q3: Are the importance weights for RNE "better" than the weights computed with FKC? Is there any intuition of why one should be better than the other?  Do better importance weights work with fewer particles during sampling?
>
>
>
> RNE provides a more general and substantially broader design space than FKC. In particular, RNE allows us to freely choose the forward and backward kernels so that they are better aligned with each other and with the target dynamics. FKC corresponds to a specific, more constrained choice within this space, and can therefore be suboptimal in general.
>
> Intuitively, when the forward and backward kernels are better aligned, the resulting importance weights have lower variance. This directly improves the effective sample size (ESS) of the SMC sampler. As a result, for a fixed computational budget, RNE can achieve better performance with fewer particles than FKC in settings where the FKC choice of kernels is not optimal.
>
> ---
>
> We thank you again for your valuable feedback and would be happy to further discuss any additional questions or suggestions you may have.

---

> > ### Comment · Reviewer_WESr · 2025-11-26
> >
> > Thank you for the detailed responses.
> >
> > My main concern was scaling to higher dimensions, which is clearly shown to be possible in Figure 8.
> > The additional empirical results in Appendix F.4 are also helpful.
> >
> > Given that my score was already positive, I am inclined to keep it the same.

---

### Author Response · Authors · 2025-11-18
**Summary of reponses**

We thank all the reviewers for their insightful and constructive feedback. We have addressed all concerns point-by-point below and updated our manuscript according to the suggestions. In this global response, we summarise four key changes that we have made to our manuscript.

---

### **1. Theoretical guarantees: non-asymptotic bounds and convergence**

Following the suggestions of reviewers on more explicit convergence guarantees, we have added the following results:

- **Proposition 3.1 (time discretisation error for RNE):** A **non-asymptotic $L^2$ bound** for the RNE estimator under time discretisation, showing that the error between the ideal continuous-time RNE and its discretised counterpart scales as $\mathcal{O}(\sqrt{\Delta t})$.

- **Proposition 3.2 (RNE-based SMC weight error):** A **non-asymptotic bound for the SMC weights** computed via RNE, which explicitly accounts for both **time discretisation** and **score model error**. The error grows linearly in the score error, plus an additional $\mathcal{O}(\sqrt{\Delta t})$ term.

- **Proposition H.1 (convergence of SMC in the number of particles):** We state the convergence guarantee for the SMC algorithm, showing an $\mathcal{O}(1/M)$ decay of the error and almost-sure convergence as $M \to \infty$ under mild boundedness assumptions.

---


### **2. Experimental validation of RNE for CTMCs**



We added a new experiment on **ImageNet-256** with a **masked discrete diffusion model** to verify the applicability of our RNE for CTMC.
We consider a text-prompted reward-tilting task.   As shown in Fig. 8 of the updated manuscript, our method significantly improves alignment between generated images and text prompts, demonstrating that the CTMC extension of RNE is not only theoretically sound but also **effective in discrete diffusion in high-dimensional space**.

---

### **3. More flexible control task**

To highlight that RNE enables more general control tasks, we added a **maze-navigation "stitching" task**:

We train a diffusion model on **short trajectories**, and we seek to *stitch* these into long paths under a **global reward** enforcing endpoint and connectivity between short trajectories. This defines a **composition + reward-tilting problem over path segments**, which is not straightforward in previous works like FKC. RNC handles this task without changing the general SMC weight formula and achieves a 100% success rate (Tab. 3, Fig. 7).

Together with the CTMC results, this clarifies that RNE not only generalise previous approaches allowing different drifts: it provides a **single plug-and-play recipe that can be instantiated for diverse control tasks and modalities**.

---


### **4. Empirical study on score error, time discretisation, and denoising kernels**

We presented a empirical study on how RNE behaves under **score error**, **coarse time discretisation (few steps)**, and **different denoising kernels**:

- **Score error and number of steps (Appendix F.4):**
  On ALDP inference-time annealing, we vary the number of steps and the score quality (via early stopping in training). While RNC degrades when using very few steps or a poorly trained score, it still outperforms the heuristic baseline without SMC.

- **Effectiveness of the reference process with improved denoising kernels (Appendix F.5):**
  We compare standard EM discretisation vs. advanced denoising kernels with estimated variance, each with and without our reference process. We observe that better denoising kernels reduce the error of RNE, and the reference process provides a substantial additional gain in both cases. This verifies the effectiveness and robustness of our proposed reference process.

---

In summary, we have strengthened our theoretical contributions and extended the empirical scope of our experiments. Again, we sincerely thank all reviewers for their time and thoughtful feedback, which has helped us substantially improve our manuscript.

---

### Author Response · Authors · 2025-12-01
**Summary of Reviewers and Our Reply and Revisions**

We thank all reviewers again for the time and effort spent evaluating our manuscript, and we also thank the Area Chair for taking on additional responsibility in assessing our work. We have addressed each question and concern point-by-point in our rebuttal and have updated the manuscript accordingly with new theoretical results and substantial additional experiments.

**Below we concisely summarise the major concerns and our corresponding reply and updates.**

---


### **Reviewer WESr**
| Major Concerns                       | Our reply and update                                      |
|--------------------------------------|-----------------------------------------------------------|
| image experiments | add new ImageNet reward-tilting results (Fig. 8) |
| performance with fewer steps | conduct an ablation on number of steps (App. F.4, Fig. 15)




### **Reviewer hh4L**
 | Major Concerns                                                              | Our reply and update |
|-----------------------------------------------------------------------------|----------------------|
|  insufficient non-asymptotic guidance | add non-asymptotic bounds covering discretisation error, network error, and particle size (Prop. 3.1, 3.2, H.1)|
|convergence guarantees are not explicit  | make the overall convergence guarantee and assumptions explicit (Prop. H.1)|
| limited algorithmic novelty   | add a new "trajectory stitching" control task in which RNE-based SMC stitches short trajectories into long paths (Fig. 7), showing that RNE goes beyond flexible drifts and enables more control design |


### **Reviewer TNy1**

| Major Concerns                                                                                                         | Our reply and update |
|------------------------------------------------------------------------------------------------------------------------|----------------------|
| computation cost of reference process seems to be high  | clarify that the reference process adds almost no extra cost  |
| effectiveness of  reference process vs. better denoising kernels with estimated  denoising variance | add an ablation, showing that the reference process further improves accuracy even with advanced denoising kernels (App. F.5, Fig. 17)|
| advantage of RNE over FKC beyond tunable hyperparameters $c_a, c_b$ | clarify that RNE enables richer drift design spaces and more flexible control across task types and modalities (e.g., CTMC, stitching) |
| CTMC experiments |  add CTMC experiments on ImageNet (Fig. 8) |
| influence of imperfect score networks| add a theoretical bound on weight error under score error (Prop. 3.2); add an empirical ablation on score error (App. F.4, Fig. 16), showing RNE remains effective compared to heuristic baseline  |




### **Reviewer zcEE**

 | Major Concerns                                                                 | Our reply and update |
|--------------------------------------------------------------------------------|----------------------|
| SMC weights can have large variance when target is far from proposal| clarify that this issue is common to SMC, discuss it in the limitations section, and explain that RNE can mitigate it via more flexible proposal/target design |
| RNE–CFG formulation seems limited to a special case  | clarify that our CFG formulation is general and that the special case is only used to recover FKC |
|  derivations / math are dense| add short intuitive explanations before dense derivations (App. H.4) |

---

We hope this summary aids in the meta-review process.

---

### Meta-Review · Area_Chair_vEh6 · 2026-01-07

**Summary:**

Reviewers raised concerns about the scalability of the proposed method to high-dimensional settings, the gap between theoretical guarantees and practical performance, the computational cost relative to existing approaches, and potential issues of high variance and numerical instability. Some reviewers also questioned whether the method’s assumptions and estimation procedures remain reliable in realistic scenarios and whether comparisons with established methods are sufficient to contextualize the contribution.

Overall, the authors addressed many of these concerns by adding experiments on high-dimensional datasets such as ImageNet, strengthening the theoretical analysis with non-asymptotic bounds, and clarifying the computational simplicity of the approach. While some questions remain, the revisions were generally viewed positively, and the strengthened empirical and theoretical evidence supports the suggested decision.

**Reviewer Concerns:**

### Addressed

1. **Computational cost:**
   The authors clarified that the method does not rely on neural networks, resulting in minimal computational overhead.

2. **Theoretical guarantees:**
   Non-asymptotic theoretical results were provided to strengthen the connection between theory and practice.

3. **High-dimensional evaluation:**
   Additional experiments were conducted in high-dimensional settings to demonstrate the scalability and effectiveness of the approach.

### Outstanding

- **N/A**

**Reviewer Scores:**

Reviewer **WESr** raised concerns about the applicability of the proposed method to high-dimensional settings, particularly in comparison to FKC. In response, the authors added experiments on ImageNet to demonstrate the method’s potential in higher-dimensional cases. The reviewer also expressed an overall positive attitude, suggesting they are likely to remain positive.

Reviewer **hh4L** was concerned about the gap between theory and practice. The authors addressed this by adding experiments on maze navigation and ImageNet, and by introducing non-asymptotic propositions to bound the estimation error. These additions may sufficiently address the concern, and the reviewer is likely to raise their score.

Reviewer **TNy** questioned the computational cost of the method and the lack of comparisons with existing approaches. The authors clarified that the computation is simple, involves no neural networks, and therefore incurs minimal overhead, though no quantitative comparisons were provided. They also argued that their approach is orthogonal to the referenced methods and conducted additional experiments to support this claim. As a result, the reviewer may raise their score.

Reviewer **zcEE** pointed out potential issues of high variance and numerical instability introduced by the method. The authors acknowledged these issues and argued that similar approaches also suffer from such limitations. The reviewer responded that they would maintain a positive score.

---

### Decision · Program_Chairs · 2026-01-26

Accept (Poster)